# The DNA repair protein DNA-PKcs modulates synaptic plasticity via PSD-95 phosphorylation and stability

Cristiana Mollinari[1,2,10], Alessio Cardinale[3,10], Leonardo Lupacchini[4,10], Alberto Martire [5], Valentina Chiodi [5], Andrea Martinelli[6], Anna Maria Rinaldi[7], Massimo Fini[4], Simonetta Pazzaglia[8], Maria Rosaria Domenici [5], Enrico Garaci[4,9] & Daniela Merlo [1✉]

## Abstract

**The key DNA repair enzyme DNA-PKcs has several and important cellular functions. Loss of DNA-PKcs activity in mice has revealed essential roles in immune and nervous systems. In humans, DNA-PKcs is a critical factor for brain development and function since mutation of the prkdc gene causes severe neurological deficits such as microcephaly and seizures, predicting yet unknown roles of DNA-PKcs in neurons. Here we show that DNA-PKcs modulates synaptic plasticity. We demonstrate that DNA-PKcs localizes at synapses and phosphorylates PSD-95 at newly identified residues controlling PSD-95 protein stability. DNA-PKcs $-/-$ mice are characterized by impaired Long-Term Potentiation (LTP), changes in neuronal morphology, and reduced levels of postsynaptic proteins. A PSD-95 mutant that is constitutively phosphorylated rescues LTP impairment when over-expressed in DNA-PKcs $-/-$ mice. Our study identifies an emergent physiological function of DNA-PKcs in regulating neuronal plasticity, beyond genome stability.**

**Keywords** DNA-PKcs; Synaptic Plasticity; PSD-95 Phosphorylation; Cognitive Function; DNA Repair
**Subject Categories** DNA Replication, Recombination & Repair; Neuroscience

## Introduction

DNA-dependent protein kinase catalytic subunit (DNA-PKcs) is a ~460 kDa serine/threonine kinase, encoded by the *prkdc* gene, belonging to the phosphatidylinositol 3-kinase (PI3K) family (Smith et al, 1999). Its main function in the nucleus of eukaryotic cells is the repair of DNA double-strand breaks (DSBs) by non-homologous end joining (NHEJ) (Critchlow and Jackson, 1998). However, previous studies have shown that DNA-PKcs is abundant outside the nucleus, in the cytoplasm (Ferguson et al, 2012), cytoskeleton (Kotula et al, 2013), plasma membrane (Feng et al, 2004), and lipid rafts (Lucero et al, 2003).

In addition to its role in DNA DSB repair, DNA-PKcs is involved in several functions including oxidative stress response (Cardinale et al, 2022; Chen et al, 2023), transcription regulation (Goodwin et al, 2015), rRNA processing (Shao et al, 2020), telomere maintenance (Boulton and Jackson, 1998), metabolic gene regulation (Park et al, 2017) and protein stability (Cristini et al, 2016; Waldrip et al, 2021).

DNA-PKcs is abundantly expressed in mouse brain with high levels and activity in neurons and neural stem cells (Kashiwagi et al, 2018), suggesting a role of DNA-PKcs in neuronal functions. Indeed, the phenotypes of animals with defective DNA-PKcs indicate its essential role in the immune and nervous systems (Chechlacz et al, 2001; Gao et al, 1998; Gago-Fuentes and Oksenych, 2020; Jiang et al, 2015; Taccioli et al, 1998; Woodbine et al, 2013). In particular, mice expressing a catalytically inactive DNA-PKcs exhibit embryonic lethality correlated with severe neuronal apoptosis (Jiang et al, 2015), suggesting that DNA-PKcs during murine neurogenesis contributes to maintain genomic integrity in proliferating and post-mitotic neurons (Jiang et al, 2015). Moreover, DNA-PKcs is critical for human brain development and function since mutation in the *prkdc* gene causes severe neurological deficits (Woodbine et al, 2013), thus proposing a yet unknown role for DNA-PKcs related to structure/function of synapses and, in turn, to cognitive function (Merlo and Mollinari, 2023).

Postsynaptic density protein of 95 kDa (PSD-95), a member of the membrane-associated guanylate kinase (MAGUK) protein family 4, is the most abundant scaffold protein in the postsynaptic

[1]Istituto Superiore di Sanita', Department of Neuroscience, 00161 Rome, Italy. [2]Institute of Translational Pharmacology, National Research Council, 00133 Rome, Italy. [3]Holostem Srl, 41125 Modena, Italy. [4]IRCCS San Raffaele Roma, 00163 Rome, Italy. [5]Istituto Superiore di Sanita', National Centre for Drug Research and Evaluation, 00161 Rome, Italy. [6]Istituto Superiore di Sanita', Experimental Animal Welfare Sector, 00161 Rome, Italy. [7]Department of Systems Medicine, "Tor Vergata" University of Rome, 00133 Rome, Italy. [8]ENEA SSPT-TECS-TEB, Casaccia Research Center, Division of Health Protection Technology (TECS), Agenzia Nazionale per le Nuove Tecnologie, l'Energia e lo Sviluppo Economico Sostenibile (ENEA), 00123 Rome, Italy. [9]MEBIC Consortium, 00166 Rome, Italy. [10]These authors contributed equally: Cristiana Mollinari, Alessio Cardinale, Leonardo Lupacchini. ✉E-mail: daniela.merlo@iss.it

density of excitatory synapses (Chen et al, 2011). PSD-95 interacts with trans-membrane proteins such as receptors and ion channels and with cytosolic enzymes, playing a key role in synaptic transmission as well as in the formation and long-term stabilization of memory (Caly et al, 2019; Ehrlich and Malinow, 2004; Stein et al, 2003). Intensive studies of the mechanisms underlying memory formation and hippocampal long-term potentiation (LTP), a long-lasting increase of synaptic strength (Malenka and Nicoll, 1999), have delineated a central role for the phosphorylation of PSD-95 (Chetkovich et al, 2002; Steiner et al, 2008).

The intriguing findings that DNA-PKcs is a crucial enzyme during neuronal development and its deficiency is associated with altered neurological features, prompted us to further investigate DNA-PKcs role in neuronal functions. To this end, we used a mouse model deficient for DNA-PKcs to show that its kinase activity has a role in synaptic plasticity.

We find that in DNA-PKcs −/− mice LTP is impaired by lack of PSD-95 phosphorylation. Moreover, we show that DNA-PKcs phosphorylates PSD-95 at newly identified residues that control PSD-95 protein stability. Accordingly, PSD-95 mutants constitutively phosphorylated and remaining stable when over-expressed in DNA-PKcs −/− neurons, restore spine formation, dendritic architecture, and improve LTP impairment in vivo.

Altogether, these results provide novel evidence for DNA-PKcs activity-synaptic plasticity coupling via PSD-95 phosphorylation, thus reinforcing the emergent regulation of neuronal functions by DNA-PKcs beyond its role in DSB repair.

## Results

### Characterization of DNA-PKcs in brain and neurons

Previous works have shown that DNA-PKcs is highly expressed in mouse brain (Kashiwagi et al, 2018; Okawa et al, 2020), however its distribution in brain regions has not been investigated. We carried out Western blot (WB) analysis on cortical, hippocampal, cerebellar, and striatal regions of CD1 mice at 3 months of age and found that DNA-PKcs is expressed in these brain areas at comparable levels (Fig. 1A).

We then investigated the subcellular localization of DNA-PKcs in mouse cortical neurons (DIV 21), performing a biochemical fractionation (Cardinale et al, 2012; Dunah and Standaert, 2001). Comparing to the amount of DNA-PKcs in the starting total protein extract, we estimated the proportion of the kinase at the different subcellular compartments and found that in primary neurons DNA-PKcs is present in nuclei (P1, $21.57 \pm 4.4\%$), light membrane compartment (P3, $31.65 \pm 4.8\%$) and cytosol (S3, $29.13 \pm 6\%$). Interestingly, DNA-PKcs is present in the synaptosomal membrane fraction (LP1, $13.28 \pm 1.45\%$) and poorly detected in synaptic vesicle-enriched fraction (LP2, $5.43 \pm 0.93\%$), but absent from soluble subcellular fraction (LS2) (Fig. 1B). The specificity of the fractionation procedure was evaluated by using specific markers for subcellular compartments: Lamin A/C for the nuclear fraction (P1), PSD-95 for the synaptosomal membrane fraction (LP1) and Synapsin I for the synaptic vesicle fraction (LP2) (Fig. 1B).

By immunofluorescence confocal microscopy, we show that, in mouse primary cortical and hippocampal cultures (DIV 21), DNA-PKcs elicits a strong punctate staining mainly distributed in the cell soma and along dendrites of neurons labeled with anti-MAP2 antibody (Figs. 1C and EV1), resembling the distribution of synaptic proteins. Although we noticed a certain variability of DNA-PKcs nuclear staining in post-mitotic neurons, the nuclear labeling was not as strong as in proliferating cells including human neural progenitor cells (hNPCs) and human embryonic kidney 293 (HEK293) cells (Fig. EV1). DNA-PKcs antibody specificity was confirmed by the absence of immunofluorescence following antibody neutralization by incubation with an excess of recombinant DNA-PKcs protein (Fig. 1C, lower panel). Because DNA-PKcs staining has an expression pattern similar to synaptic proteins, we measured the degree of co-localization between DNA-PKcs and several pre and postsynaptic proteins using THUNDER Imager microscopy (Figs. 1D and EV2). We found that DNA-PKcs in neurites partially co-localizes with Synapsin I ($24.4 \pm 1.44\%$ of DNA-PKcs puncta co-localization), Syntaxin I ($21.5 \pm 2.1\%$) and shows a higher co-localization with PSD-95 ($39.5 \pm 3.13\%$), GluN1 ($30.9 \pm 1.73\%$) and GluA1 ($33 \pm 1.73\%$) (Fig. 1D). It is notable that DNA-PKcs co-localizes mostly to the periphery of presynaptic proteins, whereas co-localization with postsynaptic proteins, particularly with PSD-95, appears in close apposition (Fig. 1D insets: white dotted edges in the red channels highlight the position of DNA-PKcs protein with respect to synaptic proteins).

The localization of DNA-PKcs in synaptosomal membrane fraction raised the question regarding its role in this compartment. Since it was previously reported that DNA-PKcs activity in mouse is one-tenth respect to humans (Anderson and Lees-Miller, 1992; Vemuri et al, 2001), we performed a kinase assay in both human and mouse LP1 fractions (Cardinale et al, 2012) and we detected DNA-PKcs activity in LP1 from both human and mouse cortex (Fig. 1F).

### DNA-PKcs protein increases in synaptosomal membranes in response to synaptic activity and interacts with postsynaptic proteins

Synaptic activity can be stimulated in primary neuronal cultures by forskolin (50 μM) plus rolipram (0.1 μM) (F/R), resulting in prolonged LTP (chemical LTP, cLTP) (Mollinari et al, 2015; Oh et al, 2006; Otmakhov et al, 2004). As shown in Fig. 2A, after F/R stimulation, DNA-PKcs staining is visibly increased in neurites proximal to the cell body (see red arrows). Histogram data in Fig. 2A represent the mean intensity of DNA-PKcs immunolabelling in DMSO (control) and F/R treated cells (DMSO $105.8 \pm 8.9$; F/R $236.3 \pm 7.9$; $p < 0.05$; $n = 3$), measured in neurites. Interestingly, WB analyses showed no increase of DNA-PKcs protein on total neuronal extracts following F/R treatment (Fig. 2B), suggesting that the protein does not change its total expression level rather it changes localization following synaptic stimulation. To confirm that increased synaptic activity by F/R treatment in primary cortical neurons modulates the expression of DNA-PKcs specifically in synaptosomal membranes, we performed WB analysis after subcellular fractionation (Fig. 2C). Differently from DNA-PKcs present in P3 fraction (DMSO $100 \pm 11.7\%$; F/R $131 \pm 20\%$; $p = 0.241$; $n = 3$), almost undetectable in S3 fraction, DNA-PKcs is significantly increased in LP1 fraction after synaptic activity (DMSO $100 \pm 12.3\%$; F/R $273 \pm 17\%$; $p < 0.05$; $n = 3$) (Fig. 2C).

Next, we carried out co-immunoprecipitation experiments from mouse cortical synaptosomal membranes. WB analysis in Fig. 2D

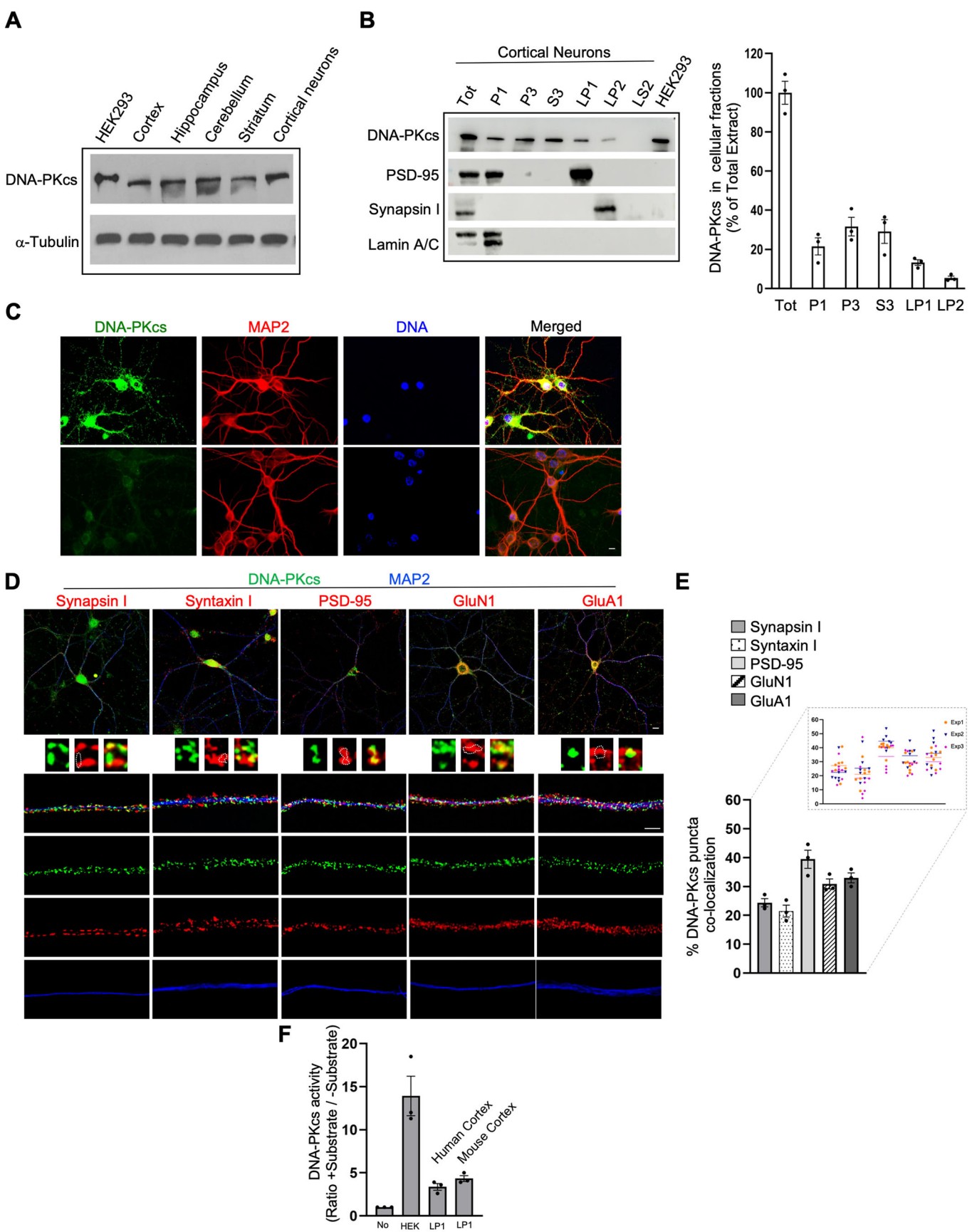

◄

**Figure 1. DNA-PKcs is localized in synaptosomal membranes where it exerts kinase activity.**

(A) Representative Western Blot analysis showing the expression of DNA-PKcs protein in different mouse brain regions at similar levels and in mouse primary cortical neurons. HEK293 cells were used as positive control. α-Tubulin was used as a loading control. (B) Western Blots representing the biochemical fractionation of mouse cortical neurons (DIV 21) showing the synaptic localization (LP1) of DNA-PKcs. Equal amounts of protein (70 μg) were loaded for each fraction ($n = 3$ independent experiments). Fractions were loaded on the gel and the specificity of the fractionation procedure was confirmed by using specific markers for subcellular compartments: Lamin A/C for the nuclear fraction (P1), PSD-95 for the synaptosomal membrane fraction (LP1) and Synapsin I for the synaptic vesicle fraction (LP2). Histogram data represent the percentage with respect to a total protein extract of DNA-PKcs in the different cellular fractions. Results are expressed as mean ± SEM. (C) Representative confocal fluorescence images of mouse primary cortical neurons (DIV 21) labeled with the anti-DNA-PKcs antibody (green channel), the neuronal marker MAP2 (red channel), and DNA dye (blue, DAPI). DNA-PKcs is strongly expressed in neurons and is distributed both in the cell soma and dendrites. DNA-PKcs antibody specificity is confirmed by the absence of immunofluorescence when the antibody is preincubated with an excess of recombinant DNA-PKcs protein (lower panel). Scale Bar 5 μm. (D) Representative triple immunofluorescence THUNDER images of mouse primary cortical neurons (DIV 21) showing the distribution of DNA-PKcs (green channel) similar to synaptic proteins such as Synapsin I, Syntaxin I, PSD-95, GluN1, and GluA1 (red channel). Scale Bar 5 μm. Images of neurites from a single neuron show the punctate co-localization of DNA-PKcs with pre- and postsynaptic proteins. Single-channel images are provided to better evaluate the localization of each protein. Insets represent enlargements of a dendritic tract showing DNA-PKcs co-localization with pre- and postsynaptic proteins. White dotted edges in the red channels highlight the position of DNA-PKcs protein with respect to synaptic proteins. Scale Bar 2 μm. (E) Histogram data represent the percentage of DNA-PKcs puncta co-localization with the different synaptic proteins analyzed. Results are expressed as mean ± SEM; $n = 3$ independent experiments. Data distribution is shown in the enlargement; $n = 21$ neurons/each synaptic marker. (F) DNA-PKcs kinase activity assay performed using human and mouse cortical membrane fractions. Protein extracts from HEK293 cells (200 μg), LP1 human cortex fraction (200 μg), and mouse LP1 fraction (1 mg) were subject to DNA-PKcs immunoprecipitation and the phosphorylation assay was performed. The assay performed without protein extract was used as a negative control. Data were expressed as ratio of values in presence/absence of the DNA-PKcs substrate p53 (presented as mean values ± SEM of kinase activity; $n = 3$ independent experiments). P1, nuclei and large debris; S3, cytosolic fraction; P3, light membrane fraction; LP1, synaptosomal membrane fraction; LP2, synaptic vesicle-enriched fraction; LS2, supernatant from LP2; IP, immunoprecipitated. Source data are available online for this figure.

shows the 460 kDa band corresponding to the full-length DNA-PKcs in both pulldown (IP) and input samples. As shown in Fig. 2E, we next identified that PSD-95, GluN1 and GluN2A/B co-immunoprecipitate with DNA-PKcs in LP1-enriched fractions from mouse cortex, whereas we failed to co-immunoprecipitate GluA1.

These findings indicate that DNA-PKcs expression in neurons responds to synaptic stimuli and that postsynaptic proteins might be novel binding partners of DNA-PKcs in synaptic membranes.

## DNA-PKcs −/− mice show impairment in synaptic transmission, long-term potentiation, and postsynaptic protein expression

To unravel the role of the DNA-PKcs activity at synapses, we exploited DNA-PKcs −/− mice, an animal model completely devoid of DNA-PKcs activity due to disruption of the catalytic domain of the kinase (Taccioli et al, 1998). DNA-PKcs −/− mice also show a strong reduction of DNA-PKcs protein levels, because of the ablation of 90 amino acids spanning the kinase domain. WB in Fig. 3A confirmed that DNA-PKcs is undetectable in the hippocampus of DNA-PKcs −/− mice, in accordance with previous work (Taccioli et al, 1998). Moreover, the lack of DNA-PKcs labeling by immunofluorescence in primary cortical and hippocampal cultures from DNA-PKcs −/− mice as compared with wild-type (WT) cultures, further excluded unspecific labeling of the DNA-PKcs antibody (Fig. 3B).

We performed electrophysiological analysis, at 8 weeks of age, by recording extracellular field potentials in the CA1 area of hippocampal slices, after stimulation of the Schaffer collaterals (Fig. 3C). We evaluated basal synaptic transmission in DNA-PKcs −/− compared to WT mice. We demonstrated a clear difference in I/O curves between the two experimental groups, with a significantly lower averaged fEPSP slope in DNA-PKcs −/− ($n = 10$) with respect to WT mice ($n = 12$) at almost every stimulus intensity (Fig. 3D), indicating an impairment in basal synaptic transmission in DNA-PKcs −/− mice ($^*p < 0.01$, $°p < 0.05$). No differences in the paired-pulse ratio (PPR) were found between the two groups (WT: 1.871 ± 0.099, $n = 11$; DNA-PKcs −/−: 1.785 ± 0.121, $n = 8$; $p = 0.351$) (Fig. 3E).

We then evaluated the induction of long-term potentiation (LTP), by stimulating Schaffer collaterals with a high-frequency stimulation (HFS) protocol consisting of two trains of 100 pulses at 100 Hz, 20 s apart. In WT mice ($n = 16$), HFS-induced LTP, measured 50 min after HFS, as a potentiation of fEPSP slope (176.30 ± 14.25% of basal value). On the contrary, in DNA-PKcs −/− mice ($n = 9$), HFS did not potentiate fEPSP (96.61 ± 5.32% of the basal value with respect to WT) (WT vs DNA-PKcs −/− $p < 0.001$) (Fig. 3F,G).

We then investigated whether alterations of LTP found in DNA-PKcs −/− mice were associated with changes in signaling pathways and postsynaptic receptors linked to this form of synaptic plasticity. After electrophysiology recordings, a single slice from a single mouse (WT or DNA-PKcs −/−) was rapidly dissected to isolate the CA1 region and frozen. Frozen CA1 regions were homogenized and protein lysates analyzed by WB to study phosphorylation changes of proteins playing a major role in LTP such as CaMKII alpha, ERK1/2, PKB/Akt, at different time points after delivering of HFS (Racaniello et al, 2010) (Fig. 3H). We found that pCaMKII alpha immunoreactivity, normalized to the total protein, strongly increases 5 min after tetanization and does not show significant differences in the two groups of animals. The increased phosphorylation is similarly sustained for at least 15 min after LTP induction. The immunoreactive levels of activated ERK2 kinase, normalized to the total protein, indicates that 5 min LTP increases ERK2 phosphorylation that remains sustained for at least 15 min. We previously published that Akt phosphorylation at Thr308 is not significantly changed after LTP in our experimental conditions (Racaniello et al, 2010). Here we show that in WT or DNA-PKcs −/− slices, LTP induction causes an increase of Akt phosphorylation at Ser473, normalized to the total protein, that is prolonged for at least 15 min (Fig. 3H). Although it has been shown that DNA-PKcs phosphorylates Akt at Ser473 (Feng et al, 2004), this result is not surprising since it is known from the literature that there are different kinases directly phosphorylating S473-Akt during LTP (Henry et al, 2017; Liu et al, 2014).

Next, we evaluated the levels of postsynaptic proteins in total extracts and LP1 fractions from the cortex and hippocampus of WT

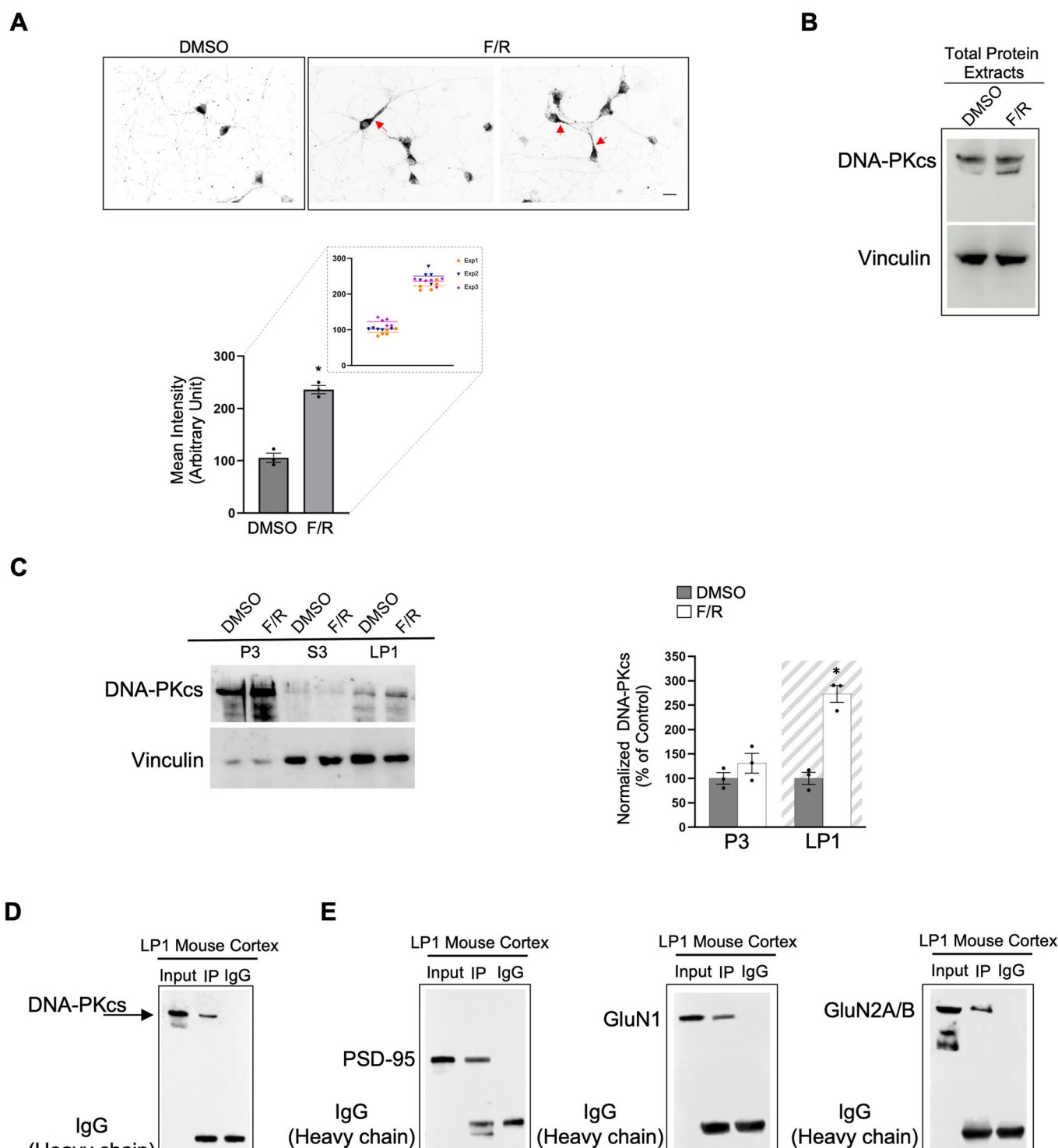

(n = 3) and DNA-PKcs −/− mice (n = 3) by WB analysis (Fig. 4A). We found no differences in expression levels in any of the postsynaptic proteins analyzed between WT and DNA-PKcs −/− total extracts. Interestingly, in LP1 from cortical tissues, we found 57.5% decrease in PSD-95 levels in DNA-PKcs −/− compared to WT mice; GluN1 levels are decreased by 40.5%; those of GluA1 by 40%; and those of GluN2A by 33.3% (Fig. 4A). Similarly, in LP1

from hippocampal tissues we found 58.2% decrease in PSD-95 levels in DNA-PKcs −/− mice compared to WT; GluN1 levels are decreased by 43.6%; those of GluA1 by 37%; and those of GluN2A by 30% (Fig. 4A). We also confirmed by immunofluorescence THUNDER microscopy a 52.2% decreased expression of PSD-95 (WT: 100 ± 7.9%, DNA-PKcs −/−: 47.8 ± 5.1%; p < 0.05; n = 3) (Fig. 4B) and a 44.4% decrement of GluA1 (WT: 100 ± 6.14%,

**Figure 2. DNA-PKcs increases in synaptosomal membrane fraction following synaptic stimuli and is associated with postsynaptic proteins.**

(A) Immunofluorescence images of mouse primary cortical neurons (DIV 9), labeled with an anti-DNA-PKcs antibody, unstimulated (DMSO), or after chemical LTP induced by forskolin/rolipram stimulation (F/R). F/R treatment induces an increase of DNA-PKcs protein levels in neurites proximal to the cell body (red arrows). Histogram data represent the mean intensity of DNA-PKcs immunolabelling in DMSO and F/R treated cells. Bars in the plots represent means ± SEM ($n = 3$ independent experiments, *$p < 0.05$ Statistics by Student's $t$-test). Data distribution is shown in the enlargement; $n = 15$ neurons/each treatment. Scale Bar 5 μm. (B) Western Blot showing no increase of DNA-PKcs in total protein extract from primary cortical neurons after F/R stimulation. Vinculin was used as a loading control. (C) Representative Western Blot showing the biochemical fractionation of mouse primary cortical neurons (DIV 9) unstimulated (DMSO) or after F/R treatment. Following F/R stimulation, DNA-PKcs is enriched in the synaptosomal membrane fraction (LP1). Vinculin was used as a loading control. Histogram data represent the percent change in DNA-PKcs band intensity in P3 and LP1 (gray background) fractions in F/R stimulated cells with respect to DMSO-treated cultures. Statistics by Student's $t$-test (unpaired, two-tailed) for F/R vs DMSO in P3 fraction ($p = 0.241$; $n = 3$ independent experiments) and for F/R vs DMSO in LP1 fraction (*$p < 0.05$; $n = 3$ independent experiments). Bars in the plots represent means ± SEM. (D) Immunoprecipitation of DNA-PKcs from LP1 (500 μg) mouse cortex. Western Blot analysis using the anti-DNA-PKcs antibody shows the full-length protein in the Input and IP lanes. IgG control antibody isotype is also shown ($n = 3$ independent experiments). (E) Immunoprecipitation of DNA-PKcs from LP1 mouse cortex. Western Blot analysis using anti-PSD-95, anti-GluN1, and anti-GluN2A/B antibodies indicates that DNA-PKcs pulls down the three postsynaptic proteins (IP lanes). IgG were loaded as non-specific antibody IP control ($n = 3$ independent experiments). Source data are available online for this figure.

DNA-PKcs −/−: 55.6 ± 4.51%; $p < 0.05$; $n = 3$) (Fig. 4C) in DIV 21 cortical cultures from DNA-PKcs −/− mice as compared with WT.

The incorporation of GluA1-containing AMPARs into synapses is essential to several forms of neuronal plasticity, including LTP (Malinow and Malenka, 2002). The reduction of both LTP and GluA1 protein in synaptosomal membranes from DNA-PKcs −/− mice prompted us to analyze GluA1 trafficking at the synapses following stimulation. First, we evaluated the presence of this subunit into LP1 of cortical cultures following cLTP induction (Mollinari et al, 2015; Otmakhov et al, 2004). We show that F/R 10 min treatment of primary cortical cultures increases GluA1 content in LP1 by 2.97 ± 0.35 and 3.4 ± 0.35 folds in WT and DNA-PKcs −/− cultures respectively ($p = 0.1579$; $n = 4$) (Fig. 4D). These results indicate that, although DNA-PKcs −/− neurons have a lower basal level of GluA1, following 10 min cLTP induction, its amount into LP1 fraction of DNA-PKcs −/− neurons is guaranteed as efficiently as in control neurons (Fig. 4D). Since electrophysiological data on hippocampal slices show that DNA-PKcs −/− mice have alterations of LTP, we then looked at GluA1 trafficking at the synapses at longer time (40 min after HFS). We assessed levels of surface GluA1 using cell-surface biotinylation of hippocampal slices before and 40 min after HFS. Steady-state (T0) surface biotinylation of the slices, normalized to total GluA1 levels, showed that DNA-PKcs −/− mice have a 48.16 ± 3.21% reduced surface levels of GluA1 as compared with WT mice ($p < 0.01$; $n = 3$) (Fig. 4E). HFS application increases the amount of cell-surface GluA1 by 2.51 ± 0.2 folds in WT mice ($p < 0.01$; $n = 3$) whereas cell-surface GluA1 levels remain constant in DNA-PKcs −/− tetanized slices ($p = 0.3791$; $n = 3$) (Fig. 4E). These results indicate that DNA-PKcs −/− mice show defects in GluA1 delivery to synapses evident at longer times after LTP induction.

## Cortical neurons from DNA-PKcs −/− mice show less spines and neurite complexity

We then investigated neuronal viability, dendrite complexity, and spine number in primary cortical neurons from WT and DNA-PKcs −/− mice. Although DNA-PKcs −/− cultures do not show differences in the number of apoptotic nuclei as compared with WT (Fig. 5A), neurons appear morphologically different from WT at both DIV 9 and DIV 21, showing a significant reduction in dendrite complexity, as indicated by immunofluorescence staining for the dendritic marker MAP2 (Fig. 5B) and Sholl analysis ($p < 0.05$;

$p < 0.001$, and $p < 0.0001$; $n = 10$ neurons for three independent experiments) (Fig. 5C). However, there is no significant difference between the two experimental groups in the average length of neurites (Fig. 5D) (WT: 55.8 ± 3.1 μm, DNA-PKcs −/−: 51.1 ± 2.9 μm; ($p = 0.335$); $n = 3$ experiments).

To better visualize neuronal spine morphology, we over-expressed the enhanced green fluorescent protein (EGFP) by recombinant adeno-associated Virus (rAAV) infection in WT and DNA-PKcs −/− primary cortical neurons (Chamberlin et al, 1998). DNA-PKcs −/− neurons (DIV 21) are confirmed to possess a reduced structural complexity and number of spines along the neurites (WT: 0.77 ± 0.054, DNA-PKcs −/−: 0.25 ± 0.03; $p < 0.005$; $n = 10$ neurons analyzed for both experimental groups, repeated for three independent batches of cultures). The results show that DNA-PKcs −/− primary neurons have significantly fewer spines and exhibit more spines with immature shapes than WT neurons (Fig. 5E).

Altogether, the results described so far point towards a role of DNA-PKcs in the modulation of synaptic plasticity associated with changes in neuronal morphology.

## DNA-PKcs phosphorylates PSD-95 in vitro

Protein phosphorylation has a key role in the regulation of a variety of synaptic functions (Turner et al, 1999), including LTP. To gain insight into the function of DNA-PKcs in LTP, we performed a phospho explorer antibody array (https://www.fullmoonbio.com/product/phospho-explorer-antibody-array/) (Dataset EV1) on protein extracts from CA1 dissected region of unstimulated (T0) and tetanized (T5') slices from WT and DNA-PKcs −/− mice (Fig. 6A), to identify novel DNA-PKcs target/s involved in LTP.

We confirmed the increased phosphorylation level of previously analyzed proteins including CaMKII alpha, ERK1/2, and PKB/Akt in both groups of animals. Among the newly identified proteins, p90rsk shows reduced phosphorylation following tetanization in DNA-PKcs −/− slices (Dataset EV1). In addition, previously published results showed that DNA-PKcs promote activation of the mitogen-activated protein kinase (MAPK)/p90rsk signaling cascade (Panta et al, 2004). We then performed an in vitro kinase assay by using human recombinant DNA-PKcs and tested its kinase activity on selected proteins based on results obtained from co-immunoprecipitation experiments and phospho array indications. We examined human recombinant proteins including ERK2, p90rsk, PSD-95, GluN1, GluA1, and p53 (as positive control). As

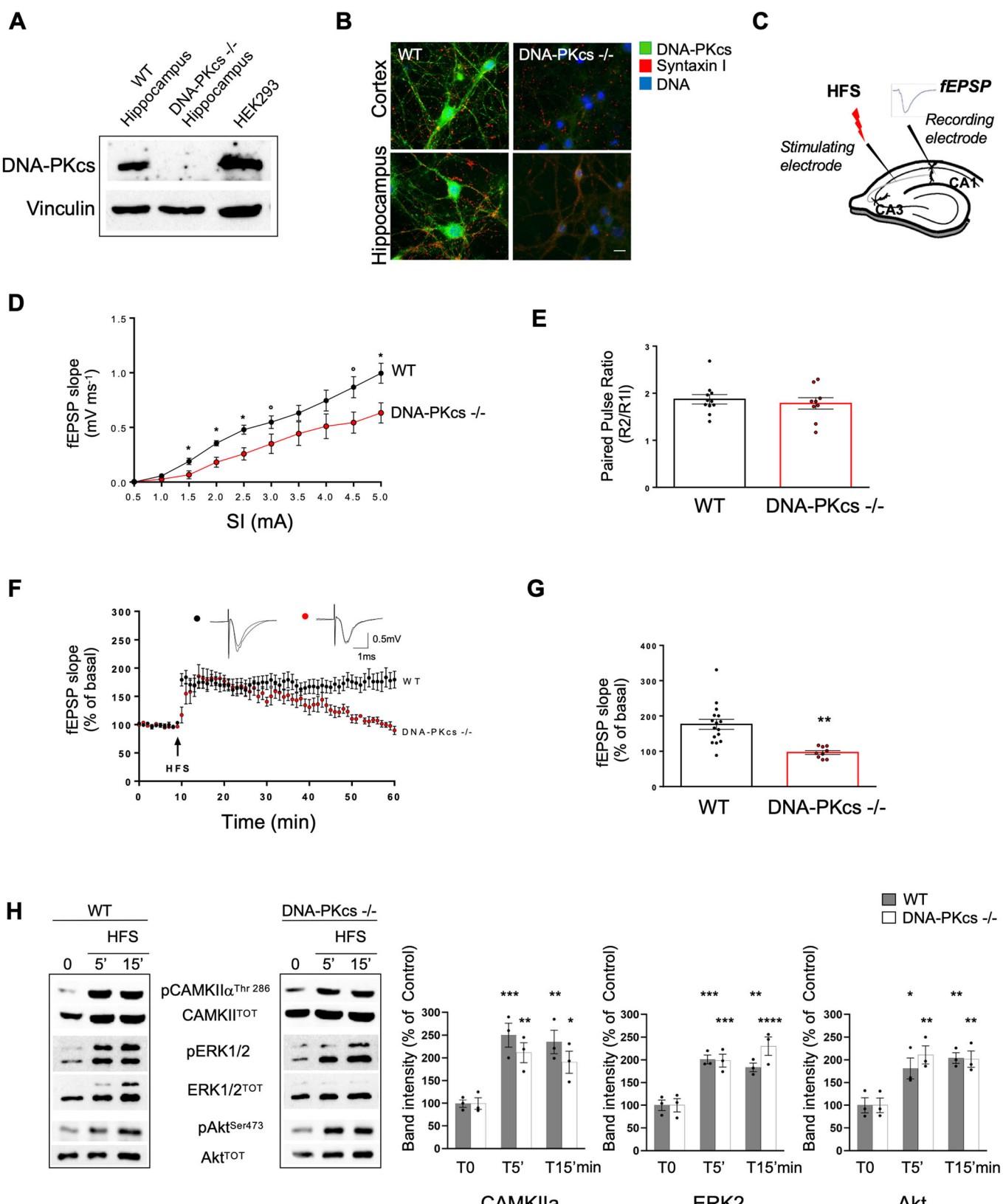

**Figure 3. DNA-PKcs −/− mice show impaired basal synaptic transmission and long-term potentiation with normal signaling pathways.**

(A) Western Blot showing that DNA-PKcs is undetectable in the hippocampus of DNA-PKcs −/− mice. HEK293 cells were used as positive control. Vinculin was used as a loading control. ($n = 3$ mice). (B) Triple immunofluorescence images of cortical and hippocampal neurons (DIV 21) showing the lack of DNA-PKcs labeling in mutant mice confirming the specificity of the anti-DNA-PKcs antibody. Scale Bar 5 µm. ($n = 3$ cultures). (C) Schematic representation of hippocampal slice showing stimulating and recording electrode positions. (D) Input-output (I/O) curve of fEPSP slope (mV/ms) versus stimulus (mA) at the Schaffer collaterals-CA1 pyramidal cell synapse in WT and DNA-PKcs −/− mice. Each point on the I/O curve was obtained by averaging responses over 2–5 min of recording and progressively increasing the stimulus strength. A reduction in I/O ratio is shown in DNA-PKcs −/− ($n = 10$) with respect to WT ($n = 12$) mice (*$p < 0.01$ and °$p < 0.05$; Mann–Whitney U-test). Error bars indicate SEM. (E) Comparable paired-pulse stimulation ratio (PPR) in WT ($n = 11$) and DNA-PKcs −/− ($n = 9$) mice ($p = 0.351$; Mann–Whitney U-test). Error bars indicate SEM. (F) DNA-PKcs −/− slices show an impairment in LTP. Time courses of fEPSP slope after HFS; data are expressed as mean ± SEM of $n = 16$ (WT) and $n = 9$ (DNA-PKcs −/−) slices (one slice tested per experiment). Slices were obtained from at least four mice for each experimental set. Insets show fEPSP recorded in basal condition and 60 min after HFS. (G) The graph summarizes the LTP magnitude in the two genotypes. LTP is expressed as the mean percentage variation of the slope from baseline (calculated in the time windows 5–10 min) and after HFS (calculated in the time windows 50–60 min). **$p < 0.001$ by Mann–Whitney U-test. Error bars indicate SEM. (H) Representative Western blots of phosphorylated and total forms of CaMKIIα, ERK1/2, and Akt at different times after delivery of HFS. Densitometric quantification of the immunoreactive bands indicates no significant differences in phosphorylation levels of these proteins between WT and DNA-PKcs −/− CA1 slices at the different time points analyzed (T5′ CaMKIIα: WT: 250 ± 26.5%, DNA-PKcs −/−: 211 ± 22%; $p = 0.529$; $n = 3$ mice; T15′ CaMKIIα: WT: 235 ± 26%, DNA-PKcs−/−: 190 ± 24.2%; $p = 0.399$; $n = 3$ mice; T5′ ERK2: WT: 201 ± 9.5%, DNA-PKcs−/−: 198 ± 14.1%; $p = 0.998$; $n = 3$ mice; T15′ ERK2: WT: 184 ± 9.5%, DNA-PKcs −/−: 230 ± 20.4%; $p = 0.100$; $n = 3$ mice; T5′ Akt: WT: 181 ± 23.1%, DNA-PKcs −/−: 211 ± 20%; $p = 0.591$; $n = 3$ mice; T15′: WT: 204 ± 12%, DNA-PKcs −/−: 201 ± 18%; $p = 0.999$; $n = 3$ mice). Values represent the percent changes in protein phosphorylation, normalized to the total protein, with respect to control (T0) for each time point. *$p < 0.05$ vs control values (T0), **$p < 0.005$ vs control values (T0), ***$p < 0.001$ vs control values (T0), ****$p < 0.0001$ vs control values (T0). Statistics by two-way ANOVA followed by Tukey's post hoc analysis. Results are expressed as mean ± SEM; $n = 3$ independent experiments. HSF high-frequency stimulation, fEPSP field excitatory postsynaptic potential, PPF paired-pulse facilitation, LTP long-term potentiation. Source data are available online for this figure.

shown in Fig. 6B, PSD-95 and p53 elicit specific radioactive incorporation of [$^{33}$Pγ-ATP] since the corresponding band, revealed by autoradiography, disappears when the assay is performed in the absence of DNA-PKcs.

## Identification of DNA-PKcs phosphorylation sites in PSD-95 protein

To identify residues in PSD-95 phosphorylated by DNA-PKcs, we performed Nanoscale liquid chromatography coupled to tandem mass spectrometry (Nano LC-MS/MS) analysis (Fig. 6C) after an in vitro kinase assay as described above but in the absence of the radioligand.

The kinase assay was carried out with or without DNA-PKcs and samples were cleaved by enzymatic digestion and enriched with TiO2 beads (Fig. 6C). Among the 31 identified phosphorylated clusters with a confidence on phosphosite assignment higher than 0.75, we observed, 6 of them, an increase of their abundance in presence of kinase (Appendix Figs. S2–S10). The 6 phosphorylated residues in PSD-95 are localized in different regions of the protein and are conserved in mammals (Fig. 6D,E). Among these, only two sites fall within the DNA-PKcs kinase recognition motif SQ (S308 and S556), but not the others (T87, S297, S521, and S523) (Fig. 6D). Interestingly, the newly identified sites phosphorylated by DNA-PKcs have a conserved short motif of two serines or threonines separated by one amino acid (Ser-X-Ser (SXS); Thr-X-Thr (TXT)) and are highly conserved (Fig. 6D–F). The SXS motif is reminiscent of Egr1 S301 phosphorylation site by DNA-PKcs (Waldrip et al, 2021).

As far as we know, specific residues in PSD-95 phosphorylated by DNA-PKcs have not been previously described.

## Phosphorylation at S308 and T87 of PSD-95 is required for its protein stability

We generated phosphorylation site-specific antibodies by selecting peptides within PSD-95 that are conserved in mammals (Fig. 6G). We focused on the phosphorylation of S308 and T87 since mass spectrometry analysis revealed their higher phosphorylation fold

increase by DNA-PKcs, and further only these residues are highly conserved across species (Fig. 6D,F).

We tested these antibodies by WB analysis on cortical and hippocampal extracts from WT and DNA-PKcs −/− brains. As shown in Fig. 7A, both pS308 and pT87 antibodies show immunoreactive bands at the expected molecular weight of PSD-95 that are highly decreased in DNA-PKcs −/− brain extracts. To verify the specificity of the newly generated phospho-antibodies, we treated the membranes after protein extract transfer, with alkaline phosphatase (CIP) for 2 h at 37 °C. CIP incubation causes the loss of both S308 and T87 phosphorylated bands but not those corresponding to the total PSD-95 (Fig. 7B). To further validate that synaptic DNA-PKcs phosphorylates PSD-95 in the identified residues, we performed an in vitro kinase assay using immune-purified DNA-PKcs from mouse cortex LP1 and human recombinant PSD-95. WB analysis in Fig. 7C shows the bands corresponding to DNA-PKcs in the pull-down lanes (lanes 2, 3, 4), the recombinant PSD-95 loaded with or without the kinase (lanes 3, 4, and lane 1) and the two bands specifically recognized by the phospho-specific antibodies pS308 and pT87 only when the kinase and PSD-95 are incubated with ATP (lane 4).

To evaluate whether the phosphorylation of S308 and T87 of PSD-95 is associated with LTP, we analyzed their phosphorylation changes at different time points after delivering HFS in WT mice ($n = 3$). According to the literature (Bao et al, 2004; Ifrim et al, 2015; Skibinska et al, 2001; Sun et al, 2021), we found that synaptic activity increases PSD-95 protein synthesis. In particular, PSD-95 protein levels, normalized to α-Tubulin, increased by 43 ± 7.6% and 62 ± 4.6% at 5 min and 15 min, respectively following tetanization. When using PSD-95 for normalization, LTP induction increases PSD-95 phosphorylation at both S308 (5 min: 132 ± 4%; $p < 0.05$ vs T0; 15 min: 124 ± 4%; $p < 0.05$ vs T0) and T87 (5 min: 136 ± 1.73%; $p < 0.005$ vs T0; 15 min: 119 ± 4.62%; $p < 0.05$ vs T0) (Fig. 7D). By using the housekeeping protein α-Tubulin for normalization, we found that, compared with non-tetanized control slices, LTP induction increases PSD-95 phosphorylation at both S308 and T87 that persisted for at least 15 min (5 min S308: 152 ± 7.2%; $p < 0.05$ vs T0; 15 min S308: 184 ± 6.5%; $p < 0.005$ vs T0; 5 min T87: 163.31 ± 6.83%; $p < 0.005$ vs T0; 15 min T87: 170 ± 5%; $p < 0.005$ vs T0) (Fig. 7D).

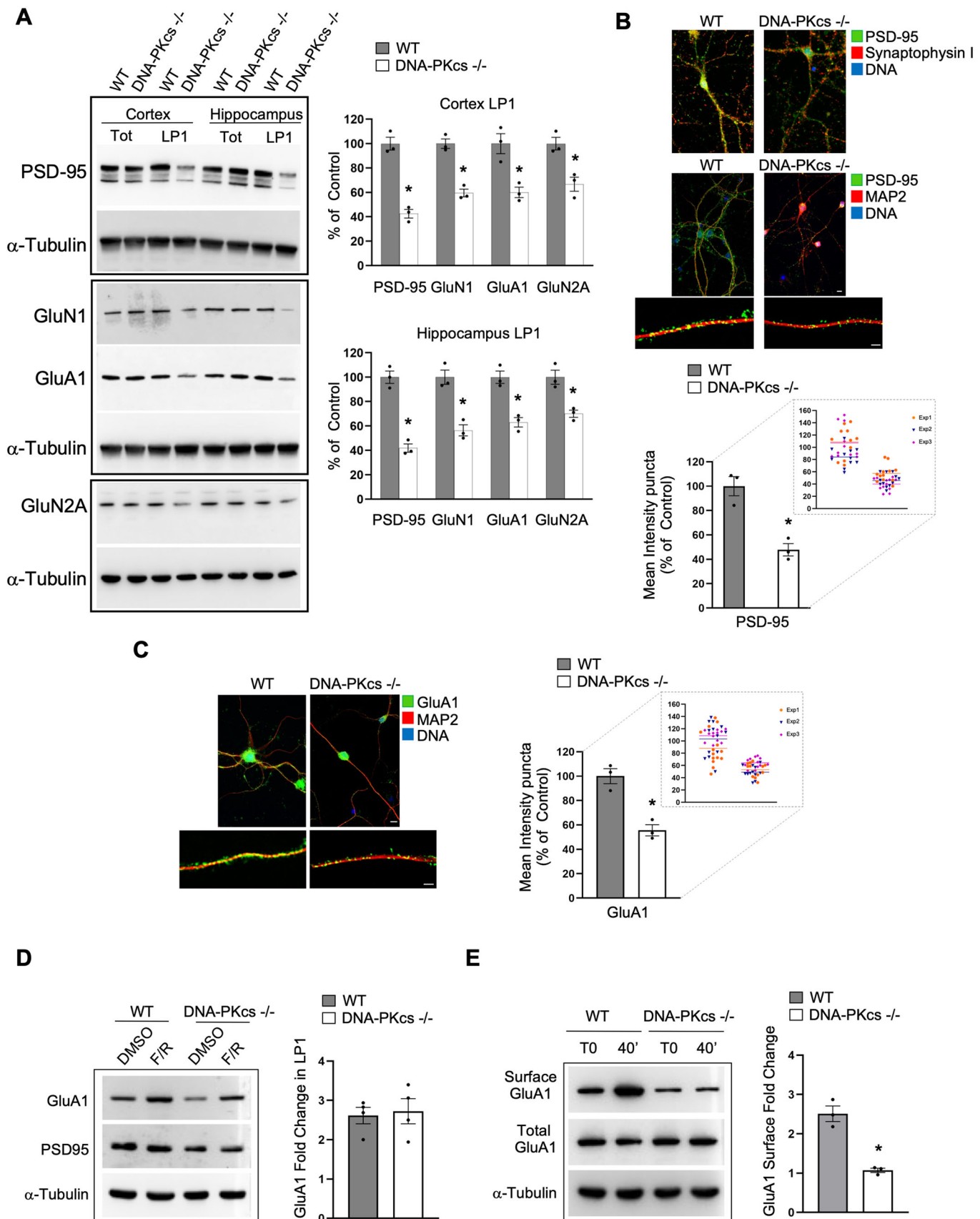

**Figure 4.  Reduced postsynaptic protein levels and delivery of GluA1 in synaptosomal membranes of DNA-PKcs −/− neurons.**

(A) Western blots of total (Tot) and synaptosomal membrane extracts (LP1) from WT and DNA-PKcs −/− cortex and hippocampus demonstrating reduction of postsynaptic proteins specifically in LP1 fractions of mutant mice. Histogram data for the cortex and hippocampus represent the normalized percent change in protein band intensity in DNA-PKcs −/− mice with respect to WT (100%). (PSD-95 in LP1 from cortex: WT: 100 ± 5.2%, DNA-PKcs −/−: 42.5 ± 3.5%; *p < 0.05; GluN1 in LP1 from cortex: WT: 100 ± 3.9%, DNA-PKcs −/−: 59.5 ± 3.2%; *p < 0.05; GluA1 in LP1 from cortex: WT: 100 ± 8.2%, DNA-PKcs −/−: 60 ± 4.2%; *p < 0.05; GluN2A in LP1 from cortex: WT: 100 ± 5.1%, DNA-PKcs −/−: 66.7 ± 5.7%; *p < 0.05); (PSD-95 in LP1 from hippocampus: WT: 100 ± 5.1%, DNA-PKcs −/−: 41.8 ± 3.4%; *p < 0.05; GluN1 in LP1 from hippocampus: WT: 100 ± 5.8%, DNA-PKcs −/−: 56.4 ± 4.5%; *p < 0.05; GluA1 in LP1 from hippocampus: WT: 100 ± 5%, DNA-PKcs −/−: 63 ± 3.9%; *p < 0.05; GluN2A in LP1 from hippocampus: WT: 100 ± 5.8%, DNA-PKcs −/−: 70 ± 2.9%; *p < 0.05). α-Tubulin was used as a loading control. Results are expressed as mean ± SEM. Statistics by Student's t-test (unpaired, two-tailed) (n = 3 mice). (B) Upper panel, fluorescence THUNDER images of cortical neurons (DIV 21) from WT and DNA-PKcs −/− mice, triple labeled with anti-PSD-95 antibody (green channel), Synaptophysin I (red) and DNA dye (blue, DAPI). Lower panel, triple fluorescence images of primary cortical neurons (DIV 21) from WT and DNA-PKcs −/− mice, labeled with anti-PSD-95 and anti-MAP2 (red channel) antibodies. Scale Bar 5 μm. High-magnification images of neurites confirm the reduction of PSD-95 staining in mutant neurons. Scale Bar 2 μm. Histogram data represent mean intensity ± SEM of fluorescence of PSD-95 labeling, plotted as a percentage of control and show a significant reduction of PSD-95 protein in mutant neurons. Statistics by Student's t-test (unpaired, two-tailed) (n = 3 independent cultures, *p < 0.05). Data distribution is shown in the enlargement; WT n = 33 neurons, DNA-PKcs −/− n = 36 neurons. (C) Immunofluorescence THUNDER images of primary cortical neurons (DIV 21) from WT and DNA-PKcs −/− mice, labeled with anti-GluA1 (green channel), anti-MAP2 (red channel) antibodies and DNA dye (blue, DAPI). High-magnification images of neurites confirm the reduction of GluA1 staining in mutant neurons. Scale Bar 2 μm. Histogram data represent mean intensity ± SEM of fluorescence of GluA1 labeling, plotted as percentage of control, and show a significant reduction of GluA1 expression in DNA-PKcs −/− neurons. Statistics by Student's t-test (unpaired, two-tailed) (n = 3 independent cultures, *p < 0.05). Scale Bar 5 μm. Data distribution is shown in the enlargement; WT n = 39 neurons, DNA-PKcs −/− n = 36 neurons. (D) Western blot analysis of synaptosomal membranes purified from WT and DNA-PKcs −/− neurons unstimulated (DMSO) or after F/R stimulation. Although synaptosomal membranes from DNA-PKcs −/− neurons have a lower expression of the GluA1 receptor subunit, after chemical stimulation, GluA1 incorporation in DNA-PKcs −/− synapses is as efficient as in WT. Band intensities, quantified and normalized by PSD-95, are expressed as fold change. α-Tubulin was used as a loading control. Error bars in histograms indicate SEM. Statistics by Student's t-test (unpaired, two-tailed) (n = 4 independent experiments, p = 0.1579). (E) Representative Western blot analysis of surface and total levels of GluA1 (assessed using cell-surface biotinylation) on hippocampal slices at T0 or 40 min after HFS delivery in WT and DNA-PKcs −/− mice. The levels of surface GluA1 remain constant up to 40 min following HFS in DNA-PKcs −/− slices (1.07 ± 0.055 folds change; p = 0.3791) as compared with WT slices that show an increased level of surface GluA1 after HFS (2.51 ± 0.2 folds change; p < 0.01). The surface-to-total ratio was calculated and expressed as mean ± SEM. Statistics by Student's t-test (unpaired, two-tailed) (n = 3 independent slices per each experimental group, *p < 0.05). Source data are available online for this figure.

These data suggest that following synaptic activity the newly synthetized PSD-95 protein undergoes phosphorylation by DNA-PKcs.

Since PSD-95 levels decrease by 57.5% and 58.2% in LP1 fraction from the cortex and hippocampus of DNA-PKcs −/− mice compared to WT, we hypothesized that DNA-PKcs might regulate PSD-95 protein amount by phosphorylation. We then performed a protein stability assay using the translation inhibitor cycloheximide (CHX). Primary cortical cultures from WT and DNA-PKcs −/− mice were treated with CHX (10 μM) to inhibit protein synthesis and PSD-95 protein content was measured over time (Fig. 7E). Quantitative WB analysis shows that PSD-95 protein content remains constant up to 48 h in cortical neurons from WT mice (n = 3), whereas it decreases in DNA-PKcs −/− mice (n = 3) starting at 9 h after CHX treatment (p < 0.05 DNA-PKcs −/− 9 h vs T0; p < 0.001 DNA-PKcs −/− 24 h vs T0; p < 0.0001 DNA-PKcs −/− 48 h vs T0; p < 0.05 DNA-PKcs −/− 9 h vs 9 h WT; p < 0.001 DNA-PKcs −/− 24 h vs 24 h WT; p < 0.0001 DNA-PKcs −/− 48 h vs 48 h WT) (Fig. 7E).

Overall, these results suggest that phosphorylation of S308 and T87 of PSD-95 might represent an important post-translational control of PSD-95 mediated by the synaptic DNA-PKcs.

## Rescue of PSD-95 protein stability, spine, and neuronal morphology in DNA-PKcs −/− neurons

To further investigate the role of S308 and T87 phosphorylation of PSD-95, two phospho-mimetic mutants (S308E and T87E), two non-phosphorylatable mutants (S308A and T87A) and PSD-95 wt were generated and expressed as Flag-tagged proteins under the control of a constitutive promoter (Fig. 8A). WB analysis in Fig. 8B shows that the Flag proteins are highly expressed in packaging cells used to produce rAAVs. Viruses over-expressing mutants were then exploited for both in vitro and in vivo experiments.

First, we infected WT cortical neurons at DIV 6 to confirm mutant expression and localization. Immunofluorescence staining, carried out at DIV 21 with an anti-Flag antibody, shows that the five constructs are abundantly expressed in neurons and localize in soma and neurites (Fig. 8C).

If phosphorylation of S308 and T87 of PSD-95 by DNA-PKcs is required for protein stability, then PSD-95 constitutively phosphorylated (PSD-95S308E and PSD-95T87E) should remain stable for a longer time in DNA-PKcs −/− cortical neurons as compared with the endogenous PSD-95. On the contrary, PSD-95 non-phosphorylatable mutants (PSD-95S308A and PSD-95T87A) should present reduced stability when over-expressed in WT neurons. Indeed, WB analysis performed on infected cortical cultures from DNA-PKcs −/− mice treated with CHX, show that PSD-95S308E and PSD-95T87E, revealed by an anti-Flag antibody, remain constant up to 48 h (Fig. 8D upper panel). In contrast, PSD-95S308A and PSD-95T87A, over-expressed in WT neurons, significantly decreases from 24 h after CHX treatment (p < 0.001 T87A 24 h vs T0; p < 0.0001 T87A 48 h vs T0; p < 0.05 S308A 24 h vs T0; p < 0.0001 S308A 48 h vs T0) (Fig. 8D lower panel; Appendix Fig. S1), similarly to the endogenous PSD-95 in DNA-PKcs −/− neurons (Fig. 7E). To exclude the possibility that PSD-95S308E and PSD-95T87E mutants were more stable in DNA-PKcs −/− cells due to less efficient degradation machinery in mutant cells, we over-expressed the constitutively phosphorylated mutants also in WT neurons. As shown in Fig. 8D (lower panel), the PSD-95S308E and PSD-95T87E mutants remain stable over time in WT neurons as efficiently as in DNA-PKcs −/− neurons. In addition, we over-expressed PSD-95 wt in DNA-PKcs −/− cortical neurons that, due to the lack of the kinase, should not be phosphorylated at S308 and T87 residues. We found that PSD-95 wt significatively decreases 24 and 48 h after CHX treatment (p < 0.05 24 h vs T0; p < 0.001 48 h vs

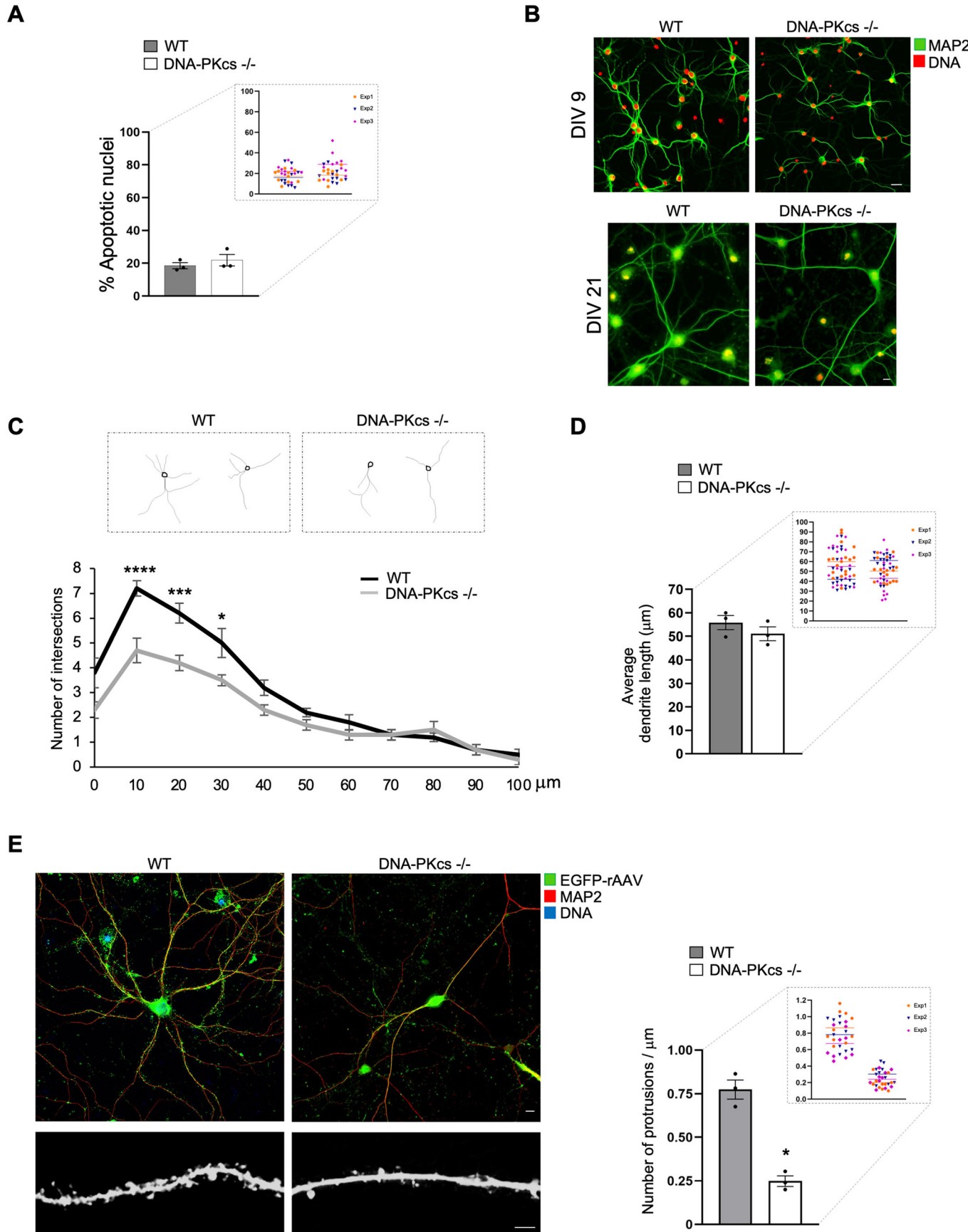

**Figure 5. The lack of DNA-PKcs affects neuronal morphology.**

(A) Histogram representing the percentage of apoptotic nuclei (expressed as pyknotic/total ratio) in WT and DNA-PKcs $-/-$ cultures, showing no differences in the two experimental groups. Results are expressed as mean ± SEM. Statistics by Student's $t$-test (unpaired, two-tailed) ($n = 3$ different cultures; $p = 0.4443$). Data distribution is shown in the enlargement; WT $n = 30$ fields, DNA-PKcs $-/- n = 30$ fields. (B) Representative confocal fluorescence images of primary cortical neurons (DIV 9 and DIV 21) from WT and DNA-PKcs $-/-$ mice labeled with anti-MAP2 (green channel) and DNA dye (propidium iodide, red channel) illustrating that, with the same number of cells, DNA-PKcs $-/-$ neurons have a less complex network of neurites as compared with WT neurons. Scale Bar 5 μm. (C) Upper panel, representative tracing images (skeletons) of neurons from WT and DNA-PKcs $-/-$ neurons. Lower panel, Sholl analysis performed on primary cortical neurons from WT and DNA-PKcs $-/-$ mice confirms a reduced neurite complexity in DNA-PKcs $-/-$ cultures as indicated by a significant reduction in the number of intersections in the range of 0–30 μm distance from the cell body. A sample of six neurons was taken for each group ($n = 3$ cultures). A number of intersections was counted in a 100-μm radius from the soma along the dendritic tree. Statistics by two-way ANOVA followed by Bonferroni post hoc analysis. Results are expressed as mean ± SEM. *$p < 0.05$; ***$p < 0.001$; ****$p < 0.0001$ DNA-PKcs $-/-$ vs WT. (D) Quantification of neurite length shows no significative difference between WT and DNA-PKcs $-/-$ neurons. $p = 0.153$ by Student's $t$-test (unpaired, two-tailed). Data distribution is shown in the enlargement; WT $n = 54$ neurons, DNA-PKcs $-/- n = 51$ neurons. (E) Immunofluorescence images acquired with a THUNDER Imager microscope of EGFP-rAAV infected neurons (green channel) from WT and DNA-PKcs $-/-$ cultures labeled with anti-MAP2 antibody (red channel) and DNA dye (DAPI, blue channel). WT neurons show a higher neurite complexity along with a higher density of more developed spines as compared with mutant cells. Scale Bar 5 μm. The black and white insets show a detail of neurites with spines. Scale Bar 2 μm. Quantification of the dendritic protrusions is expressed as number of spines per μm. DNA-PKcs $-/-$ neurons show a reduced spine number that appears less frequent along neurites as compared with WT cells. Number of spines was calculated per 50 μm dendritic length. Error bars represent SEM. A minimum of 130 spines were counted from at least ten neurons/each group, repeated for three independent experiments (*$p < 0.005$, by Student's $t$-test unpaired, two-tailed). Data distribution is shown in the enlargement; WT $n = 30$ neurons, DNA-PKcs $-/- n = 30$ neurons. Source data are available online for this figure.

T0) when over-expressed in DNA-PKcs $-/-$ neurons, whereas it remains stable in WT neurons (Fig. EV3).

Overall, these results demonstrate that the lack of DNA-PKcs-mediated phosphorylation at S308 and T87 causes PSD-95 to degrade at a faster rate, thus validating the importance of these residues for PSD-95 protein stability.

We then asked whether the higher stability showed by PSD-95S308E and T87E Flag mutants could affect the altered spine number and neuronal morphology observed in cultured neurons from DNA-PKcs $-/-$ mice. To this aim, we first performed fractionation experiments to isolate synaptosomal membranes from rAAV-infected neuronal cultures to demonstrate that S308E and T87E mutants are present in LP1 fraction (Fig. 9A). We also carried out immunofluorescence experiments showing that PSD-95S308E and T87E Flag mutants have a localization similar to the postsynaptic marker Homer 1, distributed along dendrites and in spines (Fig. 9B).

Concomitant with the increased stability of PSD-95S308E and T87E Flag mutants, we show morphological changes in mutant-infected DNA-PKcs $-/-$ cortical neurons. As shown in Fig. 9C, we noticed a significant increase in spine density by over-expressing PSD-95S308E and PSD-95T87E in DNA-PKcs $-/-$ primary cortical neurons (DIV 21) (empty vector: $0.25 \pm 0.03$; S308E: $0.6 \pm 0.035$, T87E: $0.74 \pm 0.06$; a minimum of 130 spines were counted/10 neurons for each experimental group, repeated for three independent experiments; $p < 0.005$ S308E vs empty vector; $p < 0.001$ T87E vs empty vector) (Fig. 9C). Moreover, the mutant over-expressing neurons show significant larger spine heads resembling more mature morphology as compared with spines of empty vector infected DNA-PKcs $-/-$ neurons. In addition, we analyzed spines in PSD-95 wt over-expressing DNA-PKcs $-/-$ neurons and found no significant increase in spine density (empty vector: $0.25 \pm 0.03$; PSD-95 wt $0.35 \pm 0.052$; $p = 0.8117$ PSD-95 wt vs empty vector) (Fig. 9C).

Because PSD-95 is involved in the regulation of dendrite branching, we then looked at the dendritic architecture of DNA-PKcs $-/-$ mutant-infected neurons. As shown by immunofluorescence (Fig. 10A), we demonstrate that overexpression of PSD-95T87E increases the complexity of dendrite morphology (DIV 9 and DIV 21). Indeed, Sholl analysis, using MAP2 labeling, confirmed that PSD-95T87E neurons possess a higher number of neurites as compared with both EGFP, PSD-95 wt, and PSD-95S308E DNA-PKcs $-/-$

infected neurons ($p < 0.0001$ T87E vs empty vector; $p < 0.05$ T87E vs empty vector, $n = 10$ neurons for each experimental group, repeated for three independent experiments) (Fig. 10B). Neither PSD-95S308E nor PSD-95 wt are able to reverse the decreased dendrite complexity of DNA-PKcs $-/-$ neurons as indicated by the Sholl curve and immunofluorescence (Fig. 10B,C).

## Synaptic potentiation in DNA-PKcs $-/-$ mice is improved by over-expression of PSD-95T87E mutant

Since PSD-95T87E was able to reverse both the spine number and dendritic complexity in neurons, we determined whether over-expression of PSD-95T87E mutant in vivo could improve the LTP impairment found in DNA-PKcs $-/-$ mice.

To this end, we injected PSD-95T87E, PSD-95S308E, PSD-95 wt, and EGFP rAAV expressing vectors into the cerebral lateral ventricles of DNA-PKcs $-/-$ mouse pups (P1) (Mollinari et al, 2015) (Fig. 10D). Some hippocampal slices from infected mice ($n = 3$ for each experimental group) were analyzed by quantitative WB to demonstrate similar rAAV-mediated expression of the different Flag proteins. As shown in Fig. 10E, T87E, S308E, and PSD-95 wt proteins are expressed at comparable levels in hippocampal regions at 8 weeks post-injection.

At 8 weeks post-injection, we analyzed the HFS-induced LTP in the four experimental groups, and demonstrated that synaptic potentiation in DNA-PKcs $-/- +$ T87E mice is significantly increased, as compared with DNA-PKcs $-/- +$ EGFP, S308E, and PSD-95 wt mice (EGFP $112.5 \pm 12\%$ of basal slope $n = 5$; S308E: $130.1 \pm 20.98\%$ of the basal values $n = 5$; PSD-95 wt: $137.3 \pm 10.40\%$ $n = 9$; T87E: $176.7 \pm 16.95\%$ $n = 5$ of the basal slope; $p < 0.05$ T87E vs EGFP) (Fig. 10F), thus reaching fEPSP potentiation at comparable levels of WT mice (gray background).

## Discussion

Emerging evidence demonstrates that the DNA repair kinase DNA-PKcs plays cellular roles unrelated to DNA damage repair (Cardinale et al, 2022; Chung, 2018; Ferguson et al, 2012; Kotula et al, 2013; Lucero et al, 2003; Park et al, 2017; Tian et al, 2017) even though, in

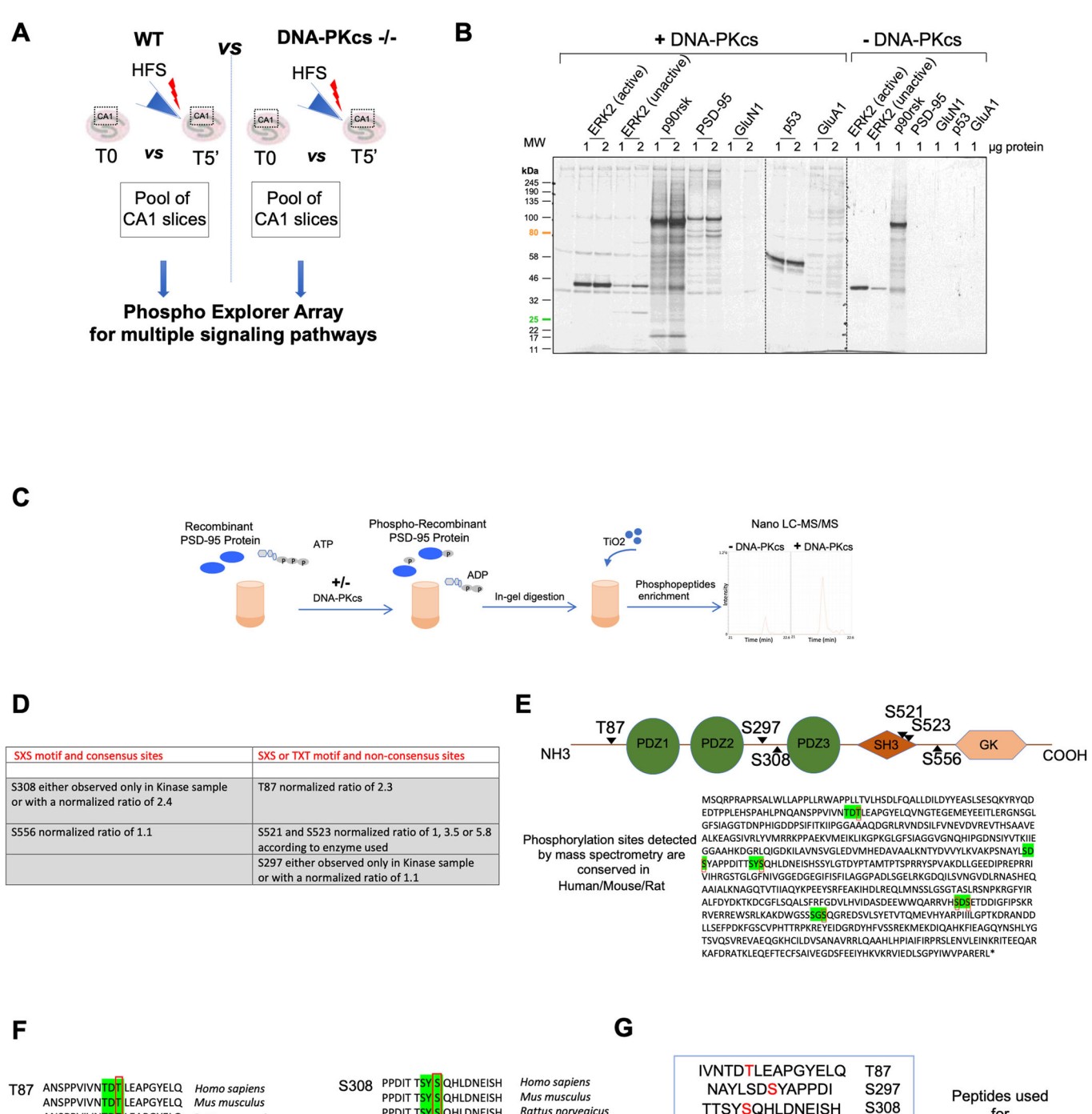

neurons, the different phenotypes caused by its deficiency have been mostly associated with the absence of NHEJ activity (Chechlacz et al, 2001; Gago-Fuentes and Oksenych, 2020; Kashiwagi et al, 2018; Matsumoto et al, 2021; Vemuri et al, 2001; Woodbine et al, 2013). However, loss of DNA-PKcs activity in humans leads to neurological severe alterations including microcephaly and seizures (Woodbine et al, 2013), thus suggesting a yet unknown role for DNA-PKcs in brain, beyond its role in DSB repair.

Here we show that DNA-PKcs has an important modulatory action in synaptic plasticity via PSD-95 phosphorylation and regulation of its protein stability. In particular, we present evidence that: (i) DNA-PKcs localizes at synapses where it exerts kinase activity and responds to synaptic stimuli; (ii) DNA-PKcs interacts with different postsynaptic proteins; (iii) DNA-PKcs phosphorylates PSD-95 at newly identified and evolutionary conserved residues and controls PSD-95 protein stability;

◄ **Figure 6. Identification of novel residues in PSD-95 phosphorylated by DNA-PKcs.**

(A) The scheme shows the approach used to perform the phospho explorer antibody array. Unstimulated (T0) and tetanized (T5′) CA1 dissected slices ($n = 6$ slices from three different mice) were pooled and protein extracts used to analyze phosphorylation changes after delivery of HFS in slices from WT and DNA-PKcs −/− mice. (B) Selected human purified recombinant proteins were used as a substrate for the in vitro DNA-PKcs kinase assay in presence or absence of the purified DNA-PKcs protein and specific incorporation of radioactive phosphate was detected by autoradiography. For each recombinant protein, two concentrations were loaded, double each other (20/40 μg/ml) (lanes 1 and 2). PSD-95 and p53 evoked specific radioactive incorporation since the corresponding band disappeared when the assay was performed in the absence of DNA-PKcs. (C) Scheme representing the in vitro kinase reaction using recombinant PSD-95 and DNA-PKcs proteins. After reaction and enzymatic digestion, phospho-peptides were enriched with TiO2 beads and analyzed by Nanoscale liquid chromatography coupled to tandem mass spectrometry. Extracted ion chromatogram of m/z 1016.8078Th (Threonine 87) in samples without kinase or with DNA-PKcs, enzymatically digested by GluC is shown on the right. (D) The table shows the newly identified residues in PSD-95 protein phosphorylated by DNA-PKcs and divided into a group that presents the DNA-PKcs kinase recognition motif SQ (two residues) and another group that does not (four residues). Phosphorylation fold increase by DNA-PKcs is also indicated in the table. (E) Upper panel, drawing of PSD-95 protein represented as modular protein containing different domains. The position of the six identified residues is indicated. Lower panel, protein sequence of human PSD-95 protein. The six identified residues phosphorylated by DNA-PKcs are highly conserved in mammals. The phosphorylated residues are boxed in red. The motifs SXS or TXT are highlighted in green. (F) Multiple alignments of short segments of PSD-95 protein show that T87 and S308, in particular, are conserved across species. (G) The peptides of human PSD-95 containing the phosphorylated residues were used to generate phospo-specific antibodies. The numbers in the peptides represent the amino acid position. Source data are available online for this figure.

(iv) DNA-PKcs −/− mice show deficits in LTP associated with changes in neuronal morphology, and reduced levels of post-synaptic proteins; (v) over-expression of PSD-95 protein constitutively phosphorylated in DNA-PKcs −/− mice improves LTP deficits by restoring PSD-95 protein levels.

Prolonged changes in synaptic strength, such as those occurring in LTP are thought to contribute to learning and memory processes through multiple mechanisms, many of which involve protein phosphorylation (Soderling and Derkach, 2000). Here we show that DNA-PKcs has a relevant role in the induction of LTP. Accordingly, we find that DNA-PKcs is associated with the synaptosomal compartment, where it translocates following the induction of a synaptic stimulus and exerts kinase activity. Curiously, DNA-PKcs was previously reported to localize in lipid rafts, confirming an interaction with resident membrane proteins (Lucero et al, 2003). Indeed, by co-immunoprecipitation experiments, we show that the kinase selectively interacts with PSD-95, GluN1, and GluN2A/B, but not GluA1. Further evidence will corroborate whether the interaction of DNA-PKcs with NMDAR subunits is directly established or mediated by PSD-95/NMDARs association (Niethammer et al, 1996). It is interesting to note that AMPARs comprising GluA1 are anchored at the synapse by PSD-95 via interaction with the auxiliary protein Stargazing (Bats et al, 2007; Chamberlin et al, 1998; Schnell et al, 2002), and this may account for the absence of association of GluA1 with DNA-PKcs.

PSD-95 is highly abundant in the postsynaptic density of excitatory synapses and is known to play a role in different forms of synaptic transmission (Bats et al, 2007; Cane et al, 2014; Ehrlich et al, 2007; El-Husseini et al, 2000; Schluter et al, 2006; Steiner et al, 2008) as well as in the formation and long-term stabilization of memory (Caly et al, 2019; Ehrlich and Malinow, 2004; Stein et al, 2003). PSD-95 modulates synaptic activity through its highly regulated phosphorylation state (Gardoni et al, 2006; Vallejo et al, 2017). By in vitro kinase assay and Nano LC-MS/MS analysis, we have demonstrated that DNA-PKcs behaves as a novel kinase phosphorylating PSD-95 at 6 unexpected residues localized in different regions of the protein and comprising S308 and T87.

Furthermore, LTP causes a strong, acutely regulated increase of PSD-95 protein synthesis and phosphorylation, particularly at residues S308 and T87, thus suggesting that the newly synthetized PSD-95 protein is phosphorylated by DNA-PKcs to guarantee its stability at synapses during synaptic activity. Noteworthy,

phosphorylation increment at S308 and T87 after tetanization bears a resemblance to the trend of activation of the major molecules playing a role in LTP, such as CaMKII and ERK1/2 (Racaniello et al, 2010).

Of relevance in this context, lack of DNA-PKcs activity in mice produces specific deficits in LTP in the Schaffer collateral-CA1 pathway, for at least the first 60 min post-tetanus, in addition to basal synaptic transmission impairment, whereas no difference is observed in PPF, a form of short-term presynaptic plasticity, suggesting that DNA-PKcs activity is limited to postsynaptic terminals.

DNA-PKcs −/− mice show reduced levels of several post-synaptic proteins, including PSD-95, a protein involved in the trafficking and stabilization of numerous postsynaptic molecules, including NMDA and AMPA receptors (De Roo et al, 2008; El-Husseini et al, 2000; Ehrlich et al, 2007). In parallel with the reduced levels of PSD-95, we also find a reduction of GluN1, GluN2A, and GluA1 protein receptor levels, specifically in the synaptic membrane compartment LP1. Thus, the impaired LTP demonstrated in DNA-PKcs −/− mice may depend on the reduced levels of PSD-95 anchoring protein, in turn affecting protein organization at the synapses, as confirmed by the reduced surface expression of additional GluA1 receptors during LTP.

The mechanisms required for LTP include signaling by different protein kinases such as CaMKII, Akt and ERK1/2 (Nicoll, 2017; Racaniello et al, 2010). Importantly, we do not find any difference in the activation of these kinases in DNA-PKcs −/− mice. These findings support the hypothesis that DNA-PKcs activity by itself is required for LTP early expression and acts in parallel with canonical pathways. In accordance with this hypothesis, our unpublished data (Cardinale et al, 2006) carried out using a SCID mouse model, having only a 8 kDa COOH terminus truncation of DNA-PKcs protein with no enzymatic activity, resulted in similar electrophysiological defects, thus confirming the role of kinase activity of DNA-PKcs in synaptic plasticity.

PSD-95 affects synapse maturation, stabilization, and number (De Roo et al, 2008; El-Husseini et al, 2000; Ehrlich et al, 2007). Knockdown of PSD-95 reduces the development of synaptic structures (Ehrlich et al, 2007), and PSD-95 mutant mice have alterations in dendritic spine density in several brain regions (Vickers et al, 2006). In line with this evidence, cortical neurons from DNA-PKcs −/− mice display a significant decrease in

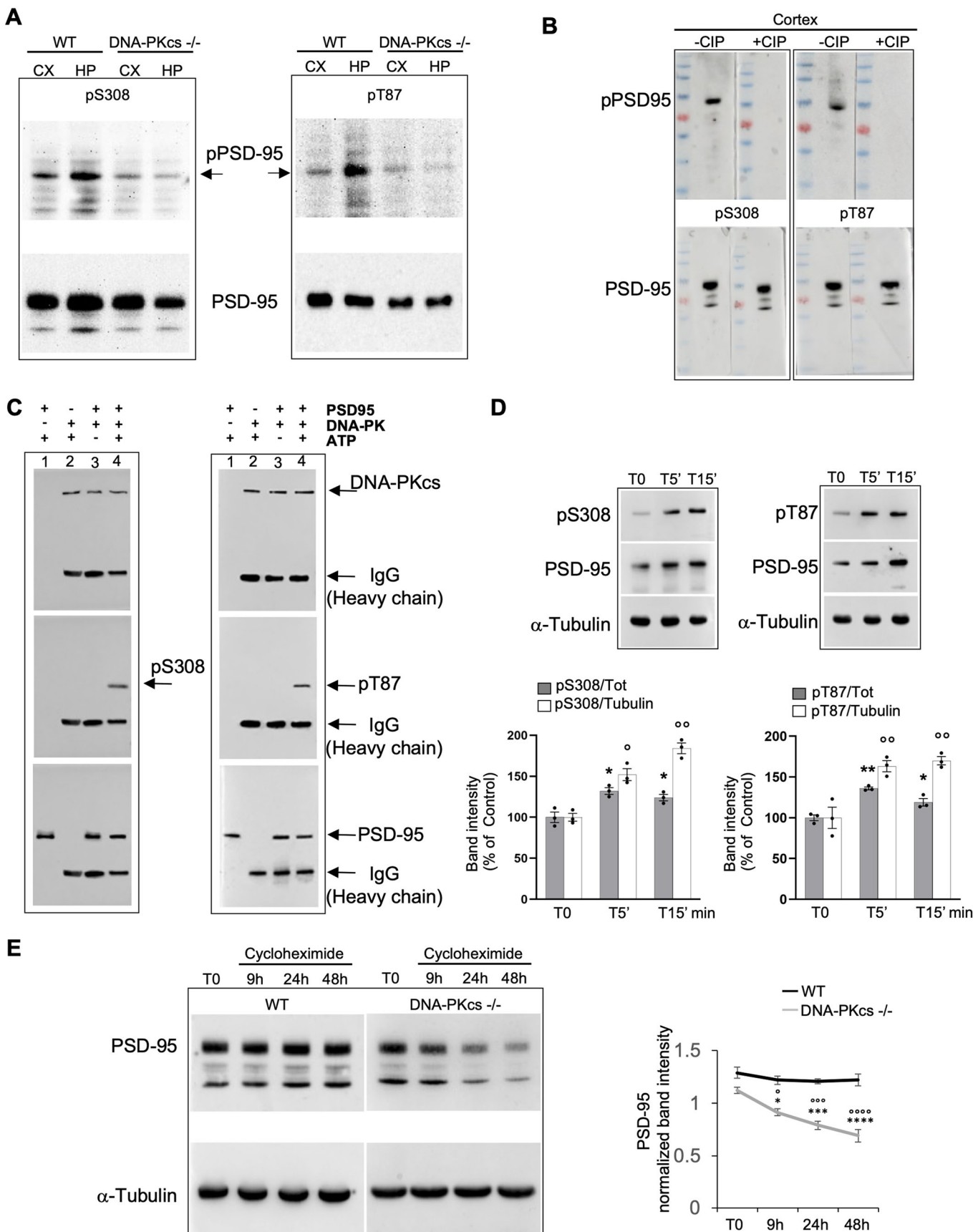

**Figure 7. Effect of PSD-95 phosphorylation at S308 and T87 on protein stability.**

(A) Representative Western Blots of protein extracts from the cortex ($n = 3$) and hippocampus ($n = 3$) of WT or DNA-PKcs −/− mice probed with the anti-phospho-S308 (pS308) and anti-phospho-T87 (pT87). A single band at the expected molecular weight is recognized by the rabbit phospho-antibodies in WT brain extracts. No bands are detected in brain regions of mice lacking DNA-PKcs activity. PSD-95 was used as a loading control. (B) Nitrocellulose membranes were treated with or without calf intestinal alkaline phosphatase (CIP) and probed with the anti-pT87 and anti-pS308 antibodies. Antibodies specificity was confirmed by the disappearance of both S308 and T87 phosphorylated bands (+CIP lanes) but not of those corresponding to the total PSD-95. (C) In vitro kinase assay using immune-purified DNA-PKcs from mouse cortical LP1 extracts and recombinant PSD-95 protein. PSD-95 phosphorylation was analyzed by Western blot using the phospho-specific antibodies pS308 and pT87. The kinase assay ($n = 3$ independent experiments) was performed incubating: recombinant PSD-95 with ATP in the absence of DNA-PKcs (lane 1); DNA-PKcs with ATP without PSD-95 (lane 2); the two proteins without or with ATP (lane 3 and 4). IgG heavy chain bands in lanes 2, 3, and 4 indicate DNA-PKcs immunoprecipitation. The positions of the purified DNA-PKcs, pPSD-95 (S308 and T87), total PSD-95, and IgG are indicated by arrows. (D) Representative Western blots of phosphorylated PSD-95 at S308 and at T87 showing the increase of PSD-95 protein synthesis and phosphorylation at both residues at different time points after delivery of HFS to hippocampal slices. Densitometric quantifications of the immunoreactive bands, normalized to α-Tubulin or total PSD-95, are represented as percent changes in protein phosphorylation with respect to control (T0) for each time point (means ± SEM; $n = 3$ independent experiments). $*p < 0.05$ vs control values (T0), $**p < 0.005$ vs control values (T0). $°p < 0.05$ vs control values (T0), $°°p < 0.005$ vs control values (T0). Statistics by two-way ANOVA followed by Tukey's post hoc analysis. (E) Representative Western Blots of protein extracts of cortical neurons from WT and DNA-PKcs −/− mice showing that PSD-95 protein remains constant, after cycloheximide treatment, up to 48 h in WT cortical neurons, whereas it decreases over time in DNA-PKcs −/− neurons. Values in the plot represent the quantification of PSD-95 protein levels over time following cycloheximide treatment normalized to α-Tubulin. (means ± SEM; $n = 3$ independent experiments). Statistics by two-way ANOVA followed by Tukey's post hoc analysis. $*p < 0.05$ DNA-PKcs −/− 9 h vs DNA-PKcs −/− T0; $***p < 0.001$ DNA-PKcs −/− 24 h vs DNA-PKcs −/− T0, $****p < 0.0001$ DNA-PKcs −/− 48 h vs DNA-PKcs −/− T0, $°p < 0.05$ DNA-PKcs −/− 9 h vs WT 9 h, $°°°p < 0.001$ DNA-PKcs −/− 24 h vs WT 24 h, $°°°°p < 0.0001$ DNA-PKcs −/− 48 h vs WT 48 h. Source data are available online for this figure.

dendritic complexity and spine density associated with immature spine morphology.

PSD-95 localization and function are regulated through various post-translational modifications including phosphorylation, palmitoylation, and ubiquitination (Vallejo et al, 2017). In particular, in response to NMDA receptor activation, PSD-95 is ubiquitinated and rapidly removed from synaptic sites by proteasome-dependent degradation, which is mediated by the ubiquitin E3 ligase murine double minute 2 (Mdm2) (Colledge et al, 2003). Thus, the ubiquitination of PSD-95 through the Mdm2-mediated pathway is critical in regulating AMPA receptor surface expression and plays a fundamental role in synaptic strength and plasticity in the mammalian brain. We have found that phosphorylation at S308 and T87 of PSD-95 by DNA-PKcs is required for PSD-95 protein stability. Interestingly, these newly identified sites phosphorylated by DNA-PKcs have a conserved short motif of two serines or threonines separated by one amino acid (Ser-X-Ser (SXS); Thr-X-Thr (TXT)) and are highly conserved in vertebrate animals. Notably, the phosphorylation of the C-terminal SXS motif of receptor-regulated SMADs (R-SMADs) is indispensable for cellular responses to TGF-beta receptors (Shi and Massague, 2003). After C-terminal phosphorylation and translocation to the nucleus, R-SMADs become ubiquitinated and degraded by proteasomes (Moustakas et al, 2001). Moreover, we have noticed that the phosphorylation at S301 of the transcriptional factor Egr1 by DNA-PKcs (Waldrip et al, 2021), is within the conserved motif SXS and this phosphorylation is required for prevention of Egr1 proteasomal degradation. Based on these results, it is reasonable to speculate that the activity-dependent phosphorylation of PSD-95 by DNA-PKcs may exert the control of PSD-95 protein stability probably preventing ubiquitination/degradation of PSD-95, thus influencing the dynamic changes in receptor level at the synapse. Further studies are needed to verify whether phosphorylation of PSD-95 mediated by the synaptic DNA-PKcs is required to avoid its proteasomal degradation.

Moreover, we have found that the PSD-95 phosphomimetics over-expressed in DNA-PKcs −/− cortical neurons, remain stable up to 48 h following CHX treatment. Conversely, non-phosphorylatable mutants over-expressed in WT neurons,

significantly decrease 24 h after CHX treatment, similar to the endogenous PSD-95 in DNA-PKcs −/− neurons. The need for the phosphorylation of PSD-95 at S308 and T87 for its stability is further confirmed by the fact that PSD-95 wt protein does not remain stable when over-expressed in DNA-PKcs −/− neurons.

The augmented stability of PSD-95S308E and T87E mutants in DNA-PKcs −/− neurons is associated with a significant increase in spine density whereas PSD-95 wt, due to the lack of DNA-PKcs-mediated phosphorylation, is not able to restore the phenotype of mutant neurons. However, only PSD-95T87E can reverse the decrease in dendrite complexity. Significantly, over-expression of PSD-95T87E mutant in vivo can improve the LTP impairment found in DNA-PKcs −/− mice.

Taken together, our data support a model in which phosphorylation of PSD-95 at T87 specifically is required for structural and functional plasticity underlying LTP. This result is not astonishing considering that PSD-95, likely through its multiple protein–protein interaction motifs, regulates many distinct and separable aspects of synapse structure, function, and plasticity. The enhanced PSD-95 stability of the two constitutively phosphorylated mutants could not be the only factor determining dendritic complexity or LTP in neuronal cultures or in rAAV injected DNA-PKcs −/− mice. Indeed, it is known that PSD-95 shapes dendrite architecture in cultured neurons by altering microtubule dynamics via interaction with different cytoskeleton proteins (Brenman et al, 1998; Charych et al, 2006; Sweet et al, 2011). Moreover, the spine formation/maturation and dendritic arbor development are not necessarily concurrent phenomena (Bustos et al, 2014). Therefore, it is plausible that the different mutations mimicking PSD-95 phosphorylation, by causing conformational changes, have independent effects on spine formation/maturation, dendritic complexity, and LTP due to different regulation of PSD-95 interactions with other partners. In particular, structural analysis could achieve a detailed understanding of the role of these phosphorylations not only in PSD-95 protein stability, but also in the regulation of PSD-95 interactions with other partners. Thus, further studies are needed to unveil the role of PSD-95 phosphorylation at S308 residue.

Other DNA repair kinases, such as ATR and ATM are supposed to elicit alternative roles in post-mitotic neurons. Indeed, both

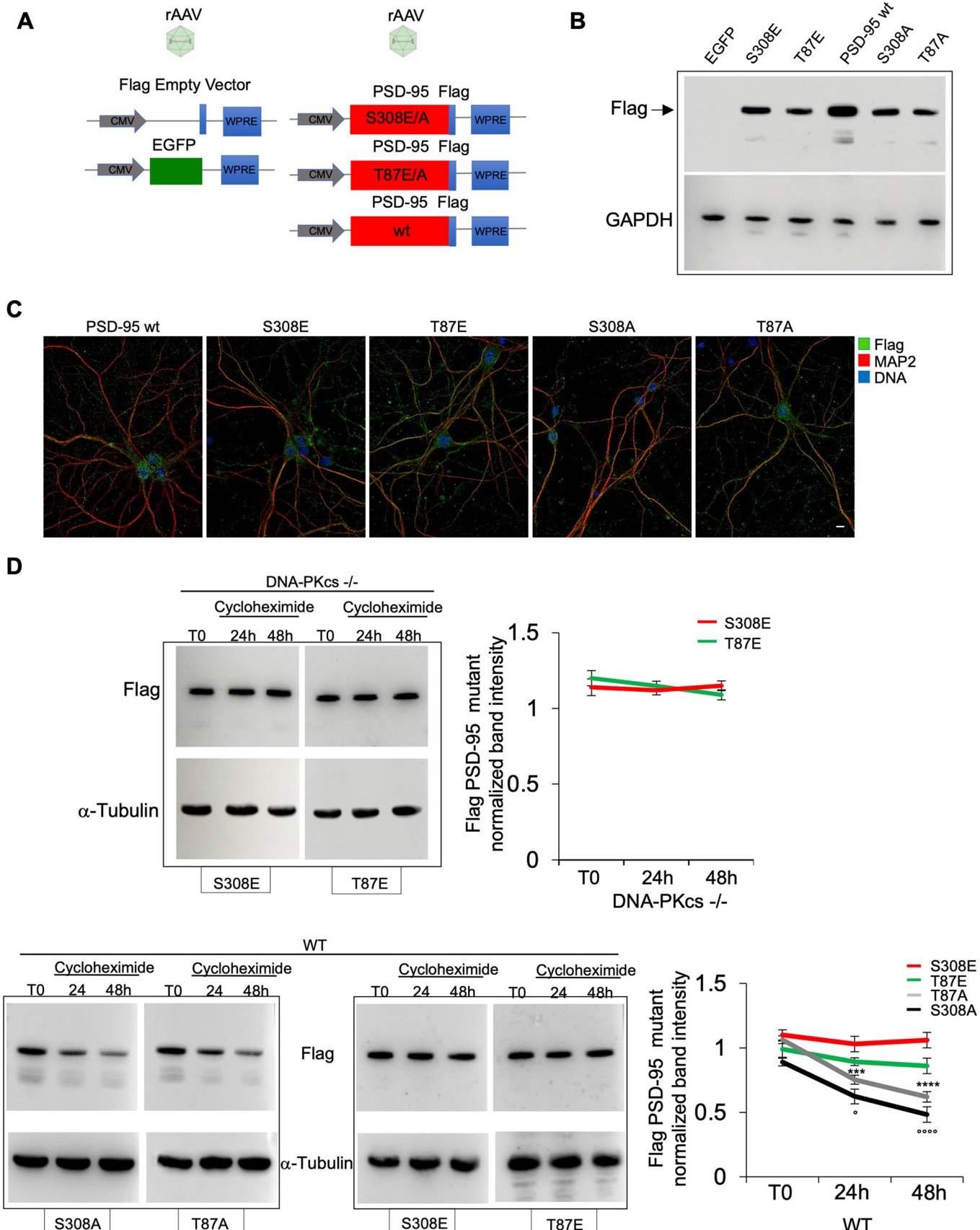

◀ **Figure 8. Phospho-mimetic mutants PSD-95S308E and PSD-95T87E remain stable in DNA-PKcs −/− neurons.**

(A) Two phospho-mimetic, two non-phosphorylatable mutants, and PSD-95 wt Flag-tagged proteins were generated and inserted in the backbone of a rAAV vector, under constitutive promoter CMV, for virion production. Control viral vectors were a Flag empty vector and an EGFP expressing vector both under CMV promoter. (B) Representative Western Blot of protein extracts from primary cortical neurons ($n = 3$) infected with rAAV viruses to confirm mutant protein expression when probed with an anti-Flag antibody. A rAAV vector over-expressing EGFP protein was used as a control. GAPDH was used as a loading control. (C) High-magnification images of cortical neurons (DIV 21) transduced with rAAV to express PSD-95 wt, the phospho-mimetic, and the non-phosphorylatable PSD-95 mutants show a strong over-expression in culture. Neurons were labeled with anti-Flag (green channel), anti-MAP2 (red channel) antibodies, and DNA dye (blue, DAPI). Scale Bar 5 µm. (D) Upper panel, representative Western Blots of protein extracts from DNA-PKcs −/− primary cortical neurons transduced with PSD-95S308E and PSD-95T87E phospho-mimetic mutants. Membranes probed with anti-Flag antibody revealed that the mutant proteins remain stable in DNA-PKcs −/− neurons up to 48 h after cycloheximide treatment. Values in the plot represent quantification of PSD-95S308E and PSD-95T87E mutant protein levels over time following cycloheximide treatment and normalized to α-Tubulin (means ± SEM; $n = 3$ independent experiments). Statistics by two-way ANOVA followed by Bonferroni post hoc analysis. PSD-95T87E 24 h vs T0 $p > 0.999$; PSD-95T87E 48 h vs T0 $p > 0.999$; PSD-95S308E 24 h vs T0 $p > 0.999$; PSD-95S308E 48 h vs T0 $p > 0.999$. Lower panel, representative Western blots of protein extracts from WT cortical neurons transduced with PSD-95S308A and PSD-95T87A non-phosphorylatable mutants (Left panel) or PSD-95S308E and PSD-95T87E (Right panel). Membranes probed with anti-Flag antibody revealed that non-phosphorylatable mutant proteins significantly decrease from 24 h after cycloheximide in WT neurons while the PSD-95S308E and PSD-95T87E proteins remain stable up to 48 h. Values in the plot represent the quantification of PSD-95S308A, PSD-95T87A, PSD-95S308E, and PSD-95T87E mutant protein levels over time following cycloheximide treatment and normalized to α-Tubulin. (means ± SEM; $n = 3$ independent experiments). Statistics by two-way ANOVA followed by Tukey's post hoc analysis. ***$p < 0.001$ PSD-95T87A 24 h vs T0; ****$p < 0.0001$ PSD-95T87A 48 h vs T0; °$p < 0.05$ PSD-95S308A 24 h vs T0; °°°°$p < 0.0001$ PSD-95S308A 48 h vs T0. CMV cytomegalovirus, EGFP enhanced green fluorescent protein, WPRE Woodchuck hepatitis virus post-transcriptional regulatory element. Source data are available online for this figure.

proteins are found in the neuronal cytoplasm and synaptic vesicles. In the cytoplasm, ATM forms a complex with two synaptic vesicle proteins, VAMP2 and synapsin I, and in ATM- or ATR-deficient neurons, spontaneous vesicle release is reduced (Li et al, 2009). Moreover, ATR and ATM may contribute to maintaining the excitatory/inhibitory balance of synaptic inputs by regulating the dynamics of different populations of synaptic vesicles (Cheng et al, 2018). Interestingly, ATM-deficient mice exhibit a reduction of both LTP and PPF, suggesting its involvement in synaptic plasticity in adult brain (Vail et al, 2016), whereas ATR deletion in mice compromises presynaptic functionality and neurotransmitter release, thereby elevating neuronal excitability and leading to increased epileptiform activity (Kirtay et al, 2021).

In humans, mutations in ATM result in ataxia-telangiectasia (A-T), a multisystem disease that includes a prominent neurodegenerative phenotype mostly affecting the cerebellum (McKinnon, 2004). A partial deficiency of ATR leads to devastating neurological consequences that include microcephaly and mental retardation (O'Driscoll et al, 2003), a disorder known as Seckel syndrome (Ogi et al, 2012).

We have previously demonstrated that beta-amyloid, a potential proximate effector of neurotoxicity in Alzheimer's disease (AD), inhibits DNA-PKcs kinase activity (Cardinale et al, 2012). Accordingly, NHEJ activity and protein levels of DNA-PKcs are significantly lower in AD brains compared with control subjects (Shackelford, 2006). Interestingly, reduced expression of PSD-95 has been observed in brain tissue from AD patients (Gylys et al, 2004; Savioz et al, 2014) and in mouse models of AD (Shao et al, 2011). Part of the mechanism responsible for this decrease involves the ubiquitin-proteasomal degradation system (Colledge et al, 2003). In addition, it is known that beta-Amyloid co-localizes with PSD-95 specifically at excitatory synapses in human post-mortem AD brains as well as in cultured murine neurons exposed to beta-Amyloid oligomers (Lacor et al, 2004) where it might induce the degradation of PSD-95 specifically by proteasomes (Roselli et al, 2005). It is also worth mentioning that PSD-95 protects synapses from beta-amyloid toxicity, suggesting that low levels of synaptic PSD-95 may determine synapse vulnerability to beta-amyloid in AD (Dore et al, 2021).

Consistent with these studies and in light of our findings, it is therefore plausible to hypothesize that, at least in AD, the reduced levels of DNA-PKcs activity, mediated by beta-amyloid, can

determine the reduction of PSD-95 through the lack of its phosphorylation and the consequent proteasomal degradation.

Mutations in PSD-95 have been identified as associated with diseases such as stroke, intellectual disability, autism spectrum disorder (Feyder et al, 2010), schizophrenia (Coley and Gao, 2019; Volk et al, 2015). Moreover, it was observed that pluripotent stem cells from peripheral fibroblasts of schizophrenia patients show decreased PSD-95 protein amounts and reduced neurite number when differentiated into neurons (Brennand et al, 2011).

Since the loss of PSD-95 at synapses affects synaptic function and strength, we believe that molecules and signaling pathways, including post-translational modifications, which regulate the integrity of PSD-95, may have therapeutic potential for reducing synaptic loss and cognitive impairment in a wide range of diseases.

In conclusion, our findings reveal a new and specific role of DNA-PKcs at the synapses and prompt further investigation of DNA-PKcs activity in various disorders characterized by neurological deficits.

# Methods

## Mice

We used 2/3 months old male DNA-PK catalytic subunit-null (DNA-PKcs −/−) mice and their wild-type littermates (CD1, WT), kindly provided by ENEA, Centro Ricerche Casaccia, 00123 – Roma- ex DGSAF 7586 (Approval number DM 365/2015-PR to S. Pazzaglia). WT mice were housed under conventional conditions, while *DNA-PKcs −/−* mice were housed in sterilized filter-topped cages kept in laminar flow isolators. All mice were maintained with food and water *ad libitum* and in 12-h light/dark cycle. *DNA-PKcs −/−* mice were kept in a CD1 background. Homozygous mice were obtained by heterozygous crosses and the desired genotypes analyzed by PCR on genomic tails as previously described (Tanori et al, 2019; Tanori et al, 2011). Animal procedures were carried out according to the European Community Guidelines for Animal Care, DL 26/2014, application of the European Communities Council Directive, 2010/63/EU, FELASA and ARRIVE guidelines, and approved by the Italian Ministry of Health and by the local Institutional Animal Care and Use Committee (IACUC). Animal studies were approved and

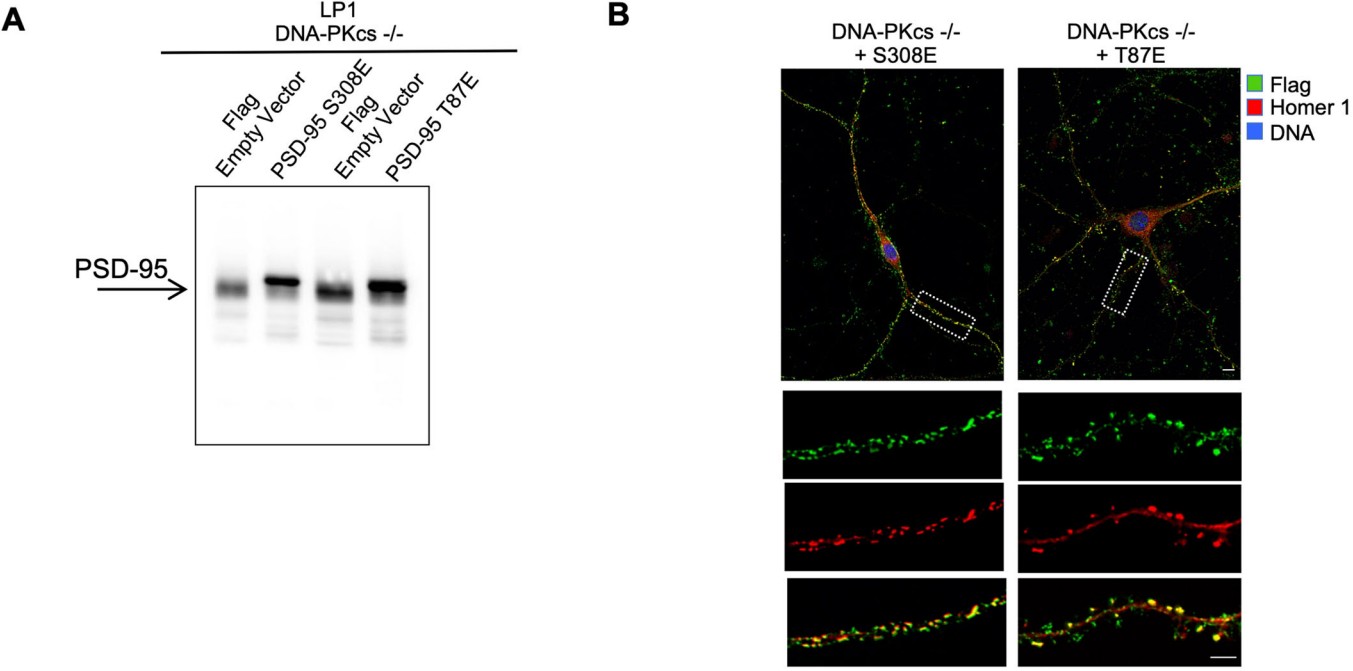

◀ **Figure 9. Phospho-mimetic mutants localize at synaptic membranes and restore spine number in DNA-PKcs −/− neurons.**

(A) Representative Western Blot of LP1 protein extracts from DNA-PKcs −/− primary cortical neurons (n = 3 cultures) after transduction with rAAV empty vector or rAAV PSD-95 phospho-mimetic mutants, probed with the anti-PSD-95 antibody. Both mutants show a localization in synaptosomal membranes after over-expression. (B) Upper panels, triple immunofluorescence THUNDER images of rAAV-infected primary cortical neurons (DIV 21) labeled with the anti-Homer 1 (red channel) and anti-Flag (green channel) antibodies showing a strong synaptic expression of the phospho-mimetic mutants. Scale Bar 5 μm. Lower panels, high magnification views of dendrites, marked by white boxes in upper panels, showing the distribution of PSD-95 mutants in spines and their co-localization with the postsynaptic protein Homer 1. Single-channel images are provided to better evaluate the localization of each protein. Scale Bar, 2 μm. (C) Representative THUNDER microscope immunofluorescence images of cortical neurons (DIV 21) from DNA-PKcs −/− mice infected with the empty vector or the PSD-95 wt, or S308E or T87E proteins, were further infected with EGFP-rAAV virus to visualize spine morphology, and then stained with the anti-MAP2 antibody (red channel) and DNA dye (DAPI, blue channel). Scale Bar 5 μm. The over-expression of both phospho-mimetic mutants in DNA-PKcs −/− cortical neurons is able to increase the number of spines along dendrites with significantly larger heads, as shown in the high-magnification images (black and white), as compared with both PSD-95 wt and the empty vector. Number of spines was calculated per 50 μm dendritic length. Histogram data represent the number of protrusions per μm. Error bars represent SEM. A minimum of 130 spines were counted from at least 10 neurons/each group, repeated for three independent experiments (**$p < 0.005$ DNA-PKcs −/− S308E vs empty vector; ***$p < 0.001$ DNA-PKcs −/− T87E vs empty vector; *$p < 0.05$ DNA-PKcs −/− S308E vs PSD-95 wt; **$p < 0.005$ DNA-PKcs −/− T87E vs PSD-95 wt). Statistics by two-way ANOVA followed by Bonferroni post hoc analysis. Data distribution is shown in the enlargement; DNA-PKcs −/− Empty vector n = 30 neurons, DNA-PKcs −/− PSD-95 wt n = 30 neurons, DNA-PKcs −/− S308E n = 30 neurons, DNA-PKcs −/− T87E n = 30 neurons. Source data are available online for this figure.

permission was issued by "Ministero della Salute" (Approval number DM 90/2016-PR to D. Merlo).

## Genotyping

Genomic DNA was prepared from tail tip by standard methods, and genotyping was performed by PCR amplification as previously described (Tanori et al, 2019; Tanori et al, 2011).

## Human tissues

Human brain tissues were obtained from the Medical Research Council (MRC) London Neurodegenerative Diseases Brain Bank hosted at the Institute of Psychiatry, Psychology and Neuroscience, KCL. All cases were collected under informed consent, and the bank operates under a license from the Human Tissue Authority, and ethical approval as a research tissue bank (08/MRE09/38 + 5). All methods were carried out in accordance with relevant guidelines and regulations, and the study was approved by the Institutional Review Board of the University of Rome "Tor Vergata" (Protocol N°98.18).

## Human cells

Human neural progenitor cells (Thermo Fisher Scientific), a kind gift from A.M. Rinaldi, were maintained in proliferative state as previously described (Mollinari et al, 2009). HEK293 cells were grown in culture in DMEM high glucose medium containing 10% fetal bovine serum and antibiotics.

## Primary neuronal cultures and treatments

For embryonic cortical neurons, female wild-type and DNA-PKcs −/− mice were euthanized by $CO_2$ and cervical dislocation. Cells were prepared according to (De Chiara et al, 2016). Cortical (E15) or hippocampal (E18) tissues of mouse embryos were dissected and dissociated by trypsin treatment followed by trituration. After removal of trypsin, neurons were seeded onto Petri dishes with or without coverslips coated with poly-L-lysine, and plated at the density of 600,000 cells in 35-mm for imaging experiments, or at a density of 800,000 cells in a 35-mm plate for biochemical analysis or at $5 \times 10^6$ cells in 100-mm diameter plates for LP1 purification and biochemical analysis. The cultures were maintained in Neurobasal supplemented

with B27 (Invitrogen), 1 mM sodium pyruvate, 2 mM glutamine, and antibiotics in a humidified incubator at 37 °C and 5% $CO_2$. At DIV 5, 50% of freshly made complete medium was added without arabinofuranosyl cytidine (AraC). For chemical LTP, neuronal cultures (DIV 9) were first incubated in ACSF (125 mM NaCl, 2.5 mM KCl, 1 mM $MgCl_2$, 2 mM $CaCl_2$, 33 mM D-glucose, and 25 mM HEPES pH 7.5) for 30 min at RT, followed by stimulation with 50 μM forskolin (Sigma) and 0.1 μM rolipram (Calbiochem, San Diego, CA, USA) in ACSF (no $MgCl_2$). After 10 min of stimulation, neurons were washed in ACSF, collected by pipetting, and then subjected to subcellular fractionation.

Cycloheximide assay was performed by adding 10 μM cycloheximide (Sigma-Aldrich) at primary cortical neurons (DIV 9) for the indicated timepoints (9, 24, and 48 h).

## Subcellular fractionation of brain tissues and primary cortical neurons

Brain tissues and mice neuronal cultures were subjected to fractionation by differential centrifugation as previously described (Cardinale et al, 2015; Dunah and Standaert, 2001) with minor modifications.

## Protein extraction

P1, P3, LP1, and LP2 fractions, pieces of brain tissues and HEK293 cells were lysed for 30 min in ice-cold RIPA Buffer 0.1% SDS (50 mM Tris-HCl pH 7.5, 150 mM NaCl, 1% Triton X-100, 0.1% SDS, and 0.5% sodium deoxycholate) containing complete protease and phosphatase inhibitor cocktails (Sigma-Aldrich). For the extraction of protein derived from primary neuronal culture fractions (P3, LP1, and LP2) and from hippocampal slices used for electrophysiology, was applied ice-cold RIPA Buffer 1% SDS (50 mM Tris-HCl pH 7.5, 150 mM NaCl, 1% Triton X-100, 1% SDS, and 0.5% sodium deoxycholate) for 30 min on ice. For extraction of DNA-PKcs from the P1 fraction, 5 freeze and thaw cycles were used in whole cell extract high salt (WCE$_{HS}$) (20 mM HEPES pH 7,6, 450 mM NaCl, 0.2 mM EDTA, 25% glycerol, and 0.5 mM DTT) containing complete protease and phosphatase inhibitor cocktails (Sigma-Aldrich). Protein concentration was determined using the bicinchoninic acid kit (Micro BCA, Pierce, Rockford, IL, USA) and the appropriate amount of protein was subjected to SDS-PAGE.

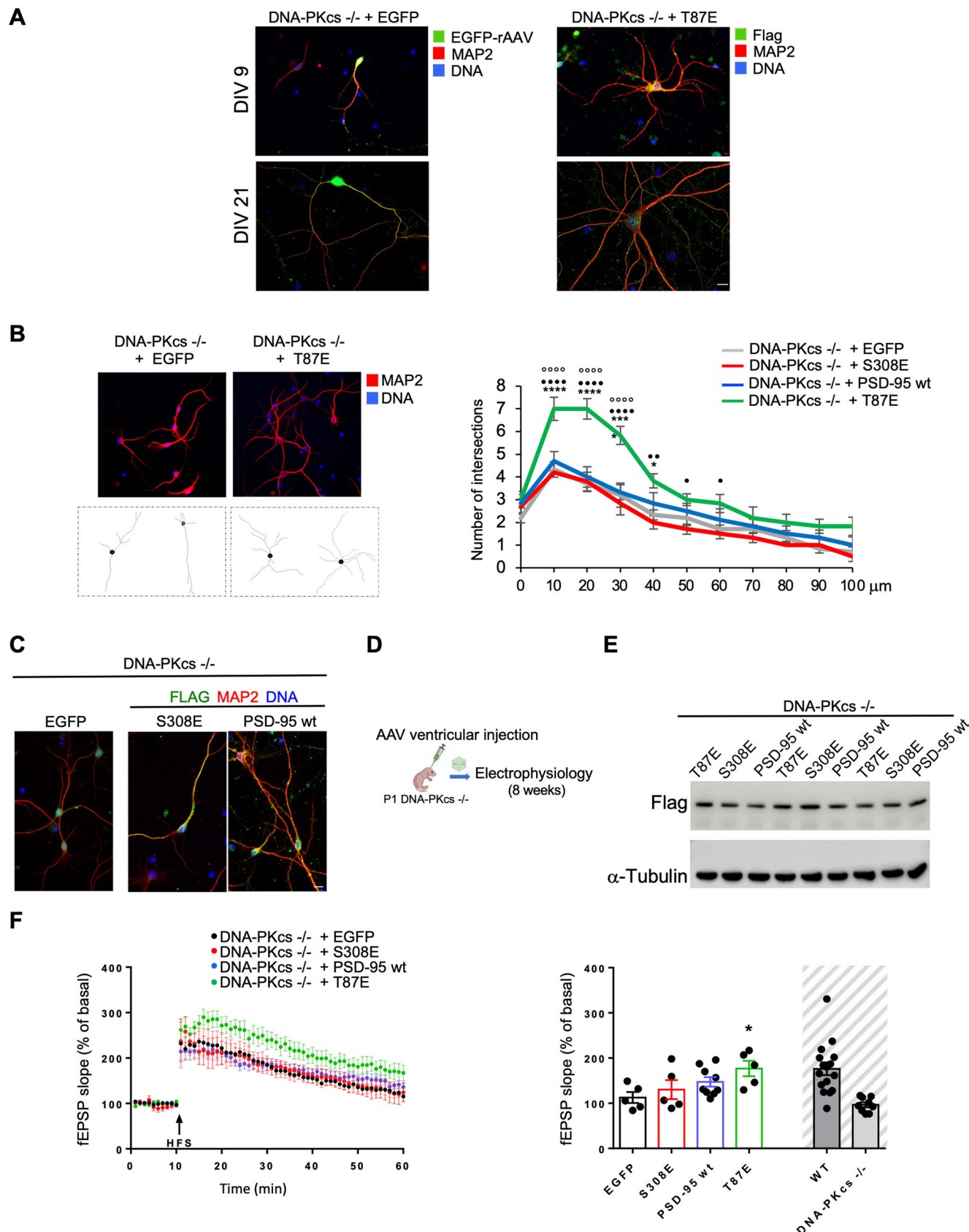

◀ 

**Figure 10.  The phospho-mimetic mutant PSD-95T87E increases dendrite complexity in DNA-PKcs −/− neurons and improves LTP induction in DNA-PKcs −/− mice.**

(A) Representative confocal and THUNDER microscopy images of triple labeled DNA-PKcs −/− primary cortical neurons (DIV 9 upper panel, and DIV 21 lower panel) stained with the anti-MAP2 (red channel), anti-Flag (green channel) antibodies and DNA dye (DAPI, blue channel). The over-expression of the phospho-mimetic T87E is able to increase the dendrite complexity in DNA-PKcs −/− cortical neurons as compared with neurons over-expressing the EGFP control vector. Scale Bar 5 μm. (B) Left panel, MAP2 immunofluorescence (red channel) images of DNA-PKcs −/− primary cortical neurons transduced with EGFP control vector or PSD-95T87E mutant. T87E transduced neurons acquire a multipolar aspect due to an increased number of neurites as compared with EGFP DNA-PKcs −/− infected neurons. Representative tracing images (skeletons) of DNA-PKcs −/− infected cultures are shown. Scale Bar 5 μm. Right panel, Sholl analysis of DNA-PKcs −/− cortical neurons transduced with EGFP control vector, PSD-95 wt or the phospho-mimetic mutants PSD-95S308E and PSD-95T87E. T87E mutant is able to increase neurite complexity in DNA-PKcs −/− cultures as indicated by a significant increase in the number of intersections in the range of 0–60 μm distance from the cell body. A sample of six neurons was taken for each group ($n = 3$ cultures). A number of intersections was counted in a 100-μm radius from the soma along the dendritic tree. Statistics by two-way ANOVA followed by Bonferroni post hoc analysis. Results are expressed as mean ± SEM. ****$p < 0.0001$ and *$p < 0.05$ T87E vs EGFP DNA-PKcs −/− infected neurons; ●●●●$p < 0.0001$, ●●$p < 0.005$, and ●$p < 0.05$ T87E vs S308E; °°°°$p < 0.0001$ T87E vs PSD-95 wt. (C) Representative confocal images of triple labeled DNA-PKcs −/− primary cortical neurons (DIV 9) stained with the anti-MAP2 (red channel), anti-Flag (green channel) antibodies, and DNA dye (DAPI, blue channel). Neither PSD-95 wt nor the phospho-mimetic S308E over-expression is able to reverse the decreased dendrite complexity of DNA-PKcs −/− neurons. Scale Bar 5 μm. (D) PSD-95T87E, PSD-95S308E, PSD-95 wt, and EGFP rAAV expressing vectors were injected into the cerebral lateral ventricles of P1 DNA-PKcs −/− mouse pups and 8 weeks hippocampal slices used for electrophysiological analysis. (E) Representative quantitative Western blot showing similar expression levels of the different Flag proteins in the hippocampus of DNA-PKcs −/− injected mice 8 weeks after rAAV brain injection ($n = 3$ hippocampi/each injected vector). (F) Left panel, time courses of fEPSP slope after HFS in slices of DNA-PKcs −/− mice injected with PSD-95T87E, PSD-95S308E, PSD-95 wt, and EGFP rAAV expressing vectors. Right panel, bar graph showing fEPSP potentiation in the four experimental groups 60 min after HFS ($n = 9$ for PSD-95 wt, $n = 5$ for each other experimental group). *$p < 0.05$ T87E vs DNA-PKcs −/− injected with EGFP, one-way ANOVA followed by Dunnett's test; Bar graph with gray background shows fEPSP potentiation in WT and DNA-PKcs −/− slices (WT $n = 16$ slices and DNA-PKcs −/− $n = 9$ slices) to compare rescue of LTP levels inT87E rAAV injected mice with uninfected mice. Gray background indicates that the two group of mice (injected and no injected) are not statistically compared. Source data are available online for this figure.

## Co-immunoprecipitation

Immunoprecipitation experiments were performed using 500 μg of protein extracts from LP1 fraction. Lysates were incubated overnight at 4 °C in slow rotation with 2.5 μl of DNA-PKcs Ab-4, and the protein–antibody complex was precipitated by adding 40 μl of protein G beads and incubation for 2–4 h 4 °C. At the end of incubation time, the samples were centrifuged at $1000 \times g$ at 4 °C for 1 min, and bound proteins were extensively washed. Proteins were released in 4x electrophoresis buffer and subjected to SDS-PAGE and Western blotting by using different antibodies (GluA1, GluN1, and GluN2 and PSD-95) to reveal co-immunoprecipitating proteins. Alternatively, immunoprecipitated-DNA-PKcs bound to beads was used for in vitro kinase assays as indicated below.

## Western blot analysis

Appropriate amounts of protein extracts were boiled for 5 min in SDS-PAGE Laemmli buffer (50 mm Tris-HCl, pH 6.8, 2% SDS, 10% glycerol, 0.1% bromophenol blue, and 50 mm DTT) and separated by SDS-PAGE (5% polyacrylamide for DNA-PKcs, 7.5 or 10% for GluN1, GluN2, GluA1, GluN2A, CaMKII alpha, ERK1/2, PKB/Akt, PSD-95, pPSD-95(T87), pPSD-95(S308), p90rsk, Flag, α-Tubulin, and actin). Proteins were electrotransferred to nitrocellulose membrane (HybondTM C-extra, GE Healthcare) at 30 V overnight at 4 °C for DNA-PKcs detection, while for other proteins at 100 V for 1 h 4 °C. Membranes were blocked for 1 h RT with 10% (w/v) milk in TBS-T solution (Blocking buffer, 0.1% Tween-20 in 1.3 m NaCl, 200 mm KCl, 250 mm Tris, pH 7.5) and incubated with primary antibodies in TBS-T containing 2 or 5% milk or BSA for 2 h RT or overnight at 4 °C with gentle shaking, using the following dilutions: mouse anti-actin 1:1000 (Sigma A3853); mouse anti-DNA-PKcs Ab-4 mixture 1:400 (Neo Markers MS-423-P); rabbit anti-total AKT 1:1000 (Cell Signaling CST-9272); rabbit anti phospho-AKT (Ser473) 1:1000 (Cell Signaling CTS-9271 S); rabbit anti-total ERK1/2 (Cell Signaling CST-9102S); rabbit anti phospho-ERK (p42 and p44) 1:1000 (Cell Signaling CST-9101S); rabbit anti CaMKIIpan 1:1000 (Cell Signaling CST-3362S); rabbit anti phospho-CaMKII (Thr286) 1:1000 (Cell Signaling CST-12716S); rabbit anti-total RSK1/RSK2/RSK3 1:1000 (Cell Signaling CST-9355S); rabbit anti phospho-p90RSK (Thr359/Ser363) 1:1000 (Cell Signaling CST-9344S); rabbit anti-Synapsin I/II (1:1000, Synaptic Systems 106002); rabbit anti-GluA1 (clone C3T) 1:2000 (Sigma-Aldrich 04-855); rabbit anti-GluN1 1:1500 (Millipore AB9864); rabbit anti-GluN2A 1:1000 (Millipore 07-632); rabbit anti-GluN2A/B (5 μg for immunoprecipitation) (Sigma, Aldrich 1548); rabbit anti-GluN2A (Millipore AB1557); mouse anti-Flag antibody (1:1000, GenScript, NJ, USA, A00187-100); goat anti Lamin A/C (1:1000, Santa Cruz sc6215); mouse anti-total PSD-95 1:1000 (Synaptic Systems 124011); rabbit anti phospho-PSD-95 (Ser308) 1:2000 (GL Biochem (Shanghai) Ltd); rabbit anti phospho-PSD-95 (Thr87) 1:250 (GL Biochem (Shanghai) Ltd); rabbit anti-α-Tubulin (1:2000, Bios bs-0159R); mouse anti Vinculin (1:1000, Millipore V9131); rabbit anti-GAPDH (1:1000, RPCA-GAPDH, Encorbio). After extensive washing in TBS-T, membranes were probed for 1 h RT with HRP-conjugated antibodies (anti-rabbit IgG 1:100,000 (711-035-152) and anti-mouse IgG 1:100,000 (715-035-151) Jackson ImmunoResearch; anti-goat IgG 1:500,000 (sc-2768) Santa Cruz Biotechnology) diluted in TBS-T containing 2% milk and washed thoroughly with TBS-T. Immunoreactive bands were visualized by an enhanced chemiluminescence detection system (EuroClone) ECL. Images were acquired using an Amersham ImmageQuant 800 (GE Healthcare), and densitometric analysis was performed using ImageQuant software (GE Healthcare) and Fiji Image software (NIH).

## Surface biotinylation

For surface biotinylation in brain slices, after electrophysiology recordings, a single slice from a single mouse (WT or DNA-PKcs −/−) was rapidly isolated on ice-cold ACSF, and washed 3x in cold ACSF and transferred to chilled ACSF containing Ez-link Sulfo-NHS-LC-biotin (1 mg/ml; Thermo Fisher Scientific) for 45 min. The biotin reaction was then quenched by washing the slices twice for 25 min at 4 °C in 10 mM glycine in ACSF, followed by 3x in cold ACSF (5 min). Slices were harvested in RIPA buffer

supplemented with protease inhibitors, and protein content was quantified. Total GluA1 protein was detected before streptavidin-agarose bead precipitation. Equal amounts of total proteins were used for biotinylated protein precipitation using streptavidin-agarose beads (Thermo Fisher Scientific) at 4 °C for 2 h, centrifuged at $13,800 \times g$ for 15 min, and washed twice with cold PBS. Proteins were then eluted from beads using Laemmli sample buffer, and boiled for 5 min to determine the GluA1 cell surface.

## Immunofluorescence microscopy and image acquisition

Neurons grown on poly-L-lysine-coated glass coverslips were fixed with 4% paraformaldehyde/phosphate-buffered saline containing 4% sucrose for 10 min at 37 °C, washed for 5 min with phosphate-buffered saline, permeabilized with 0.2% Triton X-100 in phosphate-buffered saline for 10 min and washed three times for 5 min with phosphate-buffered saline. Cells were then processed with primary antibodies followed by secondary antibodies and counterstained with DAPI, or in certain experiments with propidium iodide after RNAse treatment, as described (Carunchio et al, 2008). The following primary antibodies were used (60 min, 37 °C): anti-DNA-PKcs Ab-4 mixture 1:300 (MS-423-P, mouse, Thermo Fisher Scientific); anti-MAP2 (1:250, AB5622 rabbit, Millipore, Burlington, MA, USA), (M4403 mouse, Sigma-Aldrich, Milan, Italy) and (1:500, GT22102 goat, Neuromics, Edine USA); anti-Synapsin I (1:250, AB1543 rabbit, Millipore, Burlington, MA, USA); anti-Sintaxin I (1:300, PA5-29765 rabbit, Thermo Fisher Scientific); anti-Flag antibody (1:250, A00187-100 mouse, Gen-Script, NJ, USA), anti-PSD-95 (1:200, 04-1066, rabbit Millipore) and (1:300, N3702 FluoTag®-X2 anti-PSD-95 alpaca, NanoTag Biotechnologies), anti-GluA1 (1:250, rabbit clone C3T, Sigma 04-855) and (1:250, 182003 rabbit, Synaptic Systems, Germany); anti-Homer 1 (1:300, 160003 rabbit, Synaptic Systems, Germany); anti-GluN1(1:300, 114103 rabbit, Synaptic Systems, Germany). All antibodies were diluted in phosphate buffer containing 3% bovine serum albumin and 0.05% Tween-20 and incubated 1 h at 37 °C. After rinsing, the coverslips were incubated with fluorescently labeled secondary antibodies (1:250) (Thermo Fisher Scientific) for 30 min at RT. After a thorough rinse, coverslips were mounted in ProLong Glass medium and analyzed by confocal microscope Nikon Eclipse Ti2 microscope, with an Eclipse 80i Nikon Fluorescence Microscope, equipped with a VideoConfocal (ViCo) system. Moreover, for high-resolution images, coverslips were also examined using a THUNDER Imager 3D Live Cell & 3D Cell Culture microscope (Leica) with the innovative technology of computational clearing. Generally, Z-series were obtained by imaging serial confocal planes at 0.25-μm intervals and finally, maximum projection reconstructions of Z-series images were used for morphometric analysis and quantification.

Fiji image analysis software (NIH Image) was used to analyze and quantify the whole-cell morphometry and mean intensity of fluorescence. For fluorescence intensity analysis, the settings for image acquisition were constant for all of the scans. Mean fluorescence intensity in neurites was calculated as average of all pixels within regions of interests (ROIs).

Background subtraction was performed by subtracting the mean intensity value estimated from a single background ROI placed within an unlabeled region in the same image.

For puncta analysis, an ImageJ129 plugin called Puncta Analyzer was used kindly provided by Cagla Eroglu Durham, USA (https://sites.duke.edu/eroglulab/tools/) (Ippolito and Eroglu, 2010). Previous studies showed that this quantification method yields an accurate estimation of the number of synapses in vitro and in vivo which were previously confirmed by other methods. The quantification of synaptic puncta in each channel and colocalized synaptic puncta between the two channels was obtained following the author's instructions as described by (Ippolito and Eroglu, 2010). DNA-PKcs puncta were quantified by Puncta Analyzer on maximum projection 63x magnification images ($n = 7$ per each group), separated into red and green channels, background subtracted (rolling ball radius = 50), and thresholded in order to detect discrete puncta without introducing noise. User-defined thresholding of images in order to detect puncta with a minimum length of 0.25 μm that corresponds to synapses. Equally, thresholding was applied across acquired images.

A ×20 objective was used to count piknotic nuclei and measure the mean dendrite length. The total neurite length of DIV 9 neurons was analyzed by tracing MAP2-labeled neurites using the NeuronJ (Meijering et al, 2004) plugin for Fiji (NIH) Software. All dendrites of individual neurons were traced, and the number of pixels was automatically counted and converted to micrometers.

For Sholl analysis, concentric circles of 10 μm in diameter around the cell body were used to count the number of dendrites (MAP2-labeled) passing through each circle. The analysis was conducted on ten neurons per experimental group, as described by (Mollinari et al, 2015). Briefly, cortical mouse neurons were chosen from cell culture and scanned using a Nikon (40x, 1.3 NA or 60x, 1.42 NA). Images were imported into Fiji image analysis software, and dendritic complexity was analyzed from 8-bit images by using the Sholl Analysis plug-in. The average dendritic length and number of intersections of each individual neuron were analyzed.

For the analysis of dendritic spines, DIV 21 neurons were imaged with a 63x, 1.4 NA oil immersion objective and at a 2048 × 2048-pixel resolution using a THUNDER Imager 3D Live Cell & 3D Cell Culture microscope (Leica). Z-series were obtained by imaging serial confocal planes at 0.25-μm intervals and finally, maximum projection reconstructions of Z-series images were used for spine analysis and quantification. All imaging and counting procedures were performed blind to genotype. To quantify the neuronal spines, we analyzed GFP-positive cells after rAAV infection, further immunolabeled with antibody against MAP2 and synaptic proteins. Regions of dendrite corresponding to 50 μm at different distance from the cell body were analyzed and dendritic spines, clearly visible (without excessive crossing of multiple dendrites or axons) were counted with the plugin called SpineJ of Fiji image analysis software. Ten neurons were considered per experimental group. A minimum of 130 spines were counted per ten neurons, and the count was repeated in three different independent experiments. Spine density was determined by summing the total number of spines per dendritic segment (50 μm) and calculating the average number of spines per micrometer. In order to increase the magnification for a better view of the spines without loss of image quality, the resolution of the stack image was increased by a factor of 6 in the X and Y directions with the plugin Transform J Scale (Meijering et al, 2001).

## Electrophysiology

Mice were sacrificed by cervical dislocation, and the brains were isolated and immersed in ice-cold artificial cerebrospinal fluid (ACSF) containing (in mM): 126 NaCl, 3.5 KCl, 1.2 NaH$_2$PO$_4$, 1.2 MgCl$_2$, 2 CaCl$_2$, 25 NaHCO$_3$, 11 glucose (pH 7.3) saturated with 95% O$_2$, and 5% CO$_2$. Parasagittal slices (400 μm) containing hippocampus were maintained at room temperature (RT) (22–24 °C) in ACSF for at least 1 h, then, each slice was transferred to a submerged recording chamber and continuously superfused at 32–33 °C with ACSF at a rate of 2.6 ml/min. Extracellular field excitatory postsynaptic potentials (fEPSPs) were recorded in the *stratum radiatum* of the CA1 with a glass microelectrode filled with 2 M NaCl solution (pipette resistance 2–5 MΩ) upon stimulation of Schaffer collaterals with an insulated bipolar twisted NiCr electrode (50 μm OD). Responses were characterized by the application of paired-pulse stimulation (PPS, two consecutive pulses 50-msec apart), and only fEPSPs showing paired-pulse facilitation (PPF, R2/R1 ratio ≥1) were used for recording. Paired-pulse ratio (PPR) was used as an indicator inversely proportional to presynaptic neurotransmitter release (Schultz, 1998). After PPS, during conventional recording, each pulse was delivered every 20 s (square pulses of 100 μs duration at a frequency of 0.05 Hz), and three consecutive responses were averaged. Signals were acquired with a DAM-80 AC differential amplifier (WPI) and analyzed with the LTP program (Anderson and Collingridge, 2001). The input/output (I/O) curves were plotted as the relationship between the fEPSP slope (mV ms$^{-1}$) and the stimulation intensity (μA). Long-term potentiation (LTP) was induced by using a high-frequency stimulation (HFS) protocol consisting of two trains of 100 pulses at 100 Hz, 20 s apart. The magnitude of HFS-induced posttetanic potentiation (PTP) and LTP were quantified by calculating the mean percentage of the slope in the time windows 0–3 min and 45–55 min after HFS, respectively.

After electrophysiological recording, individual slices were immediately frozen and stored in liquid nitrogen and subsequently used for protein extraction.

## In vitro DNA-PKcs kinase assay

Protein extracts from HEK293 cells (200 μg), P3 and LP1 human cortex fractions (200 μg), and mouse LP1 fraction (1 mg) were precleared by incubation with 40 μl of protein G bead slurry (50%) (Pierce) for 60 min at 4 °C with end over end rotation. After centrifugation at 1000 × g at 4 °C for 1 min, the protein G was separated from the supernatant and discarded. Cleared lysate was incubated for 14–16 h at 4 °C in slow rotation with 2.5 μg of DNA-PKcs Ab-4, and protein–antibody complex was precipitated by adding 40 μl of protein G beads and incubation for 2–4 h at 4 °C. At the end of incubation time, the sample was centrifuged at 1000 × g at 4 °C for 1 min, and bound proteins were washed. Kinase reactions were carried out in vitro according to the manufacturer's instructions (Promega, SignaTect DNA-dependent Protein kinase Assay System). Kinase reactions were conducted with 20 μl aliquots of the resuspended DNA-PKcs-bound to protein G beads for 30 min at 30 °C and were performed in both the presence and absence of a biotinylated DNA-PKcs p53-derived substrate.

Moreover, the in vitro kinase assay was performed by using immune-purified DNA-PKcs from LP1 mouse cortex incubated with human recombinant PSD-95 (Origene) to confirm the specificity of the phosphorylation. Briefly, the kinase reactions were conducted with

20 μl aliquots of the resuspended DNA-PKcs-bound to protein G beads for 30 min at 30 °C and were performed in both the presence and absence of PSD-95 and ATP. After the reaction, each condition was run on ready precast gradient polyacrylamide gels, blotted and incubated in sequential order with the following antibodies: anti-DNA-PKcs, anti-pS308 or -pT87, and anti-PSD-95. Immunoreactive bands were visualized by an enhanced chemiluminescence detection system (EuroClone) ECL.

## In vitro customized kinase activity assay

The phosphorylation of seven different proteins by DNA-PKcs was evaluated: DNA-PKcs LOT002, (Invitrogen product PR9107A, Lot#1484325 A), ERK2, active (ProQinase product #0634-0000-7, Lot#008), ERK2, non-activated (ProQinase product #0634-0000-1, Lot#005), RPS6KA1 LOT002, (Invitrogen product PV4049, Lot#386267Z1A), DLG4 (Origene product TP315178, Lot#060811), GRIN1 (Origene product TP319368, Lot#100814) TP53 (Origene product TP300003, Lot#17J10L19), and GRIA1 (Origene product TP326253, Lot#BJ0A64E). The potential substrate proteins and the kinase were incubated with ATP and radioactive tracer ATP (33P-γ-ATP) and samples were analyzed by SDS-PAGE and subsequent autoradiography. Kinase and substrates were mixed in a total reaction volume of 50 μl in Eppendorf reaction vessels in the following order: 15 μl of 3.33x assay buffer, 20 μl substrate solution (RBER-IRStide),10 μl kinase solution, 5 μl ATP solution. The assay contained 70 mM HEPES-NaOH pH 7.5 (6.5), 3 mM MgCl$_2$, 3 mM MnCl$_2$, 2.5 μg/ml DNA, 3 μM Na-orthovanadate, 1.2 mM DTT, ATP at 10 μM, [γ-33P]-ATP (~2.5 × 1007 cpm per sample), DNA-PKcs (2 μg/ml) and substrate (20/40 μg/ml). The reaction cocktails were incubated at 30 °C for 60 min. The reaction was stopped with 20 μl LDS-Gel loading buffer (Invitrogen) and incubated for 3 min at 95 °C. About 20 μl of the mixture were loaded on a 4–20% Bis-Tris-SDS-PA gel. The samples were analyzed by electrophoresis and the gel was stained by Coomassie to visualize the total protein loading. The gel was dried and exposed to X-ray film at various times.

To identify the possible phosphorylation sites in PSD-95 by DNA-PKcs, the experiments were repeated in a non-radioactive condition and the reaction products were sent to the service for Mass Spectrometry analysis (CEA Grenoble, France).

## Mass spectrometry-based proteomic analysis and mass spectrometry-based proteomic data processing (Appendix materials and methods)

### Phosphoarray

CA1 hippocampal regions from different animals with 5 min of HFS stimulation (T5') or without (T0) were pulled and subjected to protein extraction. Four different conditions were taken into account for the array: WT mouse brain tissue unstimulated (WT T0); WT mouse brain tissue stimulated (WT T5); DNA-PKcs −/− mouse brain tissue unstimulated (KO T0); DNA-PKcs −/− mouse brain tissue stimulated (KO T5). The samples were extracted in RIPA buffer: Tris 50 mM. 150 mM NaCL, 1%Triton X-100, 05% Sodium deoxycholate, 0.1% SDS, DTT 0.1 mM, protease inhibitors, and phosphatase inhibitors. Brain extracts were then frozen in liquid nitrogen and sent to FullMoon BioSystems to perform a Phospho Explorer Antibody Array (PEX100).

## PSD-95 mutant vectors

The sequence (NM_001365.4) corresponding to *Homo sapiens* disks large MAGUK scaffold protein 4 (DLG4), transcript variant 1, was used as a template to obtain the PSD-95 wt and PSD-95 mutants. We obtained from the GenScript company (Leiden, The Netherlands) four mutants containing a single mutation either to mimic constitutive phosphorylation (S/T into E: S308/E and T87/E) or non-phosphorylatable mutants (S/T into A: S308/A and T87/A). The mutants were generated by PCR and the construcs confirmed by DNA sequencing. The different PSD-95 fragments were subcloned into the NheI-HindIII sites of a modified rAAV vector containing a CMV promoter "BankIt2829339 Plasmid PP808633" (Mollinari et al, 2015) from a backbone kindly provided by Hilmar Bading, Heidelberg, Germany) (Lau and Bading, 2009). All the PSD-95 proteins were expressed with a Flag tag at the COOH terminus to be distinguishable from the endogenous PSD-95 for Western blotting and immunofluorescence experiments.

## Recombinant AAV viruses and infection

rAAV (rAAV1/2) mosaic vectors containing a 1:1 ratio of AAV1 and AAV2 capsid proteins with AAV2 inverted terminal repeats (ITRs) were generated by cross-packaging as previously described (Hauck et al, 2003). For in vitro experiments (DIV 9), 2 μl of the rAAV viral particles were added into the culture medium at DIV 4 neurons, resulting in a visual EGFP signal after 3–4 days and almost 100% transduction efficiency. For DIV 21 culture experiments, 2 μl of the rAAV viral particles (PSD-95 wt and PSD-95 mutants) were added into the culture medium at DIV 6 neurons, followed by a second infection with EGFP-rAAV viral particles performed at DIV 11, to obtain a better visualization of dendritic spines. For in vivo experiments, postnatal day 1 mouse pups (P1) were injected as described (Li and Daly, 2002; Mollinari et al, 2015).

## Generation of phospo-specific antibodies

Specific phospho-peptides were ordered from GL Biochem (Shanghai) Ltd. Phospho-peptides (incorporating a cysteine residue at the N terminus) conjugated to keyhole limpet hemocyanin (KLH), were used to immunize New Zealand White rabbits (Covance). For affinity purification of the phospho PSD-95 antibodies, each antiserum was passed through a SulfoLink column (Pierce) coupled with the corresponding non-phosphorylated peptides. Pass-through fractions were subsequently purified on a SulfoLink column coupled with the phosphorylated peptide. After washing with Tris-buffered saline, the bound antibody was eluted with 0.1 M glycine (pH 2.7). The following phosphopeptides were synthetized: 82-94aa: IVNTD(T)LEAPGYE-Cys T87; 291-303aa: NAYLSD(S)YAPPDI-Cys S297; 304-317aa: TTSY(S)QHLDNEISH-Cys S308; 517-531aa: Cys-RRVH(S)DSETD-DIGFI S521; 517-531aa: Cys-RRVHSD(S)ETDDIGFI S523; 548-561aa: Cys-KDWGSSSG(S)QGRED S556.

## Statistics

Statistical analyses were performed using GraphPad Prism (version 6.05, USA). First, the dataset were subjected to the Shapiro–Wilk test to check for the normal distribution. Then, significant differences between two groups were evaluated with an unpaired two-tailed Student's *t*-test. One-way ANOVA and two-way ANOVA were used for the multiple comparison followed by post hoc tests. For details, see the description in results and figure legends. Results from experiments were expressed as mean ± standard error of mean (SEM). The exact *p* values are provided unless it is at least $p < 0.05$, accepted as the level of significance for all of the tests.

## Data availability

The datasets produced in this study are available in the following database: BioStudies (Accession Number: S-BSST1420). Phospho Array Antibodies, Information on Phospho-Antibody Production, and DNA constructs are available at: https://www.ebi.ac.uk/biostudies/studies/S-BSST1420. The constructs and antibodies generated in this study are available upon request from the corresponding authors with the appropriate Material Transfer Agreement (MTA).

The source data of this paper are collected in the following database record: biostudies:S-SCDT-10_1038-S44319-024-00198-3.

## Peer review information

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

## Acknowledgements

This research has been funded by Istituto Superiore di Sanità, Fasc.R 301 (Project title "Ruolo della DNA-PK nei deficit cognitivi associati alle malattie neurodegenerative") to DM. This study has also been supported by the Italian Ministry of Health (Ricerca Corrente 10/2405 to LL) and by the Italian National Research Council (CNR) by FOE2022 to LL. We thank Anne-Marie Hesse from the EDyP-Service platform in CEA of Grenoble (France) for performing Nanoscale liquid chromatography coupled to tandem mass spectrometry (nano LC-MS/MS) analysis and helping present data. We thank Eleonora Petrucci (ISS) for helping with THUNDER microscope imaging and Flavia Mayer for helping with statistics. We thank Cinzia Volonté for the critical revision of the manuscript. The synopsis was created with BioRender.com (Agreement number XN26ZFDXOQ).

## Author contributions

**Cristiana Mollinari**: Conceptualization; Data curation; Software; Investigation; Methodology; Writing—original draft; Writing—review and editing. **Alessio Cardinale**: Conceptualization; Data curation; Investigation; Methodology; Writing—review and editing. **Leonardo Lupacchini**: Conceptualization; Data curation; Software; Funding acquisition; Investigation; Methodology; Writing—review and editing. **Alberto Martire**: Conceptualization; Data curation; Formal analysis; Investigation; Methodology; Writing—review and editing. **Valentina Chiodi**: Investigation; Methodology. **Andrea Martinelli**: Methodology. **Anna Maria Rinaldi**: Methodology. **Massimo Fini**: Funding acquisition; Project administration. **Simonetta Pazzaglia**: Investigation; Methodology. **Maria Rosaria Domenici**: Conceptualization; Data curation; Formal analysis; Investigation; Methodology; Writing—review and editing. **Enrico Garaci**: Conceptualization; Formal analysis; Funding acquisition; Project administration; Writing—review and editing. **Daniela Merlo**: Conceptualization; Data curation; Formal analysis; Supervision; Funding acquisition; Investigation; Writing—original draft; Project administration; Writing—review and editing.

Source data underlying figure panels in this paper may have individual authorship assigned. Where available, figure panel/source data authorship is listed in the following database record: biostudies:S-SCDT-10_1038-S44319-024-00198-3.

## Disclosure and competing interests statement

The authors declare no competing interests.

# Expanded View Figures

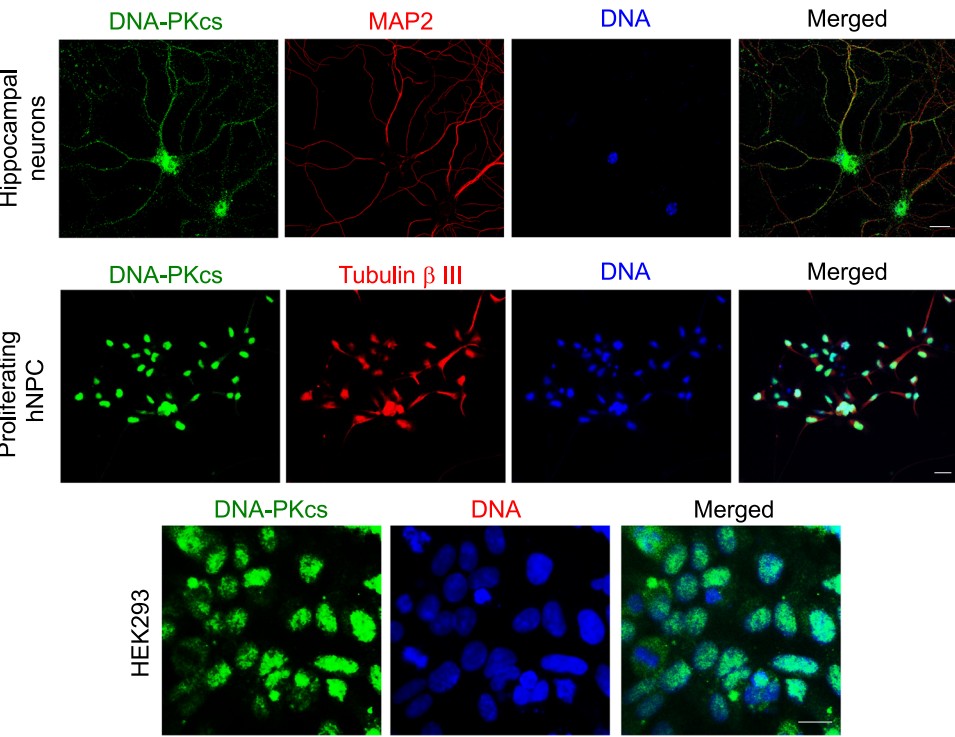

**Figure EV1. DNA-PKcs has a lower nuclear staining in post-mitotic neurons as compared with proliferating cells.**

Representative immunofluorescence images showing the distribution of DNA-PKcs (green) in post-mitotic mouse hippocampal neurons as compared with proliferating human neural progenitor cells (NPCs) and HEK293 cells. In proliferating cells, DNA-PKcs appear mainly concentrated in the nucleus, whereas in neurons, it is abundant along dendrites showing a punctate staining and lower nuclear labeling. Scale Bar 10 μm.

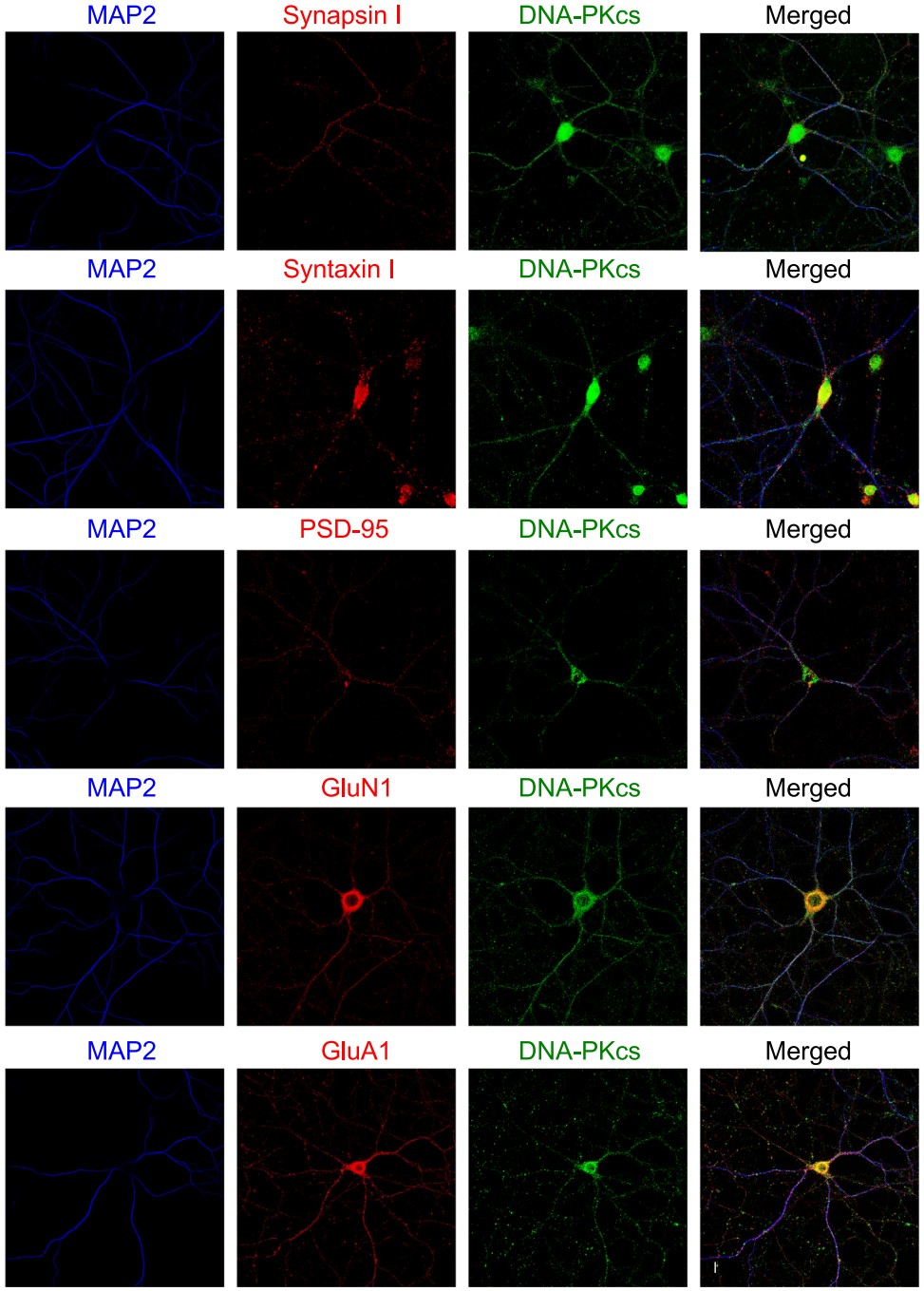

**Figure EV2.  DNA-PKcs shows a synaptic distribution in cortical neurons.**

Representative triple immunofluorescence THUNDER images of mouse primary cortical neurons (DIV 21) labeled with the anti-DNA-PKcs antibody (green channel), the synaptic markers: Synapsin I, Syntaxin I, PSD-95, GluN1, and GluA1 (red channel), one at a time, and the neuronal marker MAP2 (blue). Single-channel images are provided to better show the localization of each synaptic marker and the distribution of DNA-PKcs similar to the synaptic proteins. Scale Bar 5 μm.

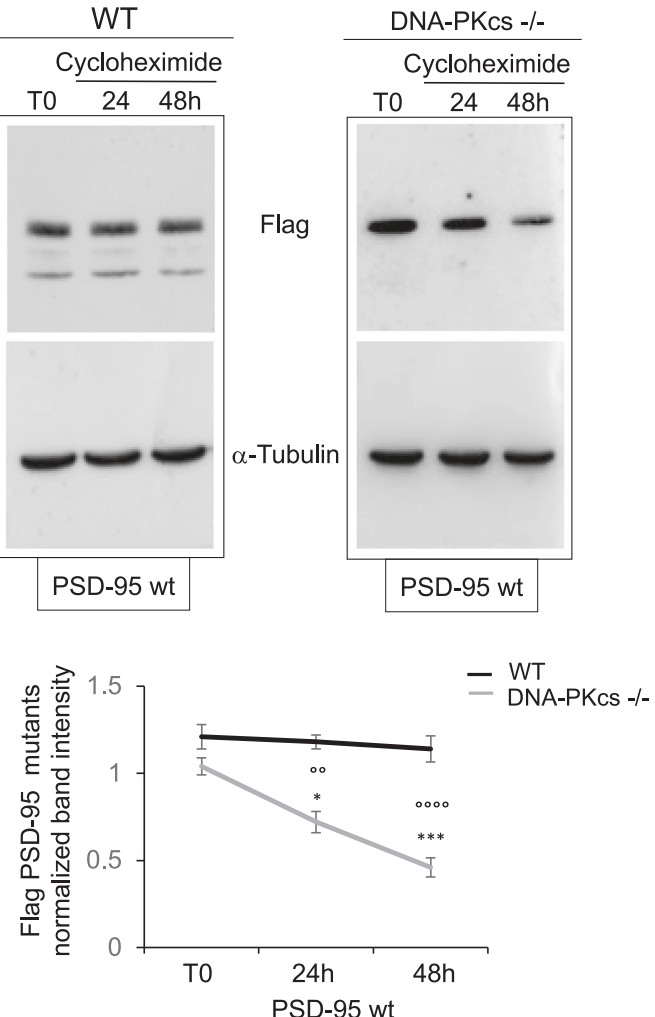

**Figure EV3. PSD-95 wt over-expressed protein is less stable in the absence of DNA-PKcs kinase activity.**

Representative Western blots of protein extracts from cortical neurons of WT and DNA-PKcs −/− mice show that PSD-95 wt protein, over-expressed in WT neurons, remains stable after cycloheximide treatment up to 48 h, whereas it decreases over time when over-expressed in DNA-PKcs −/− neurons. Values in the plot represent the quantification of PSD-95 wt protein levels over time following cycloheximide treatment normalized to α-Tubulin. (means ± SEM; $n = 3$). Statistics by two-way ANOVA followed by Bonferroni post hoc analysis. *$p < 0.05$ DNA-PKcs 24 h vs DNA-PKcs T0, ***$p < 0.001$ DNA-PKcs 48 h vs DNA-PKcs T0, °°$p < 0.005$ DNA-PKcs 24 h vs WT 24 h, °°°°$p < 0.0001$ DNA-PKcs −/− 48 h vs WT 48 h. Source data are available online for this figure.

