## [Peer Review File · EMBO Reports]

The DNA repair protein DNA-PKcs modulates synaptic plasticity via PSD-95 phosphorylation and stability

Cristiana Mollinari, Alessio Cardinale, Leonardo Lupacchini, Alberto Martire, Valentina Chiodi, Andrea Martinelli, Anna Maria Rinaldi, Massimo Fini, Simonetta Pazzaglia, Maria Rosaria Domenici, Enrico Garaci and Daniela Merlo

Corresponding author(s): Daniela Merlo (daniela.merlo@iss.it)

Review Timeline:

Submission Date:	25th Jul 23
Editorial Decision:	25th Aug 23
Appeal Received:	1st Mar 24
Editorial Decision:	8th Apr 24
Revision Received:	10th May 24
Editorial Decision:	11th Jun 24
Revision Received:	18th Jun 24
Accepted:	24th Jun 24

Editor: Esther Schnapp

Transaction Report:

Dear Dr. Merlo,

Thank you for the submission of your manuscript to EMBO reports. So far, I could only secure 2 referees for it, but given that their reports are in fair agreement, I am making a decision now in order to save you from unnecessary loss of time.

I am sorry to say that the evaluation of your manuscript is not a positive one. As you will see, while the referees acknowledge that the study is potentially interesting, they also point out that the data are not sufficiently strong to support the main conclusions. They pinpoint a number of technical issues that preclude a solid interpretation of the experimental evidence provided.

Given these expert evaluations and the fact that EMBO reports can only invite revision of papers that receive enthusiastic support from the referees, I am afraid that we cannot offer to publish your manuscript.

I am sorry to disappoint you on this occasion, and hope that the referee comments will be helpful in your continued work in this area.

Kind regards,
Esther

Referee #1:

This manuscript by Mollinari et al. explores a potential synaptic role of DNA-PKcs. Using a knockout mouse model and phosphoproteomics, the authors identified phosphorylation sites on PSD-95 that are targeted by DNA-PKcs. Further, they show potential roles for DNA-PKcs in synaptic plasticity and provide evidence that PSD-95 phosphorylation is downstream of DNA-PKcs in mediating the effects.

While these data are potentially interesting, the manuscript overall suffers from a number of technical problems in experimental design and execution, including the lack of controls, low resolution images, and low quality Western blots, which prevent the readers from properly evaluating the data. Overall the current data are inadequate to support their conclusions. In addition, the manuscript needs to be edited extensively for English language as there are a number of grammatical and typographical errors. The writing is also somewhat disorganized which makes it difficult to follow. Specific points are as follows:

1. In Figure 1b and c, it would be necessary to confirm the specificity of the DNA-PKcs antibody in immunocytochemistry. Their fractionation experiment (along with the expected nuclear function of DNA-PKcs) shows DNA-PKcs abundant in the nucleus. However the immunostaining in Fig 1c shows little nuclear localization, which raises the question of whether the immunostaining signals are specific. Since the authors have the knockout mice they can easily test this.
2. In Figure 1c, higher resolution images with more examples of co-localization are needed, especially since the colocalization of DNA-PKcs with synaptic markers appears low. It is also unclear whether the quantification of colocalization includes the soma signals or only at the synaptic level.
3. Fig 1e, a negative control is needed.
4. Figure 1f shows an increase in DNA-PKcs staining in the soma and proximal dendrites. It would be good to confirm this is the case with Western blot of whole cell lysates, as Fig. 1g implies the increase is specific to synapses.
5. Fig 1i: a non-specific antibody control (e.g. mouse IgG or anti-GFP antibody) is needed for the co-IP of synaptic proteins.
6. Figure 3b: DIV9 is too early to properly examine synapses. Higher resolution images at a later time point are needed to determine the effects on synapses.
7. Similarly, Figure 3c needs much higher resolution images to properly evaluate GluA1 puncta.
8. Figure 4d and e: Something is off with the spine quantification as it shows the WT has just over 1 spine per 100 um, which is way too low. Was the spine quantification also done at DIV9? If so the authors need to examine more mature neurons. Higher resolution images are needed.
9. An in vitro assay using purified proteins is needed to demonstrate that DNA-PKcs is the kinase directly phosphorylating PSD-95.
10. The quality of the Western blots in Figure 7d makes it difficult to evaluate the effects of the phosphorylation mutants. In several cases the bands show a lower intensity at an earlier time point than a later time point with longer CHX incubation. This does not make sense and raises the question as to whether the differences are just due to variability in transfection efficiency. Better quality blots from more standardized transfections are needed to evaluate the role of the phosphorylation mutants.
11. Figure 8b: a synaptic marker is needed to ensure the synaptic localization of the mutants.
12. Figure 8c: MAP2 is not a reliable way to visualize spines. Typically MAP2 mostly stays in the dendritic shaft. The authors need to use a cell-fill like GFP.
13. Figure 8d: similar to Figure 4, something is off with the spine quantification here. The spine density is too low. The authors

need to examine spines in more mature neurons and double check the data to ensure accurate quantifications.

14. Figure 9b: it appears that the authors used EGFP as a marker for Sholl quantification for the control while using MAP2 as a marker for the mutant condition. They need to use a consistent marker for all experimental conditions.

15. Figure 9d: control images are needed to demonstrate similar AAV-mediated expression of S308E vs T87E mutants.

Minor points:

1. In Figure 1b, the lanes LP2 and LS2 are swapped between DNA-PKcs and the fractionation controls blots. It would be better if the lane sequences are consistent to avoid confusion.

Referee #2:

The manuscript by Mollinari et al. described a new function of the kinase PKcs in regulating synaptic function of neurons. The authors found that the kinase, which has been well studied in DNA repair, is present in the synaptic membrane. Knockout of the gene reduces the abundance of different synaptic proteins and results in impaired synaptic transmission and synaptic plasticity. They further identify PKcs can phosphorylate the postsynaptic protein PSD-95 at novel sites, which the authors think this event increases the stability of the protein. Finally, the authors over-expressing the phospho-mimetic PSD-95 to rescue the phenotypes of the PKcs KO neurons.

While the idea that synaptic plasticity is regulated by PSD-95 phosphorylation at novel sites by a kinase involved in DNA repair is certainly interesting, I found the data too preliminary, with some experiments lacking the essential controls to support their conclusions. The other major problem is that the cellular data is not convincing, largely because the authors use cortical neurons that are too young and do not form mature synapses. This makes it very difficult to reconcile the data to a role in synaptic plasticity.

Specific comments:

1. The localization of PKcs at the synapse is not convincing. First the antibody specificity in staining has to be verified by comparing the staining between wild-type and KO neurons. Indeed, it is worrying that in Fig. 1c the nuclei appear to lack PKcs, which I suppose should not be the case if it is a protein for DNA repair and is much enriched in P1 (Fig. 1b), suggesting that the antibody is not specific for staining. It is also important to verify the synaptic localization by exogenously expressing a GFP- or other tagged PKcs in neurons. In addition, in the fractionation experiments the band intensity in LP1 is very weak, indicating the proportion at the synapse is modest. The authors must include the crude homogenate in the Western blot in order to give an idea about the percentage of PKcs in the synaptic membrane.
2. The authors used 9 DIV cultured cortical neurons to examine neuronal morphology related to synaptic plasticity, but this stage is too young to form mature synapses (which is indicated by the appearance of mostly filopodia or thin spines instead of mushroom-shaped spines). Without the mushroom spines it is not clear if the PKcs co-localized with PSD-95 are really synaptic. The use of cortical neurons also makes it difficult to link the cellular data with the electrophysiological data from hippocampal slices of much older mice. In general, neurons of around 21 DIV are used for this kind of analysis.
3. Related to 2, the use of young neurons makes it hard to judge the staining data in Fig. 1 and 3. There appear to be lot of puncta outside the neurons which seem to be background but not real signals. It is important to co-stain the neurons with MAP2 (or GFP if the neurons are infected) which outline the dendrites for quantifying the puncta, thus avoiding the non-specific background staining. The image quantification is also poorly described. For figure 3b, c, is it the number, area or intensity that the authors measured? How did they determine the co-localization (by Manders coefficient using software or were they manually quantified)? Were the images put on a threshold to define the puncta or they were defined arbitrarily? Were the quantification performed blind? This essential information in image analysis is entirely missing in the manuscript.
4. Fig. 1 h and i: it is essential to include IgG control in the co-IP experiments (or even better, use the KO samples as the negative control).
5. Fig. 3a: It is important to include the crude homogenate in order to determine if the total expression of those proteins is reduced, or only the abundance at the synaptic membrane is affected in the KO samples. In addition, does the hippocampal slice from KO mice also show reduced PSD-95 expression compared to wild-type? This is important because later the authors show that expressing the phospho-mimetic PSD-95 can rescue the LTP.
6. Fig. 4a: The cell densities appear much lower in the KO neurons. Does the gene knockout affect neuronal survival? Can the observed differences in neuronal morphology be caused indirectly by the different cell density instead of direct function of the gene?
7. Fig. 6: it is important to add the in vitro kinase assay using the two phospho-specific antibodies to validate purified PKcs indeed phosphorylates the recombinant PSD-95. Fig. 6c: the authors must use total PSD-95 instead of tubulin as the control to demonstrate the induction of phosphorylation. Fig. 6d: the tubulin bands for WT are clearly decreased in the 24 and 48 hr time points, raising the possibility that the two samples were over-loaded and masked the decrease in PSD-95 expression at these later time points.
8. Fig. 7: it is hard to understand why the stability of the SE and SA mutant are compared in different neurons. Since the Ser site has been replaced by either E or A, it can no longer be phosphorylated by PKcs and thus they should all be expressed in wild-type neurons for fair comparison. On the other hand, expressing the two sets of mutants in different neurons is problematic because one cannot rule out the possibility that the protein degradation machinery is altered in the KO neurons. Fig. 7d: the band for T87E at 24 hr is clearly down-regulated compared to T0 but the quantification indicated no difference.
9. Fig. 8 and 9: the conclusion about the importance of PSD-95 is not convincing. To show that PSD-95 phosphorylation is the

major downstream event of PKcs in regulating those processes, the appropriate way should be comparing between the SE mutant and the WT in rescuing the phenotypes. Just expressing SE mutant in the rescue experiments does not lead to the conclusion that the PSD-95 phosphorylation is important. It is possible that expressing the WT or even the SA mutant can also rescue the phenotypes.

10. The format of writing throughout the manuscript (especially the introduction) is strange because there are too many separate sentences, like putting together different bullet points.

** As a service to authors, EMBO Press provides authors with the ability to transfer a manuscript that one journal cannot offer to publish to another journal, without the author having to upload the manuscript data again. To transfer your manuscript to another EMBO Press journal using this service, please click on Link Not Available

March 1st, 2024

Esther Schnapp
Senior Editor
EMBO Reports

Dear Editor,

We are submitting a modified version of the manuscript entitled “The DNA repair protein DNA-PKcs modulates synaptic plasticity via PSD-95 phosphorylation and stability” by Mollinari C., Cardinale A., Lupacchini L., Martire A., Martinelli A., Chiodi V., Rinaldi A.M., Fini M., Pazzaglia S., Domenici M.R., Garaci E., Merlo D.

In this new version we have made the changes in accordance with the criticisms of the Referees.

Here we provide the point-by-point answers to the Referees as follows:

Referee #1:

Comment 1. In Figure 1b and c, it would be necessary to confirm the specificity of the DNA-PKcs antibody in immunocytochemistry. Their fractionation experiment (along with the expected nuclear function of DNA-PKcs) shows DNA-PKcs abundant in the nucleus. However, the immunostaining in Fig 1c shows little nuclear localization, which raises the question of whether the immunostaining signals are specific. Since the authors have the knockout mice they can easily test this.

Answer: The DNA-PKcs antibody specificity is confirmed by the absence of immunofluorescence (IF) staining following antibody neutralization by incubation with an excess of recombinant DNA-PKcs protein as shown in Figure 1c, lower panel. This result had already been specified in the Legend to Figure 1 of the previous version as follows: “DNA-PKcs antibody specificity is confirmed by the absence of immunofluorescence when antibodies are pre-incubated with an excess of recombinant DNA-PKcs protein (data not shown)”. Moreover, as suggested by Referee 1, we now show DNA-PKcs antibody specificity by the absence of IF staining on primary cortical and hippocampal cultures (DIV 21) of knockout mice (Figure 3b). Furthermore, in Figure 3a (lane DNA-PKcs^{-/-} hippocampus, previously shown) Western Blot (WB) analysis on hippocampal protein extract from knockout mice confirms the specificity of the used antibody in WB by the lack of DNA-PKcs immunoreactive band.

Please note that the antibody we used is a mixture of 4 different monoclonal antibodies called “DNA-PKcs Ab-4 (Cocktail)”, designed for a sensitive detection of the kinase and recommended from the manufacturer for different applications such as Western blot, immunoprecipitation and immunofluorescence (Levy N et al., 2006 Nucleic Acids Res. 2006 Jan 5;34(1):32-41. doi: 10.1093/nar/gkj409; Vemuri MC et al., 2001 Cell Death Differ 8, 245-255. doi: 10.1038/sj.cdd.4400806).

When we use this antibody in IF on proliferating cells, including human neural progenitor cells (hNPCs) and human embryonic kidney 293 (HEK293) cells (Supplementary Fig.1), the Ab-4 gives a strong labelling of nuclei thus recognizing a strong nuclear localization of DNA-PKcs in these cells. Although we noticed a certain variability of DNA-PKcs nuclear intensity in post-mitotic neurons, the nuclear staining is not as strong as in proliferating cells (Supplementary Fig. 1). In addition, we now show a new image of neurons with different intensity of DNA-PKcs nuclear staining (Figure 1c, upper panel and Supplementary Fig. 1). Regarding the discrepancy in nuclear detection between Western blot and IF in the previous version of the manuscript, thanks to the Referees, we now evaluate the DNA-PKcs nuclear amount by performing the fractionation experiments using pure neuronal cultures (primary cortical neurons) instead of cortical tissue. Indeed, as shown in Figure 1b, in primary neurons the percent of nuclear DNA-PKcs is about 20% of the starting total protein extract indicating a relative low nuclear abundance of DNA-PKcs in post mitotic neurons (see Results Section, “Characterization of DNA-PKcs in brain and neurons” paragraph). Overall, we are confident in our data and we do exclude possible unspecific labelling of the DNA-PKcs Ab-4 antibody used.

Comment 2. In Figure 1d, higher resolution images with more examples of co-localization are needed, especially since the colocalization of DNA-PKcs with synaptic markers appears low. It is also unclear whether the quantification of colocalization includes the soma signals or only at the synaptic level.

Answer: We have performed co-localization immunofluorescence experiments in DIV 21 primary cortical neurons with two presynaptic proteins (Syntaxin I and Synapsin I) and three postsynaptic markers (GluA1; GluN1 and PSD-95) along with DNA-PKcs and MAP2 (for neurite labelling) antibodies to show similar pattern in neurites and colocalization (Figure 1d). Moreover, we have acquired the new images at high resolution with THUNDER Imager 3D Live Cell & 3D Cell Culture microscope (Leica) with innovative technology of computational clearing, as specified in Results Section, “Characterization of DNA-PKcs in brain and neurons” paragraph.

The quantification of co-localization did not include the soma signals but was carried out in neurites. We have further specified this in the Material and Methods Section.

Comment 3. Fig 1e, a negative control is needed.

Answer: We had these data and modified the histogram adding this control as requested in Figure 1e.

Comment 4. Figure 1f shows an increase in DNA-PKcs staining in the soma and proximal dendrites. It would be good to confirm this is the case with Western blot of whole cell lysates, as Fig. 1g implies the increase is specific to synapses.

Answer: We agree with the referee and we performed experiments on primary cortical neurons by stimulating cells with F/R and carried out WB analysis on total cell extract to better show that the enrichment of DNA-PKcs protein following stimulation does not occur in the whole cell but is specific for synaptic compartment (Figure 2b). We specified in the text “Interestingly, WB analyses showed no increase of DNA-PKcs protein on total neuronal extracts following F/R stimulation (Fig. 2b)”, Result Section, “DNA-PKcs protein increases in synaptosomal membranes in response to synaptic activity and interacts with postsynaptic proteins” paragraph.

Comment 5. Fig 1i: a non-specific antibody control (e.g. mouse IgG or anti-GFP antibody) is needed for the co-IP of synaptic proteins.

Answer: We repeated IP experiments and run the samples on precast gradient gels showing Input, IP and IgG control antibody isotype lanes for all the proteins tested (Figure 2d and e).

Comment 6. Figure 3b: DIV9 is too early to properly examine synapses. Higher resolution images at a later time point are needed to determine the effects on synapses.

Answer: We performed IF and show images of DIV 21 primary cortical neurons and acquired images at high resolution (Figure 4b and c).

Comment 7. Similarly, Figure 3c needs much higher resolution images to properly evaluate GluA1 puncta.

Answer: We performed IF and show images of DIV 21 primary cortical neurons and acquired images at high resolution to evaluate GluA1 and PSD-95 puncta (Figure 4b and c)..

Comment 8. Figure 4d and e: Something is off with the spine quantification as it shows the WT has just over 1 spine per 100 μm , which is way too low. Was the spine quantification also done at DIV9? If so the authors need to examine more mature neurons. Higher resolution images are needed.

Answer: We thank Referee 1 for this comment. Regarding the spine quantification, there was a mistake since we reported on the graph “protrusion per unit (50 μm)” meaning that 50 micrometers were the interval dendrite analyzed but the calculation was per micrometer. We have now modified text and Figure 5d as follows: Number of protrusions/ μm . Moreover, we repeated the experiments keeping in culture neurons for 21 DIV and acquired spine images at high resolution.

Comment 9. An in vitro assay using purified proteins is needed to demonstrate that DNA-PKcs is the kinase directly phosphorylating PSD-95.

Answer: The in vitro assay using recombinant proteins was performed and it was shown in Figure 5b (now Figure 6b) and clearly demonstrate that DNA-PKcs phosphorylates PSD-95 in vitro directly. However, thanks also to Referee 2 comment, we performed an in vitro kinase assay using immune-purified DNA-PKcs from mouse cortex LP1 and human recombinant PSD-95 protein. WB in Figure 7c shows the two bands specifically recognized by the phospho-specific antibodies pS308 and pT87 only when the kinase and PSD-95 are incubated with ATP (lane 4) thus confirming that DNA-PKcs from synaptosomal membranes phosphorylates PSD-95 at S308 and T87 residues. We added this result in the text Results Section, “Phosphorylation at S308 and T87 of PSD-95 is required for its protein stability” paragraph.

Comment 10. The quality of the Western blots in Figure 7d makes it difficult to evaluate the effects of the phosphorylation mutants. In several cases the bands show a lower intensity at an earlier time point than a later time point with longer CHX incubation. This does not make sense and raises the question as to whether the differences are just due to variability in transfection efficiency. Better quality blots from more standardized transfections are needed to evaluate the role of the phosphorylation mutants.

Answer: We carried out new CHX experiments and performed new WB to show high quality images (Figure 7e, Figure 8d) confirming our previous results.

Comment 11. Figure 8b: a synaptic marker is needed to ensure the synaptic localization of the mutants.

Answer: The synaptic localization of mutants was demonstrated by WB in Figure 8a (now Figure 9a) where PSD-95 mutants are enriched in LP1 fraction (synaptosomal membranes). However, we acquired new images with the postsynaptic marker Homer 1 and the over-expressed mutants at high resolution (Figure 9b).

Comment 12. Figure 8c: MAP2 is not a reliable way to visualize spines. Typically, MAP2 mostly stays in the dendritic shaft. The authors need to use a cell-fill like GFP.

Answer: We agree with Referee 1 and repeated the experiments on cortical neurons (DIV 21) from DNA-PKcs $-/-$ mice infected with the empty vector or the PSD-95 wt, or S308E or T87E proteins (Figure 9c). Neurons were further infected with EGFP-rAAV virus to visualize spine morphology, and then stained with the anti-MAP2 antibody (red channel) and DNA dye (blue channel). We specified that we have double-infected neurons to better visualize spines in Methods Section and in Legend to Figure 9.

Comment 13. Figure 8d: similar to Figure 4, something is off with the spine quantification here. The spine density is too low. The authors need to examine spines in more mature neurons and double check the data to ensure accurate quantifications.

Answer: See above the answer to point 8.

Comment 14. Figure 9b: it appears that the authors used EGFP as a marker for Sholl quantification for the control while using MAP2 as a marker for the mutant condition. They need to use a consistent marker for all experimental conditions.

Answer: We had previously performed and now we show in Figure 10b images with neurons labelled with anti-MAP2 antibody for both EGFP and T87 infected cultures. MAP2 was used as marker for Sholl quantification as indicated in text Result Section, “Rescue of PSD95 protein stability, spine and neuronal morphology in DNA-PKcs $-/-$ neurons” paragraph.

Comment 15. Figure 9d: control images are needed to demonstrate similar AAV-mediated expression of S308E vs T87E mutants.

Answer: We verified the issue raised by Referee 1 on hippocampal slice extracts from infected mice and we show the quantitative Western Blot in Figure 10e. We added in the text Result Section, “Synaptic potentiation in DNA-PKcs $-/-$ mice is improved by over-expression of PSD-95T87E mutant” paragraph the following sentence: “some hippocampal slices from infected mice (n = 3 for each experimental group) were analyzed by quantitative WB to demonstrate similar rAAV-mediated expression of the different flag proteins. As shown in Fig. 10e, T87E, S308E and PSD-95 wt proteins are expressed at comparable levels in hippocampal regions at 8 weeks post-injection”.

Minor points:

Comment 1. In Figure 1b, the lanes LP2 and LS2 are swapped between DNA-PKcs and the fractionation controls blots. It would be better if the lane sequences are consistent to avoid confusion.

Answer: We performed new fractionation experiments by using primary cortical neurons (see Answer to Comment 1) and run the corresponding gels with the correct lane sequence (Figure 1b).

Referee #2:

Specific comments:

Comment 1. The localization of PKcs at the synapse is not convincing. First the antibody specificity in staining has to be verified by comparing the staining between wild-type and KO neurons. Indeed, it is worrying that in Fig. 1c the nuclei appear to lack PKcs, which I suppose should not be the case if it is a protein for DNA repair and is much enriched in P1 (Fig. 1b), suggesting that the antibody is not specific for staining. It is also important to verify the synaptic localization by exogenously expressing a GFP- or other tagged PKcs in neurons. In addition, in the fractionation experiments the band intensity in LP1 is very weak, indicating the proportion at the synapse is modest. The authors must include the crude homogenate in the Western blot in order to give an idea about the percentage of PKcs in the synaptic membrane.

Answer: As for point 1 of Referee 1:

The DNA-PKcs antibody specificity is confirmed by the absence of immunofluorescence (IF) staining following antibody neutralization by incubation with an excess of recombinant DNA-PKcs protein as shown in Figure 1c, lower panel. This result had already been specified in the Legend to Figure 1 of the previous version as follows: “DNA-PKcs antibody specificity is confirmed by the absence of immunofluorescence when antibodies are pre-incubated with an excess of recombinant DNA-PKcs protein (data not shown)”. Moreover, as suggested by Referee 1, we now show DNA-PKcs antibody specificity by the absence of IF staining on primary cortical and hippocampal cultures (DIV 21) of knockout mice (Figure 3b). Furthermore, in Figure 3a (lane DNA-PKcs^{-/-} hippocampus, previously shown) Western Blot (WB) analysis on hippocampal protein extract from knockout mice confirms the specificity of the used antibody in WB by the lack of DNA-PKcs immunoreactive band.

Please note that the antibody we used is a mixture of 4 different monoclonal antibodies called “DNA-PKcs Ab-4 (Cocktail)”, designed for a sensitive detection of the kinase and recommended from the manufacturer for different applications such as Western blot, immunoprecipitation and immunofluorescence (Levy N et al., 2006 Nucleic Acids Res. 2006 Jan 5;34(1):32-41. doi: 10.1093/nar/gkj409; Vemuri MC et al., 2001 Cell Death Differ 8, 245-255. doi: 10.1038/sj.cdd.4400806). When we use this antibody in IF on proliferating cells, including human neural progenitor cells (hNPCs) and human embryonic kidney 293 (HEK293) cells (Supplementary Fig.1), the Ab-4 gives a strong labelling of nuclei thus recognizing a strong nuclear localization of DNA-PKcs in these cells. Although we noticed a certain variability of DNA-PKcs nuclear intensity in post-mitotic neurons, the nuclear staining is not as strong as in proliferating cells (Supplementary Fig. 1). In addition, we now show a new image of neurons with different intensity of DNA-PKcs nuclear staining (Figure 1c, upper panel and Supplementary Fig. 1).

Regarding the discrepancy in nuclear detection between Western blot and IF in the previous version of the manuscript, thanks to the Referees, we now evaluate the DNA-PKcs nuclear amount by performing the fractionation experiments using pure neuronal cultures (primary cortical neurons) instead of cortical tissue. Indeed, as shown in Figure 1b, in primary neurons the percent of nuclear DNA-PKcs is about 20% of the starting total protein extract indicating a relative low nuclear abundance of DNA-PKcs in post mitotic neurons (see Results Section, “Characterization of DNA-PKcs in brain and neurons” paragraph). Overall, we are confident in our data and we do exclude possible unspecific labelling of the DNA-PKcs Ab-4 antibody used.

We agree with the Referee 2 that the synaptic localization of DNA-PKcs could have been verified by exogenously expressing a GFP- or other tagged PKcs in neurons. Indeed, we have asked and received, as kind gift, the plasmid with human Flag-DNA-PKcs cDNA cloned into the mammalian expression vector pCMV6 from Prof. Katheryn MeeK as described in Shin et al., J. Immunol. 2000, 164:1416-1424. Although we could overexpress DNA-PKcs protein in cell lines, we had difficulties to obtain detectable levels in primary neurons and, to our knowledge, overexpression of DNA-PKcs (MW: 450KDa) in primary cultured neuron has never been reported. Therefore, we acquired high resolution images to better show the synaptic localization of DNA-PKcs. Moreover, we exploit DNA-PKcs $-/-$ mice to unveil the possible role of this repair protein at synapses.

As requested by Referee2, we have included a total protein extract from cortical neurons in the WB shown in Figure 1b, in order to calculate the percentage of DNA-PKcs in the subcellular compartments. We have added in the text, Results Section, “Characterization of DNA-PKcs in brain and neurons” paragraph the following sentence: “Compared to the amount of DNA-PKcs in the starting total protein extract, we estimated the proportion of the kinase at the different subcellular compartments and found that in primary neurons DNA-PKcs is present in nuclei (P1, $21.57 \pm 4.4\%$), light membrane compartment (P3, $31.65 \pm 4.8\%$) and cytosol (S3, 29.13 ± 6). Interestingly, DNA-PKcs is present in synaptosomal membrane fraction (LP1, $13.28 \pm 1.45\%$) and poorly detected in synaptic vesicle-enriched fraction (LP2, $5.43 \pm 0.93\%$), but absent from soluble subcellular fraction (LS2) (Fig. 1b)”.

Comment 2. The authors used 9 DIV cultured cortical neurons to examine neuronal morphology related to synaptic plasticity, but this stage is too young to form mature synapses (which is indicated by the appearance of mostly filopodia or thin spines instead of mushroom-shaped spines). Without the mushroom spines it is not clear if the PKcs co-localized with PSD-95 are really synaptic. The use of cortical neurons also makes it difficult to link the cellular data with the electrophysiological data from hippocampal slices of much older mice. In general, neurons of around 21 DIV are used for this kind of analysis.

Answer: As for point 6 and 8 of Referee 1:

We repeated the experiments keeping in culture neurons for 21 DIV and performed IF and acquired spine images at high resolution by using THUNDER Imager 3D Live Cell & 3D Cell Culture microscope (Leica) with innovative technology of computational clearing.

The use of cortical primary neurons was due to several limiting conditions:

The cell density of neurons in a 10mm dish is about 5×10^6 cells. To perform the experiment shown in Figure 2c (Forskolin/Rolipram stimulation followed by fractionation) we used four 10 mm dishes for each experimental point (DMSO and F/R) for a total of eight dishes. These culture preparations required the use of a very high number of embryos such as 30-35 and the euthanasia of 4-5 WT pregnant females. Moreover, the number of embryos for a DNA-PKcs $-/-$ female is lower than the WT, thus requiring the euthanasia of about 6-7 pregnant females. Performing the same experiments with hippocampal primary cultures would have required four times more embryos for a total of at least 16 WT pregnant females and 24 knockout females per experiment. The three R principle is very much felt in our Institute and we tried to reduce as much as possible the number of animals euthanized, thus choosing the cortex. However, for analysis of phosphorylation levels of proteins after electrophysiology recordings, the recorded hippocampal slice was rapidly dissected to isolate the CA1 region and used for biochemistry to better link the electrophysiological data with protein phosphorylation. Moreover, in Figure 4a we now show the postsynaptic proteins levels in both hippocampus and cortex of WT and DNA-PKcs $-/-$ mice. Furthermore we performed IF on primary hippocampal neurons in Supplementary Fig. 1 and Figure 3b (DNA-PKcs antibody specificity). We also verified rAAV-mediated expression levels of the different flag mutant proteins by quantitative WB in hippocampal regions of mice 8 weeks post-injection (Figure 10e).

Comment 3. Related to 2, the use of young neurons makes it hard to judge the staining data in Fig. 1 and 3. There appear to be lot of puncta outside the neurons which seem to be background but not real signals. It is important to co-stain the neurons with MAP2 (or GFP if the neurons are infected) which outline the dendrites for quantifying the puncta, thus avoiding the non-specific background staining. The image quantification is also poorly described. For figure 3b, c, is it the number, area or intensity that the authors measured? How did they determine the co-localization (by Manders coefficient using software or were they manually quantified)? Were the images put on a threshold to define the puncta or they were defined arbitrarily? Were the quantification performed blind? This essential information in image analysis is entirely missing in the manuscript.

Answer: As for point 6 and 8 of Referee 1:

We have repeated the experiments keeping in culture neurons for longer time in vitro (21 days) and acquired images at high resolution.

We have performed triple labelling immunofluorescence on DIV 21 neurons, adding MAP2 antibody (Figure 1d, Figure 4b and c).

For figure 4b and c we measured mean intensity of fluorescence. We determined the co-localization using NIH Image software with the ImageJ129 plugin kindly provided by dr. Cagla Eroglu Durham, USA (http://labs.cellbio.duke.edu/Eroglu/Eroglu_Lab/Publications.html) (Ippolito and Eroglu, *J Visualized Experiments* 2010; Risher et al., *eLife* 2014). The Puncta Analyzer plugin uses an algorithm to detect the number of puncta that are in close alignment across the two channels, yielding quantified co-localized puncta. Puncta were quantified by Puncta Analyzer on maximum projection 63x magnification images separated into red and green channels, background subtracted (rolling ball radius = 50), and thresholded in order to detect discrete puncta without introducing noise. Equally thresholding was applied across acquired images.

The experiments were performed blind by two different researchers.

We have added more detailed information on image acquisition as required by Referee 2 in Material and Methods section.

Comment 4. Fig. 1 h and i: it is essential to include IgG control in the co-IP experiments (or even better, use the KO samples as the negative control).

Answer: As for point 5 of Referee 1:

We repeated IP experiments and run the samples on precast gradient gels showing Input, IP and IgG control antibody isotype lanes for the all proteins tested (Figure 2d and e).

Comment 5. Fig. 3a: It is important to include the crude homogenate in order to determine if the total expression of those proteins is reduced, or only the abundance at the synaptic membrane is affected in the KO samples. In addition, does the hippocampal slice from KO mice also show reduced PSD-95 expression compared to wild-type? This is important because later the authors show that expressing the phospho-mimetic PSD-95 can rescue the LTP.

Answer: We agree with Referee 2 and in Figure 4a we evaluated the levels of postsynaptic proteins in both total extracts and LP1 fractions from cortex and hippocampus of WT and DNA-PKcs $-/-$ mice by WB analysis. We found no differences in expression levels of any postsynaptic proteins analyzed between WT and DNA-PKcs $-/-$ total extracts. Interestingly, in LP1 from both cortical and hippocampal tissues we found a significant decrease in PSD-95 levels in DNA-PKcs $-/-$ mice compared to WT mice. These results are described in Results Section, “DNA-PKcs $-/-$ mice show impairment in synaptic transmission, Long-Term Potentiation, and postsynaptic protein expression” paragraph.

Comment 6. Fig. 4a: The cell densities appear much lower in the KO neurons. Does the gene knockout affect neuronal survival? Can the observed differences in neuronal morphology be caused indirectly by the different cell density instead of direct function of the gene?

Answer: We agree with Referee 2 that in Figure 4a (now Figure 5a) the cell density in KO neurons appears lower and we have now showed images of fields of culture at both 9 and 21 DIV with comparable number of neurons suggesting that the observed differences in neuronal morphology of knockout mice is not caused indirectly by the cell density.

Comment 7. Fig. 6: it is important to add the in vitro kinase assay using the two phospho-specific antibodies to validate purified PKCs indeed phosphorylates the recombinant PSD-95. Fig. 6c: the authors must use total PSD-95 instead of tubulin as the control to demonstrate the induction of phosphorylation. Fig. 6d: the tubulin bands for WT are clearly decreased in the 24 and 48 hr time points, raising the possibility that the two samples were over-loaded and masked the decrease in PSD-95 expression at these later time points.

Answer: We thank Referee 2 for this interesting suggestion. We then performed an in vitro kinase assay using immune-purified DNA-PKCs from mouse cortex LP1 and human recombinant PSD-95 protein. WB in Figure 7c show the two bands specifically recognized by the phospho-specific antibodies pS308 and pT87 only when the kinase and PSD-95 are incubated with ATP (lane 4) thus confirming that DNA-PKCs from synaptosomal membranes phosphorylates PSD-95 at S308 and T87 residues. We added this result in the text Results Section, “Phosphorylation at S308 and T87 of PSD-95 is required for its protein stability” paragraph.

It is known from the literature (Skibinska et al., Neuroreport 2001; Bao et al., Nature Neuroscience 2004; Ifrim et al., J Neuroscience 2015; Sun et al., Science Advances 2021) that, following neuronal activity, an increase of newly locally synthesized PSD-95 protein is detected using different techniques. Similarly, in our experiments we observed an increased PSD-95 total protein in CA1 hippocampal slices after HFS ($43 \pm 7.6\%$ and $62 \pm 4.6\%$ 5 min and 15 min respectively following tetanization (see Results Section, “Phosphorylation at S308 and T87 of PSD-95 is required for its protein stability” paragraph), which parallels its phosphorylation. When using PSD-95 for normalization, LTP induces a significant increase of PSD-95 phosphorylation at both S308 (5 min: $132 \pm 4\%$; $p < 0.05$ vs T0; 15 min: $124 \pm 4\%$; $p < 0.05$ vs T0) and T87 (5 min: $136 \pm 1.73\%$; $p < 0.005$ vs T0; 15 min: $119 \pm 4.62\%$; $p < 0.05$ vs T0) (Figure 7d). However, when using the housekeeping protein alpha-tubulin for normalization, we show that, compared with non-tetanized control slices, LTP induces a higher increase of PSD-95 phosphorylation at both S308 and T87 that persisted for at least 15 min (5 min S308: $152 \pm 7.2\%$; $p < 0.05$ vs T0; 15 min S308: $184 \pm 6.5\%$; $p < 0.005$ vs T0; 5 min T87: $163.31 \pm 6.83\%$; $p < 0.005$ vs T0; 15 min T87: $170 \pm 5\%$; $p < 0.005$ vs T0) (Figure 7d). We have added these data in the Results Section, “Phosphorylation at S308 and T87 of PSD-95 is required for its protein stability” paragraph.

These results suggest that, following synaptic activity the newly synthesized PSD-95 protein undergoes phosphorylation by DNA-PKcs to guarantee its stability at synapses after stimulus.

Comment 8. Fig. 7: It is hard to understand why the stability of the SE and SA mutant are compared in different neurons. Since the Ser site has been replaced by either E or A, it can no longer be phosphorylated by PKcs and thus they should all be expressed in wild-type neurons for fair comparison. On the other hand, expressing the two sets of mutants in different neurons is problematic because one cannot rule out the possibility that the protein degradation machinery is altered in the KO neurons. Fig. 7d: the band for T87E at 24 hr is clearly down-regulated compared to T0 but the quantification indicated no difference.

Answer: We have performed new experiments and infected WT primary cortical neurons with T87E, T87A and S308E, S308A - rAAV and performed CHX experiments and WB analysis using Flag antibody to compare the stability of the four mutants in WT neurons. The results are shown in Figure 8d lower panels and described in the Result Section “Rescue of PSD95 protein stability, spine and neuronal morphology in DNA-PKcs $-/-$ neurons” paragraph.

Comment 9. Fig. 8 and 9: the conclusion about the importance of PSD-95 is not convincing. To show that PSD-95 phosphorylation is the major downstream event of PKcs in regulating those processes, the appropriate way should be comparing between the SE mutant and the WT in rescuing the phenotypes. Just expressing SE mutant in the rescue experiments does not lead to the conclusion that the PSD-95 phosphorylation is important. It is possible that expressing the WT or even the SA mutant can also rescue the phenotypes.

Answer: We do not believe that any PSD-95 protein wt or mutant can rescue the phenotypes because we show that just the mutant T87E and not S308E can rescue neurite network, spine morphology and, importantly, the LTP thus excluding the possibility that any PSD-95 is able to restore plasticity. However, we have performed new experiments using the rAAV-PSD-95 wt Flag protein (Figure: 8a, b, c) both in vitro (9 and 21 DIV) and in vivo. First, we found that PSD-95 wt significantly decreases 24 and 48 h after CHX treatment when over-expressed in DNA-PKcs $-/-$ neurons (Supplementary Fig. 3). Consequently, as shown in Figure 9c and Figure 10 b, c, d, e PSD-95 wt protein, when over-expressed in DNA-PKcs $-/-$, is not able to rescue the phenotypes neither in culture nor in vivo thus demonstrating that PSD-95 phosphorylation by DNA-PKcs is necessary for its stability and, in turn, for synaptic plasticity.

Comment 10. The format of writing throughout the manuscript (especially the introduction) is strange because there are too many separate sentences, like putting together different bullet points.

Answer: We have extensively modified the text with the help of a mother tongue.

DIPARTIMENTO
NEUROSCIENZE

We thank Referees for their criticisms and suggestions and hope that our revisions are satisfactory for publishing in EMBO Reports.

Sincerely,

Daniela Merlo, PhD

Dear Daniela,

Thank you for the submission of your revised manuscript to EMBO reports. We have now received the comments from both referees, as well as cross-comments and additional comments from a third advisor/referee, all pasted below and attached to this email.

As you will see, referee 1 is not satisfied with the revised manuscript and its technical quality and is asking for more and better data, controls and explanations. Referee 2 and 3 agree with most of these comments, and they thus need to be addressed, as we discussed. Please co-submit with your final ms a detailed point-by-point response that addresses all final referee comments. It is important that all controls will be provided and that the issue with the error bars is clarified and corrected.

A few editorial requests will also need to be addressed:

- The figure files need to comply with EMBO reports requirements. You can find all information in our guide to authors online. All figures need to be in portrait format and fit on one page and the figure resolution needs to be sufficiently high.
- Please reduce the number of keywords to 5.
- Please correct the conflict of interest subheading to "Disclosure and Competing Interests Statement"
- Please resolve the discrepancy in the way authors are listed in the ms file (last name, first name initial) and in our online submission system (first name last name).
- The authors credits need to be removed from the ms file. All credits need to be entered during online ms submission.
- The REFERENCE FORMAT needs to be corrected to EMBO reports style. It needs to be alphabetical, et al needs to be used after 10 author names.
- DATA NOT SHOWN is not allowed per journal policy. Please remove/rewrite or show the data.
- Please co-submit a completed author checklist, which you can download from our author guidelines <<https://www.embopress.org/page/journal/14693178/authorguide>>. The completed author checklist will also be part of the transparent peer-review process file.
- The FUNDING INFO needs to be part of the Acknowledgement section, please correct.
- There is one Dataset uploaded but it needs the correct nomenclature: Dataset EV1 - the callout in the ms also needs to be added.
- The APPENDIX file needs a table of content with page numbers. The nomenclature needs to be corrected: Supplementary Information should be Appendix, the figures should be Appendix Figure S1, etc.; the callouts in the ms also need to be corrected accordingly; Supplementary References should be Appendix References; the Supplementary Materials and Methods should be Appendix Materials and Methods.
- The manuscript sections should be in the following order: Title page - Abstract & Keywords - Introduction - Results - Discussion - Materials & Methods - Data Availability - Acknowledgments - Disclosure Statement & Competing Interests - References - Figure Legends
- "Supplementary information" needs to be removed from the ms file.
- Please note that information related to n is missing in the legends of figures 1d-e; 2g. Please add.
- Although 'n' is provided, please describe the nature of entity for 'n' in the legends of figures 2a, c, h; 4b-d; 7d; 8d.
- Please note that the error bars are not defined in the legends of figures 1b, d; 2c-e; 4b-c; 5c; 10d. Please add.

I would like to suggest a few minor changes to the abstract. Please let me know whether you agree with the following:

The key DNA repair enzyme DNA-PKcs has several and important cellular functions. Loss of DNA-PKcs activity in mice has revealed essential roles in immune and nervous systems. In humans, DNA-PKcs is a critical factor for brain development and function since mutation of the *prkdc* gene causes neurological deficits such as microcephaly and seizures, predicting yet

unknown roles of DNA-PKcs in neurons. Here we show that DNA-PKcs modulates synaptic plasticity. We demonstrate that DNA-PKcs localizes at synapses and phosphorylates PSD-95 at newly identified residues controlling PSD-95 protein stability. DNA-PKcs $-/-$ mice are characterized by impaired Long-Term Potentiation (LTP), changes in neuronal morphology, and reduced levels of postsynaptic proteins. A PSD-95 mutant that is constitutively phosphorylated rescues LTP impairment when over-expressed in DNA-PKcs $-/-$ mice. Our study identifies an emergent physiological function of DNA-PKcs in regulating neuronal plasticity, beyond genome stability.

EMBO press papers are accompanied online by A) a short (1-2 sentences) summary of the findings and their significance, B) 2-3 bullet points highlighting key results and C) a synopsis image that is exactly 550 pixels wide and 200-600 pixels high (the height is variable). You can either show a model or key data in the synopsis image. Please note that text needs to be readable at the final size. Please send us this information along with the final manuscript.

I look forward to seeing a final version of your manuscript when it is ready.

Referee #1:

This revised manuscript examines the synaptic role of DNA-PKcs. The authors have addressed some of the concerns of the previous reviews, including providing higher resolution images, better quality Western blots, and proper controls for some experiments. While these revisions have improved the quality of the manuscript, the data remain somewhat preliminary and do not provide enough of a mechanistic advance to be in a high profile publication like the EMBO Reports. In addition, significant concerns remain with the data quality of the manuscript. Specific concerns are listed below:

1. The synaptic localization of DNA-PKcs shown in Fig 1 is not very convincing. Single channel images need to be provided to better evaluate the localization of each protein. Ideally randomization needs to be performed to ensure the colocalization exceeds random coincidence.
2. It is interesting that immunohistochemistry shows a significant increase in DNA-PKcs intensity in the soma and proximal dendrites in Fig 2a but no change in total protein levels in Fig 2b. The authors conclude that the changes were specific to synaptic membrane. However this does not explain why the staining intensity in the soma and proximal dendrites were increased. Furthermore, in Fig 2b DNA-PKcs shows a lower band which appears increased. This is also observed in the P3 fraction in Fig 2c. What does this band represent? This needs to be better quantified and the results need to be discussed.
3. In Figure 3, more details are needed on how the Western blots are normalized. The authors mention it was normalized to the T0 time point but there is no mention of whether the phosphorylated protein was normalized to the total protein. This is important since all of the representative blots show an increase in both the phosphorylation levels and the total levels of CAMKII, ERK and Akt.
4. Some of the images remain of poor resolution and quality. For example, in Fig 4b, only one neuron in the DNA-PKcs $-/-$ condition shows positive MAP2 staining, even though there are multiple cells in the field and weak punctate staining resembling neurites are observed for those cells. This suggests the possibility that there are significant issues with the health of the neurons in the culture, which may be causing the reduced synaptic staining.
5. In Fig 4d, the authors show that GluA1 delivery to synapses remain normal after LTP induction in the DNA-PKcs $-/-$ mice. However, this was done with a 10 min F/R treatment. In their fEPSP data, it seems that these mice have problems with LTP maintenance at later time points but not induction. It would be helpful to look at GluA1 trafficking at the synapses in later LTP maintenance to see whether there are any defects in the KO mice.
6. Also in Fig. 4d, it is odd that LTP treatment appears to decrease the levels of synaptic PSD-95. This should be quantified and discussed.
7. The Flag staining for PSD-95 constructs look odd. PSD-95 localizes strongly to synapses even when overexpressed. However in Fig 8c the FLAG constructs do not show synaptic localization. Adding to this concern is Fig 9c, where FLAG tagged PSD-95 constructs show little colocalization with Homer. This raises concerns over the specificity of the staining.
8. The differential effects of S308 vs T87 phosphorylation on dendritic complexity and LTP are not explained. If both sites enhance PSD-95 stability, why is S308E able to restore dendritic spine density but not dendritic complexity or LTP?
9. The relationship between the S308 and T87 phospho-sites on PSD-95 stability is also unclear. It appears that phosphorylation of either site alone is sufficient to enhance PSD-95 stability, while dephosphorylation of either site is also sufficient to destabilize PSD-95. This does not make much sense. What happens to PSD-95 stability if there is a T87E, S308A double mutation, or a T87A, S308E double mutation?
10. In Fig 10d, it would be important to compare the data with WT in terms of the level of LTP induction and maintenance. In addition, the ability for overexpressed PSD-95 constructs to rescue LTP is interesting considering that previous studies showed

overexpression of PSD-95 precludes LTP induction. This should be discussed.

11. The error bars on many of the graphs look very odd. The authors report that the error bars represent SEM (graphs show mean {plus minus} SEM). However, many of errors bars (see Fig 2c, 3h, 4a, 4c, 7d) are oddly positioned with the center of the error bar not at the mean. The positions of these error bars were also not consistent from bar to bar. This raises concerns about the possibility that these error bars were not generated automatically by the statistical software but rather manually added onto the graph later.

12. Related to the above issue, data reporting needs to be more transparent with individual data points shown on all bar graphs, similar to what they did in Fig 3e, g, and Fig 10d for their electrophysiology data.

Referee #2:

The authors have extensively revised the manuscript by adding the new experiments to address my comments. I recommend publication of the manuscript.

Cross-comments from referee 2:

I read reviewer #1's comments and agree some of them. However, I think the comment that the study lacks mechanistic insights is too harsh, as the authors have already shown the kinase can phosphorylate PSD-95, the phospho-deficient PSD-95 have lower stability, and the phospho-mimetic PSD-95 can rescue the phenotypes of the DNA-PKcs knockout neurons. These to me are quite convincing evidence pointing towards PSD-95 as a major downstream target of the kinase in regulating synaptic formation and function.

I do agree with the reviewer that the manuscript can be improved by adding single channel images (point 1); clarify and quantify the soma intensity difference in Fig 2a and discuss the extra band in the WB of Fig. 2b (point 2); explain why the exogenous PSD-95 seem diffuse and did not co-localize with Homer (point 7); clarify the strange error bars and make sure they are not manually added (point 11); and change the bar graphs by adding the individual data points (point 12). For point 4, the authors can be suggested to quantify the cell number between WT and KO neurons. The other comments are mainly asking for discussion/explanation from the authors.

Referee #3/Advisor:

Paper Review for Revised MS by Mollinari et al.: "The DNA repair protein DNA-PK-cs..." for EMBO Reports

To the Authors:

Many PI3KK family members are protein kinases, such as ATR, ATM, DNA-PK-cs (and also SMG1), mutations of which (excluding SMG1) cause chromosomal instability syndromes, ATR-Seckel, A-T and SCID, respectively. Their functions have been well studied for their key roles in DNA damage response (DDR) and DNA repair. Paradoxically, the human patients, and mouse models carrying mutations of these gene mutations, often exhibit phenotypes (e.g., neural tissues or postmitotic cells) which cannot be explained by the possible function in the nucleus, namely DDR. There is emerging evidence showing wide distribution and cytoplasmic function of these PIKK proteins in the various subcellular compartments, not only in the nucleus, raising other unknown yet important function in pathogenesis. This manuscript reported a novel function of DNA-PKcs in phosphorylation of PSD95 protein in response to synaptic activation of neurons (postmitotic). The information is potentially interesting. However, there were many flaws in the manuscript needing to be addressed.

I had chance to read the reviewer #1's comments. Below I only list my comments in addition to the comments of Reviewer #1.

1. Generally, the data presentation and discussion should be improved, also as suggested by Reviewer #1.
2. DNA-PKcs^{-/-} mice originally was designed for a kinase dead mutant (DNA-PK-KD). Unfortunately, due to the protein stability (the truncation mutant is often unstable), it behaves like a knockout (KO, also confirmed in Fig 3a). The whole manuscript is based on this mouse model and assumed the study dealing with only DNA-PK kinase activity. However, without DNA-PK KO control, one cannot rule out the contribution by its scaffold, like the case of ATR and ATM, in other studies.
3. Foscilin+Rolipram (R/F) is known to stimulate synaptic activity in LTP. One basal line of this treatment should be documented in order to show the treatment worked.
4. Fig 1d. Due to the resolution limitation, one cannot view clear localization pattern as the authors described, "periphery" or "overlapping". DNA-PKcs^{-/-} control is needed here.
5. Fig 3h. It is odd/interesting that DNA-PKcs^{-/-} neurons show high levels of pS473-AKT. This residue has been shown to be a phosphorylation site by DNA-PK, and perhaps also by ATM or ATR. What is going on here? Thus, ATM or ATR inhibitors can be used to test the substrate specificity of DNA-PK. In this regard, DNA-PK, ATR and ATM share many substrates and it would be interesting to examine whether the phosphorylation status of other proteins in the same figure as well as PSD95 when ATM/ATR are inhibited.
6. The authors tend to claim that phosphorylation of PSD95 by DNA-PK is F/R dependent. However, most of experiments, particularly in testing phosphor mutant PSD95, in Fig 7e, 8d, 9 and 10 did not use F/R treatment.
7. The description or comparisons of quantifications of some figures in the text disturbs reading; thus the details can move to the figure legends.
8. Minor. For example, NHEJ should be non-homologous end joining, not "not".

To the Authors:

The manuscript provides interesting information, which potentially could be published. However, the quality of presentation and analyses of the data seem to be rough, even after one round of revision. I read the Reviewer#1 comments on this revised version, I largely agree with her/his comments (many points), which should be addressed before acceptance. As the matter of fact, many of her/his comments are more technical, and should be “easily” addressed. If all my comments and Reviewer #1’s comments, can be addressed in a satisfactory manner, the quality of the manuscript in my view warrants publication by *EMBO Reports*.

May 8, 2024

Esther Schnapp
Senior Editor
EMBO Reports

Dear Editor,

Please enclosed you find a revised version of the manuscript entitled “The DNA repair protein DNA-PKcs modulates synaptic plasticity via PSD-95 phosphorylation and stability” by Mollinari C., Cardinale A., Lupacchini L., Martire A., Martinelli A., Chiodi V., Rinaldi A.M., Fini M., Pazzaglia S., Domenici M.R., Garaci E., Merlo D.

We have made the changes in accordance with the criticisms of the Referees.

Here we provide the point-by-point answers to the Referees as follows:

Referee #1:

Comment 1. The synaptic localization of DNA-PKcs shown in Fig 1 is not very convincing. Single channel images need to be provided to better evaluate the localization of each protein. Ideally randomization needs to be performed to ensure the colocalization exceeds random coincidence.

Answer: We provide single channel images as requested by Referee 1. In particular, we modified Figure 1D showing single channel images for colocalization of DNA-PKcs with synaptic markers in neurites and modified the Figure legend accordingly. Moreover, we provide Figure EV2 showing single channel images of the whole neuron for each synaptic protein analyzed, colocalizing with DNA-PKcs.

Comment 2. It is interesting that immunohistochemistry shows a significant increase in DNA-PKcs intensity in the soma and proximal dendrites in Fig 2a but no change in total protein levels in Fig 2b. The authors conclude that the changes were specific to synaptic membrane. However, this does not explain why the staining intensity in the soma and proximal dendrites were increased. Furthermore, in Fig 2b DNA-PKcs shows a lower band which appears increased. This is also observed in the P3 fraction in Fig 2c. What does this band represent? This needs to be better quantified and the results need to be discussed.

Answer: We thank Referee 1 for this observation and we agree that the image in Fig 2a was not fully representative. Indeed, the images in Fig 2a, provided in the previous version, were saturated leading to the wrong conclusion that DNA-PKcs increases in soma and neurites following F/R treatment. We now provide the same images with lower fluorescence intensity below saturation levels and in addition a new

image where it is more evident that the enrichment of DNA-PKcs protein, following F/R treatment, occurs mainly in neurites proximal to cell body (indicated by red arrows).

Moreover, according to Fig 1b, the percentage of DNA-PKcs in synaptic membranes is just approx. 13% of the total amount, thus its enrichment in LP1 fraction, following synaptic activity, could not allow a detectable increase in the total protein content, as shown in Fig 2b.

Regarding the lower band present in WB of Fig 2b and 2c, it is known from the literature, including the paper where our DNA-PKcs $-/-$ mouse model was generated (Taccioli et al., *Immunity*, 9 (1998) 3: 355-366), that a 220-250 kDa protein cross-reacting with certain DNA-PKcs antibodies is detectable. This low band has also been reported by other groups and does not appear to represent a proteolytic cleavage product of DNA-PKcs (Taccioli et al., *Immunity*, 9 (1998) 3: 355-366; Danska et al., *Mol. Cell. Biol*, 16 (1996), pp. 5507-5517; Peterson et al., *J. Biol. Chem*, 272 (1997), pp. 10227-10231; Priestley et al., *Nucleic Acid Res*, 26 (1998), pp. 1965-1973). Therefore, we believe that such lower band represents a not specific product. In agreement with these observations, in a more recent paper published in *EMBO J.* (Chen et al., (2023) 42: e112094), Western Blot using a different DNA-PKcs antibody (MA5-13238 from Invitrogen) also shows a lower band of about 250 kDa as shown below.

Figure 2D from Chen et al., 2023, *EMBO J.*

Comment 3. In Figure 3, more details are needed on how the Western blots are normalized. The authors mention it was normalized to the T0 time point but there is no mention of whether the phosphorylated protein was normalized to the total protein. This is important since all of the representative blots show an increase in both the phosphorylation levels and the total levels of CAMKII, ERK and Akt.

Answer: We have specified in the text, Results Section, “DNA-PKcs $-/-$ mice show impairment in synaptic transmission, Long-Term Potentiation, and postsynaptic protein expression” paragraph that the phosphorylation levels of the kinases analyzed are normalized to the their respective total protein.

Comment 4. Some of the images remain of poor resolution and quality. For example, in Fig 4b, only one neuron in the DNA-PKcs $-/-$ condition shows positive MAP2 staining, even though there are multiple cells

in the field and weak punctate staining resembling neurites are observed for those cells. This suggests the possibility that there are significant issues with the health of the neurons in the culture, which may be causing the reduced synaptic staining.

Answer: We have replaced in Fig 4b the panel showing one MAP2 positive cell with a new panel showing multiple MAP2 positive cells. Although DNA-PKcs $-/-$ neurons have a different morphology at both DIV 9 and DIV 21 as compared with WT neurons, showing a reduced dendrite complexity and spine number, DNA-PKcs $-/-$ cultures do not show differences in the number of apoptotic nuclei as compared with WT. We added a histogram representing the percentage of apoptotic nuclei in both DNA-PKcs $-/-$ and WT neurons indicating no significant differences between the two groups. The new figure is Fig. 5A. Moreover, we modified the text, Results Section, “Cortical neurons from DNA-PKcs $-/-$ mice show less spine and neurite complexity” paragraph, accordingly.

Comment 5. In Fig 4d, the authors show that GluA1 delivery to synapses remain normal after LTP induction in the DNA-PKcs $-/-$ mice. However, this was done with a 10 min F/R treatment. In their fEPSP data, it seems that these mice have problems with LTP maintenance at later time points but not induction. It would be helpful to look at GluA1 trafficking at the synapses in later LTP maintenance to see whether there are any defects in the KO mice.

Answer: We thank Referee 1 for the request of this experiment. Because cLTP was induced in primary cortical neurons by 10 min F/R treatment, we performed new experiments on hippocampal slices analyzing GluA1 levels at 40 min after HFS delivery. We assessed levels of surface GluA1 using cell-surface biotinylation of hippocampal slices before and 40 min after HFS and found that in WT mice HFS application increases the amount of cell-surface GluA1 whereas cell-surface GluA1 levels remain constant in DNA-PKcs $-/-$ tetanized slices (new Fig. 4E). We added these new results in the text, Results Section “DNA-PKcs $-/-$ mice show impairment in synaptic transmission, Long-Term Potentiation, and postsynaptic protein expression” paragraph and in the Discussion Section.

Comment 6. Also in Fig. 4d, it is odd that LTP treatment appears to decrease the levels of synaptic PSD-95. This should be quantified and discussed.

Answer: We apologize with Referee 1 for reporting a wrong information by choosing this figure. Indeed, among the 4 experiments performed, we wrongly selected this one because of the higher trafficking of GluA1 after cLTP induction in cortical cultures. However, this experiment is not the most representative regarding the total PSD-95 protein level after F/R treatment. Actually, PSD-95 levels are not reduced after LTP induction and we even find it increased following HFS delivery, as shown in Fig. 7d previous version of the manuscript. We have now changed Fig. 4d, showing a different more representative experiment.

Comment 7. The Flag staining for PSD-95 constructs look odd. PSD-95 localizes strongly to synapses even when overexpressed. However, in Fig 8c the FLAG constructs do not show synaptic localization. Adding to

this concern is Fig 9c, where FLAG tagged PSD-95 constructs show little colocalization with Homer. This raises concerns over the specificity of the staining.

Answer: We have acquired new images at high resolution by using THUNDER microscope of DIV 21 primary cortical cultures infected with rAAV FLAG constructs. The new images in Fig. 8C show a punctate neurite staining similar to synaptic proteins. Moreover, we changed Fig. 9B with new images from neurons with lower expression levels of FLAG constructs along neurites, thus showing a strong colocalization with Homer 1, particularly in spines.

Comment 8. The differential effects of S308 vs T87 phosphorylation on dendritic complexity and LTP are not explained. If both sites enhance PSD-95 stability, why is S308E able to restore dendritic spine density but not dendritic complexity or LTP?

Answer: We agree with Referee 1 and we now discuss the different effects of S308E and T87E overexpression in KO neurons and mice.

We added in the text, Discussion Section, the following sentences: “This result is not astonishing considering that PSD-95, likely through its multiple protein-protein interaction motifs, regulates many distinct and separable aspects of synapse structure, function, and plasticity. The enhanced PSD-95 stability of the two constitutively phosphorylated mutants could not be the only factor determining dendritic complexity or LTP in neuronal cultures or in r-AAV injected DNA-PKcs^{-/-} mice. Indeed, it is known that PSD-95 shapes dendrite architecture in cultured neurons by altering microtubule dynamics via interaction with different cytoskeleton proteins (Brenman et al., J Neurosci (1998) 18: 8805-13; Charych et al., J Neurosci (2006) 26: 10164-76; Sweet et al. J Neurosci (2011) 31: 1038-47). Moreover, the spine formation/maturation and dendritic arbor development are not necessarily concurrent phenomena [Bustos et al., PLoS One (2014) 9: e94037]. Therefore, it is plausible that the different mutations mimicking PSD-95 phosphorylation, by causing conformational changes, have independent effects on spine formation/maturation, dendritic complexity and LTP due to different regulation of PSD-95 interactions with other partners. In particular, structural analysis could achieve a detailed understanding of the role of these phosphorylations not only in PSD-95 protein stability, but also in the regulation of PSD-95 interactions with other partners. Thus, further studies are needed to unveil the role of PSD-95 phosphorylation at S308 residue”.

Comment 9. The relationship between the S308 and T87 phospho-sites on PSD-95 stability is also unclear. It appears that phosphorylation of either site alone is sufficient to enhance PSD-95 stability, while dephosphorylation of either site is also sufficient to destabilize PSD-95. This does not make much sense. What happens to PSD-95 stability if there is a T87E, S308A double mutation, or a T87A, S308E double mutation?

Answer: We thank Referee 1 for this interesting comment. When we overexpress single E mutation (either T87E or S308E) in DNA-PKcs^{-/-} cells we actually have a double mutant E/A because the lack of DNA-PKcs in KO neurons prevents the phosphorylation of the other residue (i.e. T87E/S308 not phosphorylated; T87

not phosphorylated/S308E) and the presence of E mutation is sufficient to enhance PSD-95 stability. In contrast, when we overexpress single A mutation in WT cells, neurons have both the abundant endogenous PSD-95 wt and the exogenous PSD-95 with a residue not mutated and thus phosphorylatable (i.e. T87 phosphorylatable/S308A; T87A/S308 phosphorylatable). In this context, DNA-PKcs kinase could have greater affinity for the endogenous protein, thus phosphorylating it preferentially compared to the exogenous. Therefore, it is possible that the higher affinity substrate is able to suppress phosphorylation of its lower affinity counterpart (Sommese and Sivaramakrishnan J Biol Chem. (2016) 291(42): 21963–21970). This could explain why in WT neurons, the mutant A is not stable despite having a residue still phosphorylatable.

Comment 10. In Fig 10d, it would be important to compare the data with WT in terms of the level of LTP induction and maintenance. In addition, the ability for overexpressed PSD-95 constructs to rescue LTP is interesting considering that previous studies showed overexpression of PSD-95 precludes LTP induction. This should be discussed.

Answer: We have now added in the graph of Fig 10d (now 10F) the bar graph (with grey background to indicate that injected and no injected mice are not statistically compared) showing fEPSP potentiation in WT and DNA-PKcs $-/-$ slices to compare rescue of LTP levels in the T87E rAAV injected mice with WT and DNA-PKcs $-/-$ mice. We modified the text and figure legend accordingly.

We agree with Referee 1 that previous studies showed that PSD-95 overexpression occludes LTP, possibly due to saturation of receptor signaling (see for example Stein et al., J. Neurosci (2003), 23(13):5503-6). It is noteworthy that those studies were carried out in WT animals. On the contrary, we overexpress PSD-95 wt in DNA-PKcs $-/-$ mice and find that it induces a small, not statistically significant increase of LTP. This effect might be due to the fact that DNA-PKcs $-/-$ neurons have weak synapses, that can still be slightly potentiated as a result of increasing the number of AMPA receptors delivered to synapses by PSD-95 wt overexpression (Ehrlich and Malinow, J Neurosci (2004) 28;24(4):916-27).

Comment 11. The error bars on many of the graphs look very odd. The authors report that the error bars represent SEM (graphs show mean {plus minus} SEM). However, many of errors bars (see Fig 2c, 3h, 4a, 4c, 7d) are oddly positioned with the center of the error bar not at the mean. The positions of these error bars were also not consistent from bar to bar. This raises concerns about the possibility that these error bars were not generated automatically by the statistical software but rather manually added onto the graph later.

Comment 12. Related to the above issue, data reporting needs to be more transparent with individual data points shown on all bar graphs, similar to what they did in Fig 3e, g, and Fig 10d for their electrophysiology data.

Answer to comment 11 and 12: We are really sorry for this inconvenience. The error bars were not manually added onto the graphs and the odd effect is probably due to a problem with pdf conversion. We

have now changed the bar graphs by adding the individual data points to be more transparent with data presentation.

Referee #2:

Comment: The authors have extensively revised the manuscript by adding the new experiments to address my comments. I recommend publication of the manuscript.

Answer: We thank the reviewer for appreciating our revised manuscript.

Referee #3:

Specific comments:

Comment 1. Generally, the data presentation and discussion should be improved, also as suggested by Reviewer #1.

Answer: We hope that our manuscript is now considerably improved thanks to new data, changes in figures and discussion of new different points provided, as suggested by the Referees.

Comment 2. DNA-PKcs^{-/-} mice originally was designed for a kinase dead mutant (DNA-PK-KD). Unfortunately, due to the protein stability (the truncation mutant is often unstable), it behaves like a knockout (KO, also confirmed in Fig 3a). The whole manuscript is based on this mouse model and assumed the study dealing with only DNA-PK kinase activity. However, without DNA-PK KO control, one cannot rule out the contribution by its scaffold, like the case of ATR and ATM, in other studies

Answer: We are very grateful to Referee 3 for giving us the opportunity to share our unpublished data derived from experiments performed in the past and presented in a Poster at FENS, carried out using a different model, SCID mice. The SCID mutation results in truncation of DNA-PKcs protein that is still expressed in most SCID cell lines (Beamish et al., Nucleic Acids Res (2000) 28(7):1506-13). By using this model, we obtained similar electrophysiological results presented in this manuscript thus confirming the role of kinase activity of DNA-PKcs, and not of its scaffold, in synaptic plasticity. Please see below some figures from Poster.

DIPARTIMENTO
NEUROSCIENZE

Figure for referee with unpublished data and its description has been removed upon request by the authors.

Comment 3. Foskolin+Rolipram (R/F) is known to stimulate synaptic activity in LTP. One basal line of this treatment should be documented in order to show the treatment worked.

Answer: Induction of cLTP by F/R treatment is a very well-established protocol to study the AMPA receptor trafficking (Otmakhov et al., J Neurophysiol (2004)91: 1955-1962; Oh et al., J Biol Chem (2006) 281: 752-758; Mollinari et al., Cell Death Dis (2015) 6: e1622). In these experiments, we exploited this protocol to evaluate GluA1 delivery into synaptic compartment and compare WT and DNA-PKcs $-/-$ neurons. The effectiveness of the treatment is shown in Fig. 4D where we demonstrate that, following F/R incubation of primary neuronal cultures, GluA1 trafficking increases as compared with DMSO basal level, as reported in the literature.

Comment 4. Fig 1d. Due to the resolution limitation, one cannot view clear localization pattern as the authors described, “periphery” or “overlapping”. DNA-PKcs $-/-$ control is needed here

Answer: We provide single channel images as also requested by Referee 1. In particular, we modified Figure 1D showing single channel images for colocalization of DNA-PKcs with synaptic markers in neurites and modified the Figure legend accordingly. Moreover, we provide Figure EV2 showing single channel images of the whole neuron for each synaptic protein analyzed, colocalizing with DNA-PKcs. We specified in the text, Result section, “Characterization of DNA-PKcs in brain and neurons” paragraph: “It is notable that DNA-PKcs co-localizes mostly to the periphery of presynaptic proteins, whereas co-localization with postsynaptic proteins, particularly with PSD-95, appears in close apposition (Fig. 1D insets: white dotted edges in the red channels highlight the position of DNA-PKcs protein with respect to synaptic proteins)”.

Comment 5. Fig 3h. It is odd/interesting that DNA-PKcs $-/-$ neurons show high levels of pS473-AKT. This residue has been shown to be a phosphorylation site by DNA-PK, and perhaps also by ATM or ATR. What is going on here? Thus, ATM or ATR inhibitors can be used to test the substrate specificity of DNA-PK. In this regard, DNA-PK, ATR and ATM share many substrates and it would be interesting to examine whether the phosphorylation status of other proteins in the same figure as well as PSD95 when ATM/ATR are inhibited.

Answer: We thank Referee 3 for this observation. It is known from the literature that there are different kinases directly phosphorylating S473-AKT during LTP. In particular, it is known that mTOR signaling is necessary for the induction of LTP in the CA1 region of the hippocampus (Henry et al., Mol Brain. (2017) 10: 50) and mTORC2 phosphorylates the Ser residues at the C-terminal regulatory domain (Ser473 in PKB α , Ser474 in PKB β , and Ser472 in PKB γ), which provides additional 10-fold increase in activation of AKT (Martinez Calejman et al., Nature Communications (2020) 11: 575). The Ser residues can also be phosphorylated by other kinases such as integrin-linked kinase (Liu et al., Neuro Oncol. (2014) 16(10): 1313–1323). These data would account for the phosphorylation of pS473-AKT in DNA-PKcs $-/-$ mice. Therefore, it is not surprising that DNA-PKcs $-/-$ neurons show S473-AKT phosphorylation. Moreover, we

are very grateful to Referee 3 for giving us again the opportunity to share our unpublished data presented in the FENS Poster obtained using SCID mice that, although having dramatically impaired kinase activity, still have phosphorylation levels of S473-AKT comparable to WT mice, following HFS delivery (Beamish et al., Nucleic Acids Res (2000) 28(7):1506-13; Vemuri et al., Cell Death Differ (2001) 8: 245-255). Please see below the figure from Poster.

Figure for referee with unpublished data and its description has been removed upon request by the authors.

Regarding PSD-95 phosphorylation by ATM/ATR, we agree with Referee 3 that this would be a very interesting information. However, we think that the simple inhibition of ATM/ATR would not be exhaustive and, to properly demonstrate PSD-95 phosphorylation by ATM/ATR, a set of experiments including kinase

assay, mass spectrometry, specific antibody production, and so on, similarly to the work done for DNA-PKcs, would be required. Moreover, ATM/ATR are found in the synaptic vesicles where form a complex with synaptic vesicle proteins, and in ATM- or ATR-deficient neurons, spontaneous vesicle release is reduced, thus suggesting their main role in the pre-synaptic compartment.

Comment 6. The authors tend to claim that phosphorylation of PSD95 by DNA-PK is F/R dependent. However, most of experiments, particularly in testing phosphor mutant PSD95, in Fig 7e, 8d, 9 and 10 did not use F/R treatment.

Answer: To evaluate whether PSD-95 is phosphorylated by DNA-PKcs following synaptic activity, we analyzed phosphorylation changes at S308 and T87 residues of PSD-95 at different time points after HFS delivery in *ex vivo* hippocampal slices and found that the phosphorylation is associated with LTP (Fig. 7d). F/R treatment of neuronal primary culture was instead the protocol used to evaluate GluA1 delivery into synaptic compartment (Fig. 4d). The PSD-95 stability experiments shown in Fig. 7e and 8d were carried out in basal condition to study the effect of the lack of DNA-PKcs kinase activity (Fig. 7e) and the rescue of PSD-95 stability and neuronal morphology by the phosphomimetic mutants (Fig. 8d and 9). In Fig. 10 we demonstrate the rescue of LTP induced by HFS in *ex vivo* hippocampal slices from rAAV-infected KO mice.

We hope we have now clarified that the phosphorylation levels of PSD-95 has been conducted in *ex vivo* hippocampal slices using HFS delivery protocol.

Comment 7. The description or comparisons of quantifications of some figures in the text disturbs reading; thus the details can move to the figure legends.

Answer: We have done it in those paragraphs where quantification values were more disturbing for reading.

Comment 8. Minor. For example, NHEJ should be non-homologous end joining, not “not”.

Answer: Done.

We thank Referees for their criticisms and suggestions and hope that our revisions are now considered satisfactory for having the manuscript published in EMBO Reports.

Sincerely,

Daniela Merlo, PhD

Dear Daniela,

Thank you for the submission of your revised manuscript. We have now received the enclosed reports from the referees that were asked to assess it. Referee 1 still has one more suggestion that I would like you to address and incorporate, and please also address referee 3's comments in the ms text. Referee 1 agrees with us in the cross-comments that no further experimentation is required at this point.

As soon as you will have addressed these last points, we can proceed with the official acceptance of your manuscript.

In our online system, you should be able to move forward all existing ms files to the new, final submission, so there is no need to upload all single files again. You can then modify the files in the new submission.

Referee #1:

The authors have made additional revisions to the manuscript and have addressed most of my concerns. I only have one additional comment. For all of the image quantifications (e.g. Figure 1E, 2A, among others), the bar graphs typically only show three data points per condition, which represents three independent experiments according to the figure legends. However, it is unclear how many cells were analyzed per condition. To get a better sense of data distribution and sample size, the authors should show all the data points instead of just the average of each independent experiment.

Referee #3:

The authors have addressed most of my comments in the revision and rebuttal. However, the original points below need to be dealt with seriously in revision.

Point #2. In response to my comment, the authors cited their unpublished work (from a Poster) based on a SCID mouse model, which carries a truncated DNA-PKcs, which showed similar electrophysiological properties as DNA-PKcs^{-/-} neurons. Firstly, this observation has not been formally documented in the current manuscript, nor in publication. Secondly, although the SCID model (which has enzymatic deficiency, while retains partial protein) showed a similar phenotype as DNA-PKcs^{-/-} mice, it cannot replace DNA-PKcs KO control (lacking both the protein and enzymatic activity), which altogether would help to differentiate the contribution in their observations by DNA-PKcs' scaffold or by its enzymatic activity only.

Point #5. It is possible that high levels of pS473-AKT in DNA-PKcs^{-/-} neurons could be caused by other kinases. However, all their arguments, by citing other kinase and other substrates, did not disprove if the S473-AKT residue is not phosphorylated by DNA-PK, but perhaps by ATM or ATR. The authors used their data in SCID data (in a Poster) to argue for their DNA-PKcs^{-/-} data. However, in both cases, DNA-PKcs^{-/-} lacks kinase activity, which cannot explain the substrate specificity, in this case pS473-AKT, which can be phosphorylated by ATM or ATR, as previous papers claimed. Therefore, my original comment still holds true. Interestingly, because ATM and ATR are found in synaptic vesicles, this control experiment is important to confirm the substrate (PSD95) specificity of DNA-PK, but not by other PI3KK members.

Cross-comments from referee 1:

I read Ref 3's comments and the relevant data again, and I do not feel it's critical for them to exclude ATM and ATR. As the authors have stated, there are several possible upstream kinases for Ser473 Akt, so it is not surprising to me that they still see Akt S473 phosphorylation in the DNA-PKcs^{-/-}. I don't see why they need to exclude ATM and ATR. These are only two of several possible upstream kinases, and identifying the exact upstream kinase for S473 Akt is not relevant to the main conclusion of this paper. Ref 3 also mentioned that ruling out ATM and ATR is important for demonstrating the substrate specificity of PSD-95. However I feel that the authors already demonstrated the specificity, and I do not understand why ruling out ATM and ATR is important here.

18th June, 2024

Esther Schnapp
Senior Editor
EMBO Reports

Dear Editor,

below we present our point-by-point response to Referee's comments.

Referee #1:

Comment 1. The authors have made additional revisions to the manuscript and have addressed most of my concerns. I only have one additional comment. For all of the image quantifications (e.g. Figure 1E, 2A, among others), the bar graphs typically only show three data points per condition, which represents three independent experiments according to the figure legends. However, it is unclear how many cells were analyzed per condition. To get a better sense of data distribution and sample size, the authors should show all the data points instead of just the average of each independent experiment.

Answer: We thank Referee 1 for appreciating our revised manuscript.

We have now showed for all of the image quantifications the data distribution for each independent experiment, in addition to the original graph. Moreover, we have specified in the figure legends the sample size.

Referee #3:

The authors have addressed most of my comments in the revision and rebuttal. However, the original points below need to be dealt with seriously in revision.

Comment 1. In response to my comment, the authors cited their unpublished work (from a Poster) based on a SCID mouse model, which carries a truncated DNA-PKcs, which showed similar electrophysiological properties as DNA-PKcs^{-/-} neurons. Firstly, this observation has not been formally documented in the current manuscript, nor in publication. Secondly, although the SCID model (which has enzymatic deficiency, while retains partial protein) showed a similar phenotype as DNA-PKcs^{-/-} mice, it cannot replace DNA-PKcs KO control (lacking both the protein and enzymatic activity), which altogether would help to differentiate the contribution in their observations by DNA-PKcs' scaffold or by its enzymatic activity only.

Answer: We thank Referee 3 for appreciating our revised manuscript.

We think that the SCID mouse model can help to differentiate the contribution of DNA-PKcs' scaffold from its enzymatic activity since it is known from the literature that SCID mutation truncates only the extreme C terminus of the kinase domain by approximately 8 kDa thus leaving intact most of the entire DNA-PKcs protein. Indeed, Western blots from previous articles (Beamish et al., 2000; Vemuri et al., 2001) show the presence of a protein with the expected molecular weight both in cells and brain tissues (about 460 kDa).

We have now mentioned our results on SCID mice (Cardinale et al., 2006, FENS Poster) in the Discussion section as follows: “In accordance with this hypothesis, our unpublished data carried out using a SCID mouse model, having a 8kDa COOH terminus truncation of DNA-PKcs protein with no enzymatic activity, resulted in similar electrophysiological defects thus confirming the role of kinase activity of DNA-PKcs in synaptic plasticity.”

Comment 2. It is possible that high levels of pS473-AKT in DNA-PKcs^{-/-} neurons could be caused by other kinases. However, all their arguments, by citing other kinase and other substrates, did not disprove if the S473-AKT residue is not phosphorylated by DNA-PK, but perhaps by ATM or ATR. The authors used their data in SCID data (in a Poster) to argue for their DNA-PKcs^{-/-} data. However, in both cases, DNA-PKcs^{-/-} lacks kinase activity, which cannot explain the substrate specificity, in this case pS473-AKT, which can be

phosphorylated by ATM or ATR, as previous papers claimed. Therefore, my original comment still holds true. Interestingly, because ATM and ATR are found in synaptic vesicles, this control experiment is important to confirm the substrate (PSD95) specificity of DNA-PK, but not by other PI3KK members.

Answer: We thank Referee 3 for bringing this issue to our attention.

We think that excluding ATM and ATR as the kinases responsible for phosphorylation of Akt at Ser473 in DNA-PKcs $-/-$ brain slices is not relevant for demonstrating the substrate specificity of PSD-95 for DNA-PKcs. Indeed, we have demonstrated that DNA-PKcs phosphorylates PSD-95 by kinase assays with both recombinant and endogenous immunoprecipitated DNA-PKcs protein.

Regarding Akt phosphorylation at Ser473 during LTP, we have now added in the text, Result section, “DNA-PKcs $-/-$ mice show impairment in synaptic transmission, Long-Term Potentiation, and postsynaptic protein expression” paragraph the following sentence: “Although it has been shown that DNA-PKcs phosphorylates Akt at Ser473 (Feng et al., 2014), this result is not surprising since it is known from the literature that there are different kinases directly phosphorylating S473-Akt during LTP (Henry et al., 2017; Liu et al., 2014)”.

We thank all Referees for carefully reading our manuscript and for giving constructive comments which helped improving our study.

Sincerely,

Daniela Merlo, PhD

Dr. Daniela Merlo
Istituto Superiore di Sanità
Neuroscience
Rome
Italy

Dear Daniela,

I am very pleased to accept your manuscript for publication in the next available issue of EMBO reports. Thank you for your contribution to our journal.
